# PRIVATE MULTI-WINNER VOTING FOR MACHINE LEARNING

## ABSTRACT

Private multi-winner voting is the task of revealing $k$-hot binary vectors that satisfy a bounded differential privacy guarantee. This task has been understudied in the machine learning literature despite its prevalence in many domains such as healthcare. We propose three new privacy-preserving multi-label mechanisms: Binary, $\tau$, and Powerset voting. Binary voting operates independently per label through composition. $\tau$ voting bounds votes optimally in their $\ell_2$ norm. Powerset voting operates over the entire binary vector by viewing the possible outcomes as a power set. We theoretically analyze tradeoffs showing that Powerset voting requires strong correlations between labels to outperform Binary voting. We use these mechanisms to enable privacy-preserving multi-label learning by extending the canonical single-label technique: PATE. We empirically compare our techniques with DPSGD on large real-world healthcare data and standard multi-label benchmarks. We find that our techniques outperform all others in the centralized setting. We enable multi-label CaPC and show that our mechanisms can be used to collaboratively improve models in a multi-site (distributed) setting.

## 1 INTRODUCTION

Differential privacy techniques for machine learning have predominantly focused on two types of mechanisms: for a given query, release either a noisy estimate using the Gaussian mechanism (Abadi et al., 2016b) or the noisy $\arg\max$ (Papernot et al., 2017). These two mechanisms appeal well to *single-label* classification per input (a.k.a. multi-class classification), which is a widely studied setting of supervised learning where a given input has only one class present. The Gaussian mechanism can enable arbitrary queries of data, for instance, differentially private stochastic gradient descent (DPSGD), which returns noisy gradients (Abadi et al., 2016b). Instead, the Gaussian noisy $\arg\max$ mechanism can only reveal the max count, which is used in the canonical *Private Aggregation of Teacher Ensembles* (PATE) (Papernot et al., 2017).

In contrast, more real world tasks, such as *multi-label* classification (Tsoumakas & Katakis, 2007), can be modeled using multi-winner elections (Elkind et al., 2017; Faliszewski et al., 2017), where more than one candidates may win, simultaneously. Outside of the obvious situation of elections, other multi-winner election scenarios include canonical computer visions tasks like object recognition, which require models to recognize all the objects present in an image (Boutell et al., 2004). Further, multi-winner problems arise when inferring the topics written about in a corpus of text (document): e.g., a news article may discuss one or more topics like politics, finance, or education. Another principal setting is healthcare, in which patient data (*e.g.,* symptom reports or X-rays) may be indicative of multiple conditions (Irvin et al., 2019; Johnson et al., 2019).

We thus focus on creating private mechanisms for releasing the outcome of a multi-winner election. We first formalize the multi-winner election and then provide a solution, termed Binary voting, where we apply a single-winner election mechanism independently to each of the $k$ candidates. Though simple, we prove that it is optimal when there is a lack of correlation among the outcomes of particular candidates. We then derive tighter data-independent privacy bounds for multi-winner elections by considering the situation where each voter is limited in their ($\ell_2$) votes for candidates, which we call $\tau$ voting. By casting multi-winner elections to an analogous task of single-winner elections, we create a third mechanism termed Powerset voting, which reveals the result for all $k$ candidates jointly.

By replacing the noisy $\arg\max$ mechanism used in (single-label) PATE, we create multi-label PATE which enables multi-label privacy-preserving classification with semi-supervised machine learning.

We empirically evaluate our algorithms on large-scale datasets, including the common multi-label benchmark Pascal VOC and three healthcare datasets, CheXpert (Irvin et al., 2019), MIMIC (Johnson et al., 2019), and PadChest (Bustos et al., 2020). We compare our work against DPSGD, which can be directly applied to multi-label machine learning, and find that our approaches achieve new state-of-the-art privately trained models, with 85% AUC (17% better than DPSGD) on the Pascal VOC dataset. Because many multi-label settings may benefit from distributed, multi-site learning (e.g., healthcare), we further integrate our methods with the collaborative protocol of Choquette-Choo et al. (2021). We find that training with our mechanisms can improve model performance significantly on large, real-world healthcare data with sensitive attributes.

Our main contributions are as follows:

1. We create three new DP aggregation mechanisms for private multi-winner voting: Binary voting, $\tau$ voting, and Powerset voting. We theoretically analyze the privacy guarantees of these mechanisms, finding that Binary voting performs better unless there is high correlation among labels.

2. We integrate our mechanisms with PATE, enabling private multi-label semi-supervised learning that achieves SOTA performance on large real-world tasks: Pascal VOC and 3 healthcare datasets.

3. We enable multi-label collaborative learning in the multi-site scenario by integrating our mechanisms into CaPC. We find that multi-label CaPC significantly improves model performance.

## 2 Background

### 2.1 Differential Privacy for Machine Learning

Differential Privacy (DP) is a canonical framework for measuring the privacy leakage of a randomized algorithm (Dwork et al., 2006). It requires the mechanism, the training algorithm in our case, to produce statistically indistinguishable outputs on any pair of adjacent datasets, i.e., datasets differing by only one and any data point. This bounds the probability of an adversary inferring properties of the training data from the mechanism's outputs.

**Definition 1** (Differential Privacy). *A randomized mechanism $\mathcal{M}$ with domain $\mathcal{D}$ and range $\mathcal{R}$ satisfies $(\varepsilon, \delta)$-differential privacy if for any subset $\mathcal{S} \subseteq \mathcal{R}$ and any adjacent datasets $d, d' \in \mathcal{D}$, i.e. $\|d - d'\|_1 \leq 1$, the following inequality holds:* $\Pr[\mathcal{M}(d) \in \mathcal{S}] \leq e^{\varepsilon} \Pr[\mathcal{M}(d') \in \mathcal{S}] + \delta$.

We use Rényi Differential Privacy (RDP) (Mironov, 2017) which enables tighter accounting for our mechanisms using Gaussian noise because of its use of the Rényi-divergence to bound privacy loss. RDP is a generalization of pure, $\delta = 0$-DP so we convert to and report final budgets using $(\varepsilon, \delta) - DP$.

**Definition 2.** *Rényi Differential Privacy (Mironov, 2017). We say that a mechanism $\mathcal{M}$ is $(\lambda, \varepsilon)$-RDP with order $\lambda \in (1, \infty)$ if for all neighboring datasets $X, X'$:*

$$D_\lambda(\mathcal{M}(X)\|\mathcal{M}(X')) = \frac{1}{\lambda - 1} \log E_{\theta \sim \mathcal{M}(X')}\left[\left(\frac{p_{\mathcal{M}(X)}(\theta)}{p_{\mathcal{M}(X')}(\theta)}\right)^\lambda\right] \leq \varepsilon$$

It is convenient to consider RDP in its functional form as $\varepsilon_{\mathcal{M}}(\lambda)$, which is the RDP $\varepsilon$ of mechanism $\mathcal{M}$ at order $\lambda$. We now state the result which our privacy analysis builds on.

**Lemma 2.1.** *RDP-Gaussian mechanism (for a single-label setting) (Mironov, 2017). Let $f : \mathcal{X} \to \mathcal{R}$ have bounded $\ell_2$ sensitivity for any two neighboring datasets $X, X'$, i.e., $\|f(X) - f(X')\|_2 \leq \Delta_2$. The Gaussian mechanism $\mathcal{M}(X) = f(X) + \mathcal{N}(0, \sigma^2)$ obeys RDP with $\varepsilon_{\mathcal{M}}(\lambda) = \frac{\lambda \Delta_2^2}{2\sigma^2}$.*

Another notable advantage of RDP over $(\varepsilon, \delta)$-DP is that it composes naturally. This will be helpful when analyzing our approach which repeatedly applies the same private mechanism to a dataset.

**Lemma 2.2.** *RDP-Composition (Mironov, 2017). Let mechanism $\mathcal{M} = (\mathcal{M}_1, \ldots, \mathcal{M}_t)$ where $\mathcal{M}_i$ can potentially depend on outputs of $\mathcal{M}_1, \ldots, \mathcal{M}_{i-1}$. $\mathcal{M}$ obeys RDP with $\varepsilon_{\mathcal{M}}(\cdot) = \sum_{i=1}^{t} \varepsilon_{\mathcal{M}_i}(\cdot)$.*

## 2.2 PATE AND CaPC

**PATE** (Private Aggregation of Teacher Ensembles) is a model agnostic algorithm for training ML models with DP guarantees (Papernot et al., 2017). Rather than training a single model on a dataset, PATE trains an ensemble of models from (disjoint) partitions of the dataset. Each model, called a teacher, is then asked to predict on a test input and to vote for one class. Teacher votes are collected into a histogram where $n_i(x)$ indicates the number of teachers who voted for class $i$. To preserve privacy, PATE relies on the noisy argmax mechanism and only reveals a noisy aggregate prediction (rather than revealing each prediction directly): $\text{argmax}\{n_i(x) + \mathcal{N}(0, \sigma_G^2)\}$, where the variance $\sigma_G^2$ of the Gaussian controls the privacy loss of revealing this prediction. A student model learns in a semi-supervised fashion by querying the ensemble of teachers for noisy labels, where each label incurs additional privacy loss. Loose data-independent guarantees are obtained through advanced composition (Dwork et al., 2014) and tighter data-dependent guarantees (see Appendix Section L) are possible when there is high consensus among teachers on the predicted label (Papernot et al., 2018). To further reduce privacy loss, *Confident GNMax* only reveals predictions that have high consensus: $\max_i\{n_i(x)\}$ is compared against a noisy threshold also sampled from a Gaussian with variance $\sigma_T^2$. PATE was created for single-label classification; our work extends it to the multi-label setting.

**CaPC** (Confidential and Private Collaboration) is a distributed collaborative learning framework that improves on PATE (Choquette-Choo et al., 2021). In addition to protecting the privacy of training data through differential privacy, CaPC also protects the confidentiality of test data and model parameters by introducing new cryptographic primitives. *Confidentiality* refers to the notion that no other party (or adversary) can view, in plaintext, the data of interest—it protects the inputs to our mechanism. This differs from *differential privacy* which protects what can be inferred (in our case about the training set for PATE models) from the outputs of our mechanism. Both protect data privacy in different ways. In CaPC, each teacher model is trained by a distinct (and different) protocol participant. The querying party initiates the protocol by (1) encrypting unlabeled data and the other (answering) parties return an encrypted label. Then, (2) the encrypted vote is secretly shared with both the answering party and the Privacy Guardian, where the Privacy Guardian follows PATE to enforce DP. Finally, (3) the Privacy Guardian engages in the secure computation with the querying party and returns a DP vote to the querying party. The detailed CaPC protocol can be found in Algorithm 1 of Appendix J.

## 2.3 PRIOR WORK ON MULTI-LABEL CLASSIFICATION

The closest to our work is that of Zhu et al. (2020), which provides a privacy-preserving nearest neighbors algorithm formulated using PATE. The nearest-neighbor training points are used as teachers, where their labels are the votes. While this work is applicable to the multi-label setting, it assumes the existence of a publicly-available embedding model, which is used to project high-dimensional data onto a sufficiently-low-dimensional manifold for the meaningful computation of distances (to determine the nearest neighbors). Instead, our approach does not make this assumption. Furthermore, we also show in Section 3 that clipping votes to the $\ell_1$ ball, as done in Zhu et al. (2020), is not optimal—given that noise is sampled from a Gaussian distribution. Instead, we obtain tighter bounds by clipping votes to an $\ell_2$ ball. The multi-label voting problem can also be formulated as simultaneously answering multiple counting queries which has been studied extensively in the theoretical literature. Though data-independent optimal error bounds for the privacy loss were proven by Dagan & Kur (2020), these mechanisms do not generally have data-dependent bounds and have yet to show promise empirically in settings such as multi-label voting. Another alternative multi-label method is differentially private stochastic gradient descent (DPSGD), which was proposed by Abadi et al. (2016a) and applied to the multi-label setting by Zhang et al. (2021) to train small multi-label models ($k = 5$ labels). We compare directly with their work and find that our methods perform better. We present a detailed description of the related work in Appendix E. Finally, we stress that our evaluation includes large-scale healthcare datasets with a natural need for privacy and the additional challenge of being severely imbalanced.

## 3 DIFFERENTIAL PRIVACY MECHANISMS FOR MULTI-WINNER ELECTION

Multi-winner election systems are direct generalizations of popularly used single-winner elections. They are adopted commonly in real-life elections and their comprehensive analysis can be found

in Elkind et al. (2017) and Faliszewski et al. (2017). Here, we formally define a multi-winner system and propose DP mechanisms to facilitate the private release of the outcome of a multi-winner election.

Compared to single-winner elections, multi-winner elections are more challenging for designing a DP mechanism: instead of outputting a single scalar, multi-winner elections output a higher-dimensional vector and thus require more noise to achieve the same level of privacy. DP mechanisms for single-winner voting have been well studied before, where the noisy $\arg\max$ achieves minimal privacy loss due to its tight sensitivity as a result of *information minimization* (Dwork et al., 2014). However, to the best of our knowledge, no prior work formally defines and analyzes DP mechanisms for multi-winner election systems, hence we provide Definition 3 and the analysis. We provide formal data-dependent and data-independent privacy loss bounds in Section 4.

**Definition 3.** *$\theta$-multi-winner election. Given a set of $n$ voters, each with a ballot of $k$ candidates (coordinates) $\boldsymbol{b} \in \{0,1\}^k$ such that $\|\boldsymbol{b}\|_2 \leq \theta$. The outcome of a $\theta$-multi-winner election with threshold $T$ is defined as*

$$f(\boldsymbol{b}_1, \ldots, \boldsymbol{b}_n) \triangleq \left\{ i : \sum_{j=1}^{n} \boldsymbol{b}_j[i] > T \right\} \in \{0,1\}^k,$$

where $T = \frac{n}{2}$ and with the binary decision per label $i$, $\sum_{j=1}^{n} \boldsymbol{b}_j[i]$ represents the number of positive votes.

### 3.1 DP MECHANISMS FOR $\infty$-MULTI-WINNER ELECTION

In an $\infty$-multi-winner[1] election, each elector can vote for any candidates in their ballots. We propose to decompose the multi-winner election into $k$ separate single-winner elections, where we instead apply the well-studied noisy $\arg\max$ separately to each of the $k$ single-winner elections. We call this Binary voting and state formally in Definition 4 below. For $m$ instances of $\infty$-multi-winner elections, the total privacy loss of this mechanism follows Lemma 2.2 which can then be tightly expressed as $(\varepsilon, \delta)$-DP guarantee for a chosen $\delta$.

**Definition 4** (DP (Noisy Argmax) Binary Voting Mechanism). *For an $\infty$-multi-winner election, the binary voting mechanism is*

$$\mathcal{M}_\sigma(\boldsymbol{b}_1, \ldots, \boldsymbol{b}_n) \triangleq \left\{ i : \left( \sum_{j=1}^{n} \boldsymbol{b}_j[i] + \mathcal{N}(0, \sigma_G^2) \right) > T_i \right\} \in \{0,1\}^k,$$

where $T_i = n - \sum_{j=1}^{n} \boldsymbol{b}_j[i]$ is the number of negative votes, which enables more noise to be added than when $T = \frac{n}{2}$. Though there are many potential per-candidate Binary voting mechanisms, we base ours on the noisy argmax mechanism because of its advantageous privacy guarantees stemming from its well-understood sensitivity: changing the training set to any adjacent one can only modify a single ballot. We also leverage propose-test-release (Dwork & Lei, 2009) shown in Algorithm 2 of Appendix K. Next, we show that Binary voting is optimal in the coordinate-independent setting.

**Definition 5.** *A function $f(X) : \mathcal{R}^n \to \mathcal{R}^k$ is coordinate-independent if the $i$'th output coordinate of $f$, $f_i$, is determined by a unique subset $P_i(X)$ of the input $X$, such that $\cap_{i=1}^k P_i = \emptyset$. $P_i(X)$ indexes (partitions) the input $X$ uniquely for each $i$ and returns the chosen subset; the chosen columns are dependent only on $i$, not the particular dataset $X$.*

We first build an intuition for our guarantee and then provide the formal statement (in Proposition 3.1). Recall from Section 2 that the crux of DP analysis is to bound the sensitivity of the random mechanism $\mathcal{M}$, which is the noisy output of the function[2] $f$.

**Definition 6.** *Sensitivity is the maximum deviation $\Delta_p f = \max\limits_{(X,X'):||X-X'||_1=1} ||f(X) - f(X')||_p$ of $f$'s output for any adjacent pair of inputs $X$ and $X'$, i.e., have a Hamming distance $||X - X'||_1$ of 1, on their rows (Dwork et al., 2014).*

---

[1]We use $\infty$ to emphasize that any vote $\boldsymbol{b} \in \{0,1\}^k$ is allowed but a tighter bound exists: $\|\boldsymbol{b}\|_2 \leq \sqrt{k} < \infty$.

[2]Because the random mechanism differs minimally from the function, we use the two interchangeably.

Observe that output $f_i$ of a Binary voting mechanism for a coordinate $i$ depends only on the corresponding $i-$th coordinate of the input ballot because we apply Binary voting independently to each candidate. For any mechanism $f$, $\Delta_p f$ is maximized when the mechanism's output for each coordinate $i$ is flipped from predicted to not predicted or vice versa. The pair of teacher votes achieving this differ in each coordinate $i$. This also means that any coordinate-independent mechanism has sensitivity at least as large as the one of Binary voting mechanism because all $k$ coordinates can be flipped. If there were coordinate-dependence (e.g., if the output coordinates $i$ and $j$ both solely depended on the input coordinate $i$), it may not be possible to find such a single pair of databases across all coordinates, simultaneously, which would decrease the sensitivity and thus the privacy loss. Proposition 3.1 below formally states our result. For more intuition, example functions, as well as the full proof, see Appendix A and Proposition A.2 therein.

**Proposition 3.1.** *For a coordinate-independent multi-label function $f_{multi}(X) : \mathcal{R}^n \to \mathcal{R}^k$, the $\ell_1$ sensitivity $\Delta_1 f_{multi}$ is equal to that of Binary voting applied per label and thus $f_{multi}$ does not provide a better privacy guarantee than $k$ applications of Binary voting.*

Note that Proposition 3.1 is exactly applicable to Binary voting if we assume candidate independence: then each candidate on all ballots only influences a single winner. Analyzing these assumptions, we also readily see two scenarios when a Binary voting mechanism can be outperformed. The first scenario is when any input coordinate $X_i$ explains $> 1$ output (coordinate dependence). Proposition A.4 of Appendix A proves that for $f_{multi}$ to achieve a strictly lesser sensitivity, we necessarily require coordinate dependence. If there is coordinate dependence, then there is some correlation between the outputs of the mechanism (in our case, the labels). When this is the case, it may be possible to achieve a tighter privacy loss bound by leveraging their correlations, especially if these correlations are high (see Section D in the Appendix). The second case we observe is whenever $p > 1$ for the sensitivity, which is a direct result of Hölder's Inequality (Hölder, 1889). We explore this in Section 3.2 by clipping the sensitivity in the $\ell_2$ norm.

## 3.2 DP Mechanisms for $\tau$-Multi-Winner Election

When the average ballot does not vote for all $k$ candidates, it is possible to obtain tighter privacy guarantees by considering the $\tau$-multi-winner election. Here, we bound the $\ell_2$ norm of each ballot, limiting the individual impact of a ballot. For this problem to be meaningful, we assume $\tau < \sqrt{k}$; otherwise, we could always use the Binary voting mechanism from Definition 4. Note that *norm clipping* is a popular method for bounding the influence of an individual's vote on the sensitivity of the mechanism (Abadi et al., 2016b). We integrate this method into Binary voting in Definition 7.

**Definition 7** (DP $\tau$ Voting Mechanism). *Mechanism for $\tau$-Multi-Winner Election. For a $\tau$-multi-winner election the $\tau$ Voting mechanism is*

$$\mathcal{M}_\sigma(\boldsymbol{b}_1, \ldots, \boldsymbol{b}_n) \triangleq \left\{ i : \left( \sum_{j=1}^n \boldsymbol{v}_j[i] + \mathcal{N}(0, \sigma_G^2) \right) > T_i \right\} \in \{0, 1\}^k, \boldsymbol{v}_j[i] \triangleq \min(1, \frac{\tau}{\|\boldsymbol{b}_i\|_2})\boldsymbol{b}_i,$$

where we choose $T_i = n - \sum_{j=1}^n \boldsymbol{v}_j[i]$, which represents the number of *clipped* negative votes. To analyze the privacy loss of this new mechanism, we propose an extension of Lemma 2.1 to make it applicable to this multi-voting task.

**Lemma 3.2.** *RDP-Gaussian mechanism for a multi-label setting. Let $f : \mathcal{X} \to \mathcal{R}^k$ obey $\|f(X) - f(X')\|_2 \leq \Delta_2$ for neighboring datasets $X, X'$. The Gaussian mechanism $\mathcal{M}(X) = f(X) + \mathcal{N}_k(0, \sigma^2 I)$ obeys RDP with $\varepsilon_\mathcal{M}(\lambda) = \frac{\lambda \Delta_2^2}{2\sigma^2}$.*

We present proof in the Appendix C. Next, we illustrate how our mechanism offers superior privacy guarantees to the similar $\ell_1$ clipping proposed in Zhu et al. (2020), which is not optimal for Gaussian mechanisms. Our bound is no larger, and often much *smaller*, than the privacy bound in the $\ell_1$ norm. We can see this by comparing the privacy loss of the two methods in the case of max possible change (as in Definition 6): $\epsilon_\mathcal{M}(\lambda) = \frac{\lambda \tau_2^2}{\sigma^2} = \frac{\lambda \Delta_2^2}{2\sigma^2} \leq \frac{\lambda \Delta_1^2}{2\sigma^2} = \frac{\lambda(2\tau_1)^2}{2\sigma^2} = \frac{\lambda 2\tau_1^2}{\sigma^2}$ where in the first inequality we use the fact that $\|x\|_2 \leq \|x\|_1$. Empirically, we confirm that $\tau$ voting achieves a much tighter bound on many multi-label datasets. Take $\tau_1$ to represent clipping in the $\ell_1$ norm and similarly for $\tau_2$. On Pascal VOC, which has $k = 20$ labels, we find that we can set $\tau_2 = 1.8$ and $\tau_1 = 3.4$ without deteriorating the utility of the mechanism (as measured using the balanced accuracy). In general, we

select $\tau$ to be marginally larger than the average p-norm of labels in the training data. For example, there are on average fewer than 2 positive labels out of 20 per query in the Pascal VOC dataset. Here, $\ell_2$ norm clipping achieves a $>6x$ tighter privacy loss compared to Zhu et al. (2020) and the multi-voting of Section 3.1.

### 3.3 Casting $\infty$-Multi-Winner Elections as Single-Winner Elections

It is also possible to approximate multi-winner elections using single-winner voting mechanisms by encoding each of the $2^k$ different outcomes as a separate candidate for a single-winner voting mechanism. We call this mechanism the powerset voting mechanism.

**Definition 8** (DP Powerset Mechanism for Multi-Winner Elections). *Denote the Powerset operator as* $\mathcal{P}(\cdot)$. *For a $\infty$-multi-winner election with ordered outcomes $\mathcal{P}(\{0,1\}^k)$, where $|\mathcal{P}(\{0,1\}^k)| = 2^k$,*

$$
\mathcal{M}_\sigma(\boldsymbol{b}_1, \ldots, \boldsymbol{b}_n) \triangleq \arg\max_i \left\{ \mathcal{P}(\{0,1\}^k)_i : \sum_{j=1}^n \mathbb{1}_{\boldsymbol{b}_j = \mathcal{P}(\{0,1\}^k)_i} + \mathcal{N}(0, \sigma_G^2) \right\} \in \{0,1\}^k,
$$

where $\mathbb{1}$ represents the indicator function. We first note that this is an *approximation* of the Multi-Winner problem. The outcome of the mechanism may differ from the plurality per candidate if there is no ballot cast with those votes. When we compare our two voting mechanisms, we find that there are two regimes where Powerset voting outperforms Binary voting.

*The first regime is when there are high correlations between the multi-winner candidates*, i.e., there is strong coordinate dependence (see Section B.4). The distribution over outcomes can be approximated by viewing the collection of single-winner outcomes as non-uniform balls and bins problem: each distinct outcome $i \in [1, \ldots, 2^k]$ has some probability $P_i$ of being selected. Then, each of the $n$ ballots is an independent ball being thrown at these bins. We can estimate the probability of each of the $2^k$ single-winner outcomes. Doing so, if the likelihood of a single $i-$th outcome is much higher than the rest, it results in decreased privacy loss.

*The second regime is when neither mechanism can leverage stronger data-dependent privacy guarantees.* When the distribution of multi-winner outcomes and single-winner outcomes approaches a uniform distribution, the best guarantee either can achieve is data-independent privacy bound based on the sensitivity of either mechanism and the amount of noise added. Because Binary voting composes $k$ separate DP queries, the privacy loss using this approach is necessarily higher than from Powerset voting which only releases a single outcome. Instead, when there is coordinate independence, or when each individual multi-winner candidate has a high probability of winning, or when the most likely outcomes are approximately the same between Binary voting and Powerset voting, we find that Binary voting performs better. See Appendix B.4 for our detailed theoretical analysis and supporting empirical evidence. See Sections 4 below for a detailed analysis of the data-independent and data-dependent privacy bounds of Binary voting and Powerset voting.

## 4 From Private Elections to Private Multi-Label Classification

In addition to the obvious application of making elections private, DP voting mechanisms play an important role in privacy-preserving machine learning. Notably, PATE adopted the noisy $\arg\max$ mechanism, a canonical DP single-winner election mechanism, for training private single-label models by transferring knowledge with semi-supervised learning. In this section, we discuss how to use our proposed DP multi-winner mechanisms to build systems for private multi-label classification.

### 4.1 Binary PATE, $\tau$ PATE and Powerset PATE

By replacing the noisy $\arg\max$ mechanism in single-label PATE with one of our multi-winner election mechanisms, we can enable multi-label PATE. In doing so, non-private multi-label models can be trained on the private centralized dataset and used as *teachers* for semi-supervised learning. We leverage a bank of unlabeled data, which is often widely available, and label $m$ of these data points with noisy estimates from the ensemble of *teachers* using our multi-winner mechanisms. We then train a *student* model on the newly DP labeled data, preserving DP of the non-private teachers' training data. We call the corresponding forms Binary PATE, $\tau$ pate, and Powerset PATE.

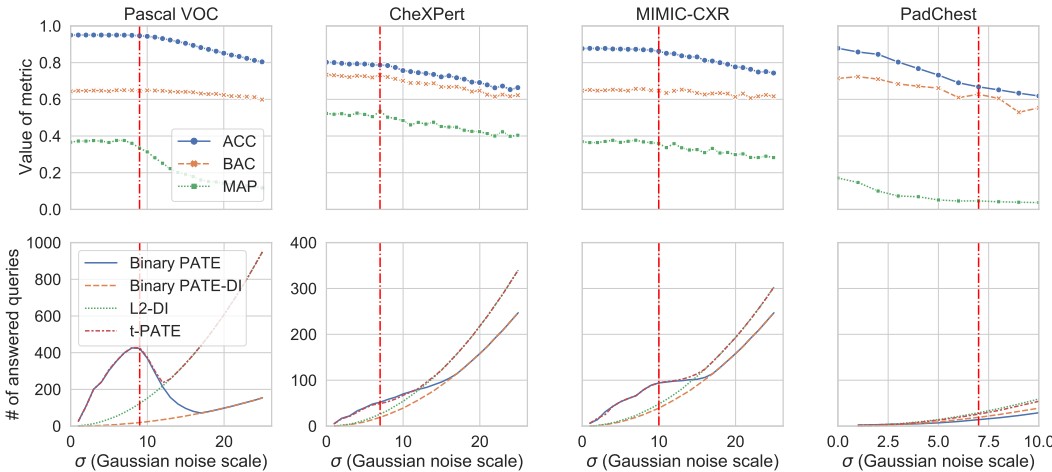

Figure 1: **With sufficient consensus, the best query-utility tradeoff obtained lies in a regime of** $\sigma_G$ **where the data-dependent bound is used.** In the 1st row, we maximize the $\sigma_G$ of $\tau$ PATE while maintaining sufficiently high values of the performance metrics. The chosen values for $(\sigma_G, \tau)$ are $9, 1.8$ for Pascal VOC, $10, 3$ for MIMIC-CXR, and $7, 2.8$ for CheXPert and $7, 2.7$ PadChest. When there is a lack of consensus (on PadChest), we see that the data-independent $\ell_2$ (L2-DI) mechanism bound outperforms all others. For a well chosen $\tau$, there is little-to-no impact on the consensus of the data-dependent regime (c.f. Binary PATE and $\tau$ PATE which leverage the data-dependent bound when it reduces privacy loss). Because of this, $\tau$ PATE achieves a competitive query-utility tradeoff. See Figure 14 in Appendix I.5 for tuning of Binary PATE as an $\infty$-Winner Election.

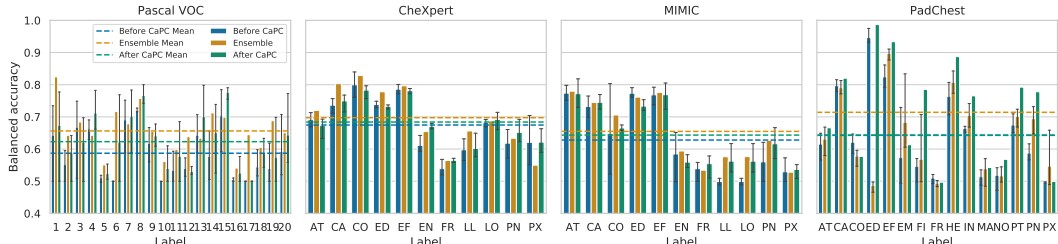

Figure 2: **Using CaPC to improve model performance.** *Dashed lines represent mean balanced accuracy (BAC).* We retrain a given model using additional data labelled by all other models from the same ensemble. We observe a mean increase of BAC by 2.0 percentage points on CheXpert.

Because we only replace aggregation mechanism in PATE, our new multi-label PATE algorithms (and their corresponding voting mechanisms) can directly leverage the tight data-dependent analysis of Papernot et al. (2018). In particular, when there is an empirically observed highly likely outcome, we can often draw a much tighter bound on the privacy loss. Note that the privacy budget must be sanitized before being released which can be done using smooth sensitivity, as shown by Papernot et al. (2018, Algorithm 2). In absence of this empirical observation, when the noisy $\arg\max$ mechanism satisfies $\varepsilon_{\mathcal{M}}(\lambda)$-RDP, our Binary voting and Binary PATE algorithm achieves a data-independent bound of $\sum_i^k \varepsilon_{\mathcal{M}}(\lambda)_i$, $\tau$-voting and $\tau$ PATE of $\frac{\lambda \tau^2}{\sigma^2}$, and Powerset voting and Powerset PATE of $\varepsilon_{\mathcal{M}}(\lambda)$, all per election (query). Note that $\tau$ PATE also leverages data-dependent analysis as with Binary PATE, but with the additional clipping of votes.

## 5 EXPERIMENTAL EVALUATION

We carry out the evaluation on three medical datasets CheXpert (Irvin et al., 2019), MIMIC-CXR (Johnson et al., 2019), PadChest (Bustos et al., 2020), and on a vision dataset Pascal VOC 2012 (Everingham et al., 2010). We use the following metrics: (1) accuracy, (2) mean recall per class denoted as balanced accuracy (BAC) (Brodersen et al., 2010), (3) area under the receiver operating characteristic curve (AUC) (Hand & Till, 2001), and (4) mean average precision (mAP). We use DenseNet-121 (Huang et al., 2017) for the medical datasets and ResNet-50 (He et al., 2016) for the vision dataset. See Supplement Section G for dataset and model descriptions. We experiment with

Table 1: **Model improvements through retraining with multi-label CaPC**

| DATASET | # OF MODELS | STATE | PB ($\varepsilon$) | ACC | BAC | AUC | MAP |
|---|---|---|---|---|---|---|---|
| PASCAL VOC | 1 | INITIAL | - | .97 | .85 | .97 | .85 |
| | 50 | BEFORE CAPC | - | .93±.02 | .59±.01 | .88±.01 | .54±.01 |
| | 50 | AFTER CAPC | 10 | **.94±.01** | **.62±.01** | .88±.01 | .54±.01 |
| | 50 | AFTER CAPC | 20 | **.94±.01** | **.64±.01** | **.89±.01** | **.55±.01** |
| CHEXPERT | 1 | INITIAL | - | .79 | .78 | .86 | .72 |
| | 50 | BEFORE CAPC | - | .77±.06 | .66±.02 | .75±.02 | .58±.02 |
| | 50 | AFTER CAPC | 20 | .76±.07 | **.69±.01** | **.77±.01** | **.59±.01** |
| MIMIC | 1 | INITIAL | - | .90 | .74 | .84 | .51 |
| | 50 | BEFORE CAPC | - | .84±.07 | .63±.03 | .78±.03 | .43±.02 |
| | 50 | AFTER CAPC | 20 | **.85±.05** | **.64±.01** | **.79±.01** | **.45±.03** |
| PADCHEST | 1 | INITIAL | - | .86 | .79 | .90 | .37 |
| | 10 | BEFORE CAPC | - | .90±.01 | .64±.01 | .79±.01 | .16±.01 |
| | 10 | AFTER CAPC | 20 | .88±.01 | .64±.01 | .75±.01 | .14±.01 |

$\varepsilon = 8$ for 5 labels on CheXpert (see Appendix I.3), $\varepsilon = 10$ for predictions on PascalVOC, $\varepsilon = 20$ for 11 labels on CheXpert or MIMIC, and also $\varepsilon = 20$ for 15 labels on CheXpert (see also Section I.9).

## 5.1 ANALYZING THE QUERY-UTILITY TRADEOFF IN PATE

In comparing how the privacy parameter $\sigma_G$ impacts the query-utility tradeoff of each mechanism, under a fixed $\varepsilon = 20$ across all datasets, we find that *under high consensus Binary voting performs best and under lower consensus $\tau$ voting performs best*. In Figure 1, we compare each multi-winner election mechanism used in $\tau$ PATE and find that each mechanism has a range of $\sigma_G$ where it performs best. In particular, as $\sigma_G \to 0$, no queries can be answered by any mechanism due to a lack of noise to satisfy the chosen $\varepsilon$. For sufficiently small $\sigma_G$ and with suitable consensus amongst voters, which is the case for Pascal VOC, CheXPert, and MIMIC-CXR, we find the data-dependent analysis for Binary voting outperforms all others while remaining in a regime of high-performance metrics. As $\sigma_G \to \infty$, the $\tau$ voting mechanism outperforms all others, though in most cases at a decrease in the performance metrics; however, on PadChest, this is the best bound. This can be explained by the fact that on PadChest we can only train 10 teachers before each individual model's accuracy degrades too much. An important distinction is that $\tau$ pate can perform worse than Binary PATE when the chosen $\tau$ bound is too small (because ballot clipping can change the vote distribution). For well-chosen values of $\tau$ we observe only marginal decrease in the number of queries answered and the performance metrics (c.f. Figure 1 with Figures 13, 12, and 14 in Appendix I). Inspecting the tradeoff of Powerset PATE (see Figure 8 in Appendix I.1), we find that a much lower noise can be tolerated before a steep decline in performance metrics. Because of this, we find that much fewer queries can be answered (see Tables 5 and 8 in Appendix I.1). Thus, we recommend $\tau$ PATE as the de-facto mechanism.

## 5.2 PRIVATE CENTRALIZED LEARNING USING MULTI-LABEL PATE

We now show that even in the centralized setting, our multi-label PATE methods outperform the competitive baselines. We train models on Pascal VOC and CheXpert which include 20 and 11 labels, respectively. We leverage the entire training set to train a single non-private model and a single private model via DPSGD (Abadi et al., 2016a). For our multi-label PATE we instead train 50 teachers each on a separate disjoint partition of the centralized training set (and thus, with $1/50$ number of samples compared with DPSGD and the non-private model). We use these teacher models to privately train a student model using semi-supervised learning with MixMatch (Berthelot et al., 2019) (where modifications were made to adapt MixMatch to the multi-label setting). Though the DPSGD algorithm does not assume a public unlabeled dataset, we compare it with our methods because DPSGD can be directly applied to this multi-winner setting. DPSGD for multi-label was studied in prior work (Zhang et al., 2021), and cannot directly leverage public unlabeled data. On the other hand, our PATE-based approaches leverage a public pool of unlabeled samples that have noisy labels provided by the ensemble of teachers. The added noise protects the privacy of the centralized training data. We then train the student on the newly labeled samples. The student model, non-private

baseline, and DPSGD baseline were all pre-trained on ResNet-50 models. Observing Table 2, we see that our Binary PATE algorithm outperforms all other privacy-preserving techniques by a significant margin. Our Binary PATE does not use $\tau$ clipping or the confident GNMax improvement so as to fairly compare with Powerset PATE. The non-private model achieves strong performance across all metrics where the model trained using DPSGD incurs significant degradation across all metrics.

Though Binary PATE outperforms DPSGD and Powerset PATE, it falls short of the non-private model by a wide margin, indicating much room for improvement in multi-label privacy-preserving techniques—in particular, in extreme multi-label settings, which we motivate and expand on in Appendix D. We observe that Powerset PATE answers much fewer queries (leading to less training data for the student model) than Binary PATE, at only 78 compared to 427. Though the student model only trains on 427 samples compared to 5717 for DPSGD, Binary PATE has the benefit of

Table 2: **DPSGD vs PATE**. Comparison between standard non-private model, DPSGD, Powerset and Binary multi-label PATE on the Pascal VOC dataset in terms of utility.

| METHOD | ACC | BAC | AUC | MAP |
|---|---|---|---|---|
| NON-PRIVATE | .97 | .85 | .97 | .85 |
| DPSGD | .92 | .50 | .68 | .40 |
| POWERSET PATE | .94 | .58 | .70 | .29 |
| BINARY PATE | .94 | .62 | .85 | .57 |

using non-private learning on this data whereas DPSGD must add noise in training which impedes model learning. Finally, DPSGD incurs a high computational cost (which multi-label PATE does not) due to the expensive per-example gradient computations (Subramani et al., 2020).

### 5.3 MULTI-LABEL CAPC

By replacing the single-label PATE in Choquette-Choo et al. (2021) with our multi-label PATE mechanisms, we enable multi-label learning in a multi-site setting: the framework of CaPC allows for distributed collaboration across models located at different sites. We scale the evaluation of multi-label CaPC learning to real-world datasets and models by providing a decentralized (independent) evaluation of each answering party, enabling large models. Our multi-label CaPC experiments replicate the setup of Choquette-Choo et al. (2021) using the source code provided.

We train 1 model per participant on separate distinct portions of the training set and then use multi-label CaPC to improve the performance of 3 participants. Details are in Supplement Section H. Observing Table 1, we see that multi-label CaPC consistently improves the BAC, with the greatest improvement of a considerable 5 percentage point increase on Pascal VOC and improvement across all other metrics. These improvements are echoed in the larger and more privacy-sensitive CheXpert and MIMIC-CXR datasets; however, we observe some performance degradation on PadChest, likely because of the degraded vote utility compared to the original training data. As one of the main applications for CaPC is the healthcare domain, the on-average performance improvements and associated privacy guarantees demonstrate the utility of multi-label CaPC in a realistic use case.

Inspecting Figure 2, we see that multi-label CaPC leads to significant improvements in low-sample, low-performance labels (e.g., labels EN and FR on CheXpert). Thus, poorer performing models and classes can gain from the noisy aggregation of more performant teacher models through multi-label CaPC. This has potential ramifications for fairness because our experiments consistently demonstrate that private multi-label CaPC can improve model performance on its poorer performing subpopulations—where Suriyakumar et al. (2021) show that differentially private can hurt performance on these subpopulations. We reiterate that these subpopulations are common in many settings such as healthcare due to imbalanced labels, e.g., rarer diseases (see Figure 6 of Appendix G).

### 6 CONCLUSIONS

We address the need for privacy in the multi-label setting with three new multi-label voting mechanisms. We show and prove that, while simple, our Binary voting cannot be outperformed without strong candidate correlations. When these correlations exist, we prove new data-independent bounds for our $\tau$ voting mechanism and theoretically analyze when Powerset voting performs better. Using these mechanisms, we create multi-label PATE which outperforms DPSGD in the centralized private learning setting. We further enable multi-label learning in multi-site scenarios by creating multi-label CaPC. Our results achieve new state-of-the-art results for private learning in multi-label settings and demonstrate a need for further exploration to lessen the gap between private methods and non-private baselines.

## 7 ETHICS STATEMENT

Our work aims at the protection of privacy and from this perspective, it should benefit society. More specifically, our work aims to provide private multi-label classification in a collaborative setting and therefore any concerns of it being integrated into applications deemed harmful to humans or the environment are a smaller subset of those which exist in regular multi-label classification—a common machine learning application.

It is not well understood what the value of $\varepsilon$ should be and therefore improper usage and large values of epsilons could leak private information. This risk can be mitigated by setting an upper threshold for the possible values of $\varepsilon$ which can be used. The values of $\varepsilon$ used in our experiments (8, 10, and 20) could be considered high. However, since our work provides multi-label classification, there are many labels that need to be released and therefore the privacy budget will naturally be higher than for single-label classification. Our common choice of $\varepsilon = 20$ is empirical. We select a value, which although is not tight, is a reasonable bound for our multi-label setting because it is in the range of values considered by most empirical evaluations (of order 10). We also test other $\varepsilon$ values, for instance, show $\varepsilon = 10$ for the Pascal VOC dataset in Table 1 and $\varepsilon = 8$ for the first 5 labels from the CheXpert dataset in Section I.3 in the Appendix.

The worst scenario in the collaborative setting and for our method is when the querying party colludes with the privacy guardian. In this situation, the labels from each answering party are revealed. There are many possible attacks that can take advantage of such information. One of them is model extraction, where the number of queries answered by an answering party can be sufficient to *replicate* the party's model and steal its intellectual property. We should make sure that the privacy guardian, which is responsible for the privacy protection, is a trusted institution, for example, a party designated by a government.

## 8 REPRODUCIBILITY STATEMENT

For theoretical results, we provide clear explanations of assumptions and complete proofs are included in the Appendix (e.g., C). We submit our code in the supplementary material. In the *README.md* file we provide the main commands needed to run our code. We also describe the pointers to the crucial parts of the code that directly reflect the implementation described in the main part of the submission. For Pascal VOC and medical datasets used in the experiments, we provide complete data processing steps in the code uploaded as supplementary materials.

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

## A  OPTIMALITY OF BINARY PATE

**Intuition behind Proposition 3.1.** Intuitively, a coordinate-independent function will be bounded below by Binary PATE because a single pair of databases $(X, X')$ will lead to the maximum sensitivity of each output coordinate. If the function were coordinate-dependent, this is not guaranteed, i.e., it is possible that some output coordinate will have a smaller sensitivity because we can only pick one pair of databases for all coordinates simultaneously. We motivate this intuition with two examples. Let $X$ be our database with 2 columns, i.e., $X \in \mathcal{R}^2$. We will consider $\Delta_1 f$. Take $f(X) = [sin(X_0), cos(X_1)]$. Because this function is coordinate-independent, we can pick $X = [\frac{-\pi}{2}, 0]$ and $X' = [\frac{\pi}{2}, \pi]$ independently to achieve the worst-case sensitivity of $4$. However, inspection of the coordinate-dependent $g(X) = [sin(X_0), cos(X_0)]$ will show $\Delta_1 g = 2\sqrt{2} < \Delta g_0 + \Delta g_1 = 4$, where no two databases $X$ and $X'$ can ever achieve $\Delta_1 g = 4$ because the minimums and maximums of $sin$ and $cos$ do not occur at the same values of $X$.

**Lemma A.1.** *If $f(X) : \mathcal{R}^n \to \mathcal{R}^k$ is coordinate-independent, there exists a pair of databases $(X, X')$ with $||X - X'||_1 = 1$ (a change in only one row) which achieves the worst-case sensitivity for each of the coordinates $f_i$.*

*Proof.* We consider the $i*$ output coordinate of $f$. We know that $f_{i*}(X) = f_{i*}([X_1, \ldots, X_n]) = f_{i*}([X_{i|i \in P_{i*}}, X_{i|i \notin P_{i*}}]) = f_{i*}([[X_{i|i \in P_{i*}}, X'_{i|i \notin P_{i*}}])$ by Definition **??**. We define the worst-case sensitivity for $f_{i*}$, $\Delta_1 f_{i*}$ as in Dwork et al. (2014). Then

$$
\begin{aligned}
\Delta_1 f_{i*} &= \max_{\substack{(X, X') \\ ||X - X'||_1 = 1}} ||f_{i*}(X) - f_{i*}(X')||_1 \\
&= ||f_{i*}([X_{i|i \in P_{i*}}, X_{i|i \notin P_{i*}}]) - f_{i*}([X'_{i|i \in P_{i*}}, X'_{i|i \notin P_{i*}}])||_1 \\
&= ||f_{i*}([X_{i|i \in P_{i*}}, X_{i|i \notin P_{i*}}]) - f_{i*}([X'_{i|i \in P_{i*}}, X_{i|i \notin P_{i*}}])||_1
\end{aligned}
$$

Therefore we can choose the members in $P_{i*}(X)$ arbitrarily and independently to achieve the worst case sensitivity for $f_{i*}$ while the members not in $P_{i*}(X)$ can take any value because they do not

affect the output. Using the fact that the $P_i$'s are disjoint, we can continue this over all values of $i*$ from 1 to $k$ to get the desired result.

∎

**Proposition A.2.** *For a coordinate-independent multi-label function $f_{multi}(X) : \mathcal{R}^n \to \mathcal{R}^k$ and a single-output function $f_{binary}(X) : \mathcal{R}^n \to \mathcal{R}$ applied once for each of the $k$ labels such that $f_{binary,i} = f_{multi,i}$, the $\ell_1$ sensitivity $\Delta_1 f_{multi}$ is equal to that of binary PATE applied per label and thus does not provide a better privacy guarantee. Note that $f_{binary,i} = f_{multi,i}$ means that they are the same function; we include this to make clear the comparison between the multi-label and binary functions.*

*Proof.* Let $X, X' \in \mathcal{R}^n$ be databases such that $||X - X'||_1 = 1$ and let $f_{binary}(X) : \mathcal{R}^n \to \mathcal{R}, f_{multi}(X) : \mathcal{R}^n \to \mathcal{R}^k$ be arbitrary coordinate-independent functions. Let $f_{.,i}$ and $X_i$ represent the $i$'th coordinate of the output and input respectively. As in Definition **??**, let $P_i(X)$ represent the partition of input coordinates that determine the output coordinates $f_{binary,i}, f_{multi,i}$, such that additionally $\cup_{i=1}^k P_i = X$. We define the sensitivity of $f$, $\Delta f$ as in Dwork et al. (2014). We now consider the $\ell_p$ sensitivity of $f_{multi}$:

$$
\begin{aligned}
\Delta f_{multi} &= \max_{\substack{X,X' \\ ||X-X'||_1=1}} ||f_{multi}(X) - f_{multi}(X')||_p \\
&= \max_{\substack{X,X' \\ ||X-X'||_1=1}} \left((f_{multi,1}(X) - f_{multi,1}(X'))^p + \cdots + (f_{multi,k}(X) - f_{multi,k}(X'))^p\right)^{\frac{1}{p}} \\
&= \max_{\substack{X,X' \\ ||X-X'||_1=1}} \left((f_{multi,1}([X_1,\ldots,X_n]) - f_{multi,1}([X'_1,\ldots,X'_n])^p + \ldots \right. \\
&\quad \left. + (f_{multi,k}([X_1,\ldots,X_n]) - f_{multi,k}([X'_1,\ldots,X'_n]))^p\right)^{\frac{1}{p}} \\
&= \max_{\substack{X,X' \\ ||X-X'||_1=1}} \left(\left(f_{multi,1}([P_1(X), \cup_{i\neq 1}^k P_i(X)]) - f_{multi,1}([P_1(X'), \cup_{i\neq 1}^k P_i(X')])\right)^p + \ldots \right. \\
&\quad \left. + (f_{multi,k}(P_k(X), \cup_{i\neq k} P_i(X)]) - f_{multi,k}([P_k(X'), \cup_{i\neq k} P_i(X')]))^p\right)^{\frac{1}{p}} \\
&= \max_{\substack{X,X' \\ ||X-X'||_1=1}} \left(\left(f_{multi,1}([P_1(X), \cup_{i\neq 1}^k P_i(X)]) - f_{multi,1}([P_1(X'), \cup_{i\neq 1}^k P_i(X)])\right)^p + \ldots \right. \\
&\quad \left. + (f_{multi,k}([P_k(X), \cup_{i\neq k} P_i(X)]) - f_{multi,k}([P_k(X'), \cup_{i\neq k} P_i(X)]))^p\right)^{\frac{1}{p}} \text{ by Definition ??} \\
&= \max_{\substack{X,X' \\ ||X-X'||_1=1}} \left(\left(f_{binary,1}([P_1(X), \cup_{i\neq 1}^k P_i(X)]) - f_{binary,1}(P_1(X'), \cup_{i\neq 1}^k P_i(X)])\right)^p + \ldots \right. \\
&\quad \left. + (f_{binary,k}([P_k(X), \cup_{i\neq k} P_i(X)]) - f_{binary,k}([P_k(X'), \cup_{i\neq k} P_i(X)]))^p\right)^{\frac{1}{p}} \\
&= ((\Delta f_{binary,1})^p + \cdots + (\Delta f_{binary,k})^p)^{\frac{1}{p}} \text{ by Lemma A.1} \\
&= \Delta f_{binary,1} + \cdots + \Delta f_{binary,k} \text{ with } p = 1
\end{aligned}
$$

Where the third last line follows since $f_{multi,i} = f_{binary,i}$.

∎

In the case where $p > 1$, we have that $((\Delta f_{binary,1})^p + \cdots + (\Delta f_{binary,k})^p)^{\frac{1}{p}} \leq \Delta f_{binary,1} + \cdots + \Delta f_{binary,k}$ by a consequence of Hölder's inequality and thus equivalently that $\Delta f_{multi} \leq \Delta f_{binary,1} + \cdots + \Delta f_{binary,k}$

**Lemma A.3.** *If a function $f(X) : \mathcal{R}^n \to \mathcal{R}^k$ is coordinate-dependent, the worst-case sensitivity is not guaranteed.*

*Proof.* We prove by counterexample. Assuming that all coordinate-dependent functions guarantee the worst-case sensitivity, we must find a function that does not satisfy this claim. Take the coordinate-dependent function $f = [sin(X_0), cos(X_0)]$, where $f(X) : \mathcal{R} \to \mathcal{R}^2$ and $X_0$ determines $> 1$ output. The worst-case sensitivity of each coordinate is 2 since $-1 \leq \sin(X_0), \cos(X_0) \leq 1$ and therefore the worst-case sensitivity of a function with two coordinates is $2 + 2 = 4$. However, since $\sin^2 + \cos^2 = 1$, a single choice of $X_0$ and $X_0'$ gives a worst-case sensitivity for $f$ as $2\sqrt{2} < 4$. We arrive at this by taking the derivative of $||[sin(X_0), cos(X_0)] - [sin(X_0'), cos(X_0')]||_1$ and set it to 0 to get for example $X_0 = \frac{\pi}{4}$ and $X_0' = \frac{5\pi}{4}$.

∎

**Proposition A.4.** *For a multi-label function $f_{multi}(X) : \mathcal{R}^n \to \mathcal{R}^k$ to have a lower sensitivity than the sum of the worst-case sensitivities for each coordinate, i.e., $\Delta_1 f_{multi} < \Sigma_{i=1}^k \Delta_1 f_{binary,i}$, $f_{multi}$ must be coordinate-dependent.*

*Proof.* We proceed by contradiction. Let $f_{multi}$ be a coordinate-independent function. By Proposition A.2, $\Delta_1 f_{multi} = \Delta_1 f_{binary,1} + \cdots + \Delta_1 f_{binary,k} = \Sigma_{i=1}^k \Delta_1 f_{binary,i}$ which contradicts $\Delta_1 f_{multi} < \Sigma_{i=1}^k \Delta_1 f_{binary,i}$. Therefore $f_{multi}$ must be coordinate-dependent. ∎

# B    POWERSET VS BINARY MULTI-LABEL PATE

## B.1    PRELIMINARIES

Recall that Binary PATE has $2 \cdot k$ events: the presence or absence independently for each of the $k$ labels. Powerset differs from Binary PATE in that it (1) has a number of events that grows exponentially ($2^k$) in the number of labels, because each subset of the binary vector's powerset is a separate event, (2) returns the entire binary vector of $k$ labels simultaneously, and thus benefits from correlations and deteriorates with the lack thereof, and (3) only adds noise once to the entire binary vector. Here, we analyze under what scenarios Powerset is better or worse than binary PATE.

Recall that there are two bounds for PATE. The data-independent bound is a function only of the noise added to the vector. The data-dependent bound is a function of the probability distribution between events, estimated using the gaps between each predicted event. These gaps are influenced by (1) the number of labels, (2) the amount of noise added, and (3) the amount of correlation between labels, as we will now show.

To analyze PATE, we upper bound the RDP using Theorem B.2. The main (random) variable influencing the this privacy loss is the gap, $q(\bar{n})$ which we calculate using Proposition B.1 from Papernot et al. (2018, Proposition 10). It shows how to select ideal higher order moments $\mu_1, \mu_2$ and $\varepsilon_1, \varepsilon_2$, using the data-dependent value for $q(\bar{n})$ (thus making the RDP bound a nonlinear function of $q(\bar{n})$ solely).

**Proposition B.1.** *For a GNMax aggregator $\mathcal{M}_\sigma$, the teachers' votes histogram $\bar{n} = (n_1, \cdots, n_k)$, and for any $i^* \in \mathcal{P}(\{0,1\}^k)$, where $\mathcal{P}(\cdot)$ denotes the Powerset, we have*

$$\mathbf{Pr}[\mathcal{M}_\sigma(D) \neq i^*] \leq q(\bar{n}),$$

*and*

$$q(\bar{n}) \triangleq \frac{1}{2} \sum_{i \neq i^*} erfc\left(\frac{n_i^* - n_i}{2\sigma}\right)$$

*Where this is a minimal modification of (Papernot et al., 2018, Proposition 7) to go from single-label PATE to multi-label Powerset PATE.*

**Theorem B.2** (From Papernot et al. (2018))**.** *Let $\mathcal{M}$ be a randomized algorithm with $(\mu_1, \varepsilon_1)$-RDP and $(\mu_2, \varepsilon_2)$-RDP guarantees and suppose that there exists a likely outcome $i$ given a dataset $D$ and bound $\tilde{q} \leq 1$ such that $\tilde{q} \geq \Pr[\mathcal{M}(D) \neq i]$. Additionally suppose the $\lambda \leq \mu_1$ and $\tilde{q} \leq e^{(\mu_2-1)\varepsilon_2}\left(\frac{\mu_1}{\mu_1-1} \cdot \frac{\mu_2}{\mu_2-1}\right)^{\mu_2}$. Then for any neighboring dataset $D'$ of $D$, we have:*

$$D_\lambda(\mathcal{M}(D)||\mathcal{M}(D')) \leq \frac{1}{\lambda-1} \log\left((1-\tilde{q}) \cdot \mathbf{A}(\tilde{q}, \mu_2, \varepsilon_2)^{\lambda-1} + \tilde{q} \cdot \mathbf{B}(\tilde{q}, \mu_1, \varepsilon_1)^{\lambda-1}\right)$$

*where $\mathbf{A}(\tilde{q}, \mu_2, \varepsilon_2) \triangleq (1-\tilde{q})/\left(1 - (\tilde{q}e^{\varepsilon_2})^{\frac{\mu_2-1}{\mu_2}}\right)$ and $\mathbf{B}(\tilde{q}, \mu_1, \varepsilon_1) \triangleq e^{\varepsilon_1}/\tilde{q}^{\frac{1}{\mu_1-1}}$.*

When the top 3 vote counts, $n_1 > n_2 > n_3$ satisfy $n_1 - n_2, n_2 - n_3 \gg \sigma$ (the gap is sufficiently large), the RDP bound can be well approximated as

$$\varepsilon_{\mathcal{M}_\sigma}^{powerset}(\lambda) \leq exp(-2\lambda/\sigma^2)/\lambda, where \ \lambda = (n_1 - n_2)/4, \tag{1}$$

which is from Papernot et al. (2018, Corollary 11). Observing Figure 9 of Appendix I, we see that this is indeed the regime we are in for Binary PATE. Assuming Powerset PATE can attain this regime as well, we analyze under what circumstances Powerset PATE outperforms Binary PATE. To do this, we will analyze the respective data-dependent bounds under the expected $gap$.

## B.2 POWERSET

We define the Powerset mechanism in Definition 9. We sort $n(x)_i$ such that $n_i \geq n_j \forall i > j$. Denote $gap \triangleq n_1 - n_2$. We seek to upper bound our best-case expected privacy loss for Powerset PATE. Using Equation 1, we get

$$\mathrm{E}[\varepsilon_{\mathcal{M}_\sigma}^{powerset}(\lambda)] = \int exp(\frac{-gap}{2 \cdot \sigma^2})dgap.$$

$\mathrm{E}[\varepsilon_{\mathcal{M}_\sigma}^{powerset}(\lambda)] \to 0$ when $gap \to \infty$. The base case gap with $O(1)$ probability can be found by viewing Powerset PATE as a non-uniform balls and bins problem. We have $2^k$ subsets, i.e., $|\mathcal{P}(\{0,1\}^k)| = 2^k$. Each subset (instance of an output binary vector) is represented by a bin with probability $P_i$ of having a ball land in it (be voted on by a teacher). The $t$ teachers ($\Sigma_i \bar{n}_i = t$) each vote independently for a bin (output binary vector).

To calculate the $gap$, we will calculate the load of the maximally loaded bin (most voted vector) and assume a best-case scenario for powerset PATE where each other bin has only 1 ball. Denote by $\tau$ the load of the maximally loaded bin. Then, $gap = \tau - 1$ in our best-case scenario. By upper-bounding $\tau$, we get an upper bound for the $gap$.

**Definition 9** (Mechanism for powerset PATE). *Denote the powerset operator as $\mathcal{P}(\cdot)$. For a sample $x$ and $k$ classes, let $f_j(x) \in \{0,1\}^k$ denote the $j-th$ teacher model binary vector prediction. Let $n_i(x)$ be the vote count for the $i-th$ subset (class), i.e., $n_i(x) \triangleq |\{j : f_j(x) = P(\{0,1\}^k)_i\}|$. We define the powerset PATE mechanism as*

$$\mathcal{M}_\sigma(x) \triangleq \arg\max_i \left\{ n_i(x) + \mathcal{N}(0, \sigma^2) \right\}.$$

To upper-bound the max load, we use an indicator random variable $\mathcal{C}_c$ representing the event of $c$ collisions occurring in any bin. Using Markov's inequality followed by Stirlings approximation, we get

$$Pr[\mathcal{C}_c \geq 1] \leq \mathrm{E}[\mathcal{C}_c]$$

$$= \binom{t}{c} \sum_i^{2^k} P_i^c$$

$$\leq (\frac{t \cdot e}{c})^c \sum_i^{2^k} \quad \text{because } (\binom{t}{c} \leq t! \leq (\frac{te}{c})^c).$$

As with $\bar{n}$, we sort our probability $[P_1, \cdots, P_{2^k}]$ such that $P_1 \geq P_2 \cdots \geq P_{2^k}$. Because $n_1 \gg n_2$, we have that $P_1 \gg P_i \forall i \neq 1$. Therefore, we have that $\sum_i^{2^k} \approx P_1^c$ and $Pr[\mathcal{C}_c \geq 1] \leq (\frac{t \cdot e}{c})^c P_1^c$. In particular, we care about the regime where the max load occurs with high-probability ($Pr[\mathcal{C}_c] \to 1$).

$$Pr[\mathcal{C}_c \geq 1] \leq (\frac{t \cdot e \cdot P_1}{c})^c \to 1$$

$$exp(\ln(\frac{t \cdot e \cdot P_1}{c})^c) \to 1$$

$$\implies c \cdot \ln(t \cdot P_1) + c - c \cdot \ln(c) \to 0$$

$$\implies c \cdot \ln(t \cdot P_1) + c = c \cdot \ln(c) \tag{2}$$

Specifying $t$ and $P_1$, we can then directly calculate the max collisions $c$, then the $gap$, and finally the best-case expected privacy loss. To compare directly with Binary PATE, we will directly calculate $P_1$ from the analysis below.

## B.3 BINARY PATE

Here, we aim to specify a probability for success of Binary PATE $p$ (note the lowercase, as opposed to uppercase $P$ for Powerset PATE). This will directly control the expected gap of Binary PATE. Further, from it, we calculate the same gap for powerset PATE to compare the two. Performing a similar analysis for Binary PATE, we can model each teacher's independent label prediction as a Bernoulli trial of probability $p_i$, $i = 1 \cdots k$. With $t$ independent trials, we can model the gap as a binomial distribution of probability $p_i$. To simplify our exposition, and since this will favour the Powerset method over Binary PATE, we assume $p = p_i \forall i$. Using composition, we get an expected privacy loss of

$$
\begin{aligned}
\mathrm{E}[\varepsilon_{binary}] &= k \cdot \sum_{g=0}^{g=t} \left( P(gap = g) \frac{4}{g} exp\left( \frac{-g}{2 \cdot \sigma^2} \right) \right) \\
&= k \cdot \sum_{g=0}^{g=t} \left( \binom{t}{g} p^g (1-p)^{1-g} \frac{4}{g} exp\left( \frac{-g}{2 \cdot \sigma^2} \right) \right) \\
&= k \cdot \sum_{g=0}^{g=t} \left( \binom{t}{g} (pe)^g (1-p)^{1-g} \frac{4}{g e^{\frac{1}{2\sigma^2}}} \right).
\end{aligned}
\tag{3}
$$

## B.4 PRIVACY ANALYSIS COMPARISON: BINARY VERSUS POWERSET PATE

From the assumptions above, we can directly calculate each $P_i$, for Powerset PATE. We do this by recognizing that a single teacher's binary vector can be modeled as a binomial distribution of $k$ (the number of labels) trials and probability $p$. To calculate the probability of any one of the $2^c$ possible binary vectors from the Powerset vector, we express each one as a $l-$hot binary vector ($l$ labels present, independent of their coordinate position). Each possible subset satisfying $l-$hot will have equal probability of occurring (because we fixed $p$ for all labels) Then, the maximum probability binary vector, when $k$ sufficiently large ($k > 5$, approximately) and $p \to 0$ or $p \to 1$ is when $l = 0$ or $l = k$ as otherwise the combinatorics leads to a probability split amongst too many choices. This implication may appear unnatural at first; however, it well models our output label distribution: when there are a set of label coordinates that share similarly high (or low) probabilities of predicting presence (or absence) of that label, then their most likely binary vector is all 1's or 0's.

**Impact of random labels** To model more complex distributions where in general $p_i \neq p$, we can bucket the ranges of $p$ present and analyze these buckets separately. Let us operate under the assumption of coordinate-independence. In this case, we have the maximal variance for the binomial distribution and thus largest gaps for Powerset and Binary PATE. In this case, we expect both methods to perform poorly: however, because the data-independent bound for Powerset PATE is tighter than for Binary PATE, Powerset PATE performs better, as shown in Figure 3. This gives us our first two observations: (1) *Powerset PATE has a better worst-case privacy loss (data-independent bound)* and (2) *both Powerset and Binary PATE degrade to their data-independent bounds as $p \to 0.5$.*

**Impact of Correlation (Coordinate Dependence)** Even when a bucket of labels has $p \approx 1$ or $p \approx 0$ (but not equal), we can still have a varying distribution of outcomes. In particular, with coordinate-independence, we can directly use our binomial analysis above and find that there is a uniform probability for all outcomes in each $l-$hot vector. In particular, there are factorially many outcomes each with equal probability. This drastically degrades the gap by reducing $P_1$. However, when there is coordinate-dependence, e.g., the best-case when each label's value directly implies the rest, then we have only two outcomes for each $l-$hot vector. This improves the gap by reducing the possible subsets and improving $P_1$. We formalize this as follows: if we have some subset of $d$ vectors that are dependent, then we can reduce $k$ in our above analyses to $(k - d)$. This gives us our third

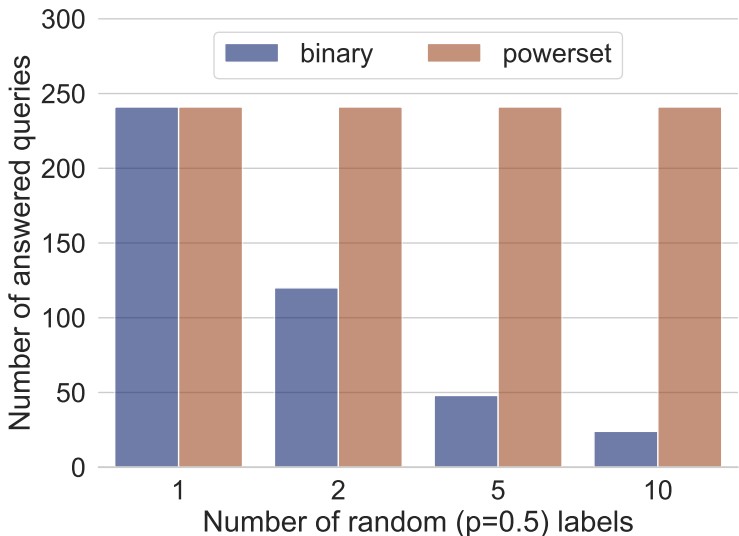

Figure 3: **Powerset PATE outperforms Binary PATE as more labels are generated randomly, i.e., p=0.5.** We use $50$ teachers, privacy noise $\sigma_{GNMax} = 7$, and the privacy budget $\varepsilon$ is set to 20. If all votes are random, then both Binary and Powerset PATE fall back on the data independent bound.

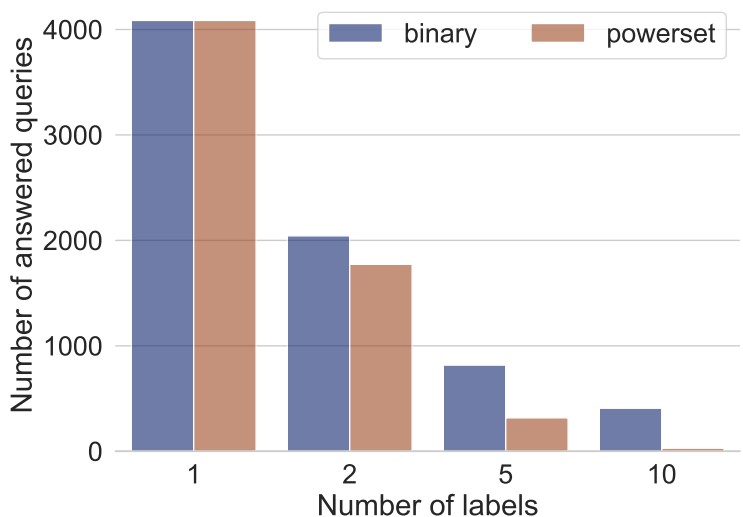

Figure 4: **Binary PATE outperforms Powerset PATE when they both have a similarly high gap.** We use $50$ teachers, privacy noise $\sigma_{GNMax} = 7$, and privacy budget $\varepsilon$ set to 2. Here, all teachers always output $0$ for all labels.

observation: (3) *coordinate dependence has a combinatoric improvement for Powerset PATE and at best a linear improvement for Binary PATE.*

**Impact of Higher Noise Multipliers**    Appealing back to the data-dependent $q(\bar{n})$ calculation of Proposition B.1, we see that Binary PATE has a gap calculated as $2P - t/2\sigma$ for $t$ teachers. This gives $q^{binary(\bar{n})} = erfc(2P - t/2\sigma)$ However, for a similar $gap$, which is bounded for both to a $\max(gap) = t$, we see that Powerset PATE will have $q^{binary(\bar{n})} = \sum_{i}^{2^k} erfc(gap/2\sigma)$, which is always worse when the $gaps$ are equal. We can see this effect in Figure 4. Further, because we union bound across the $2^k$ classes, a higher noise multiplier has a larger impact on Powerset PATE, as shown in Figure 8. These give our fourth and fifth observations: (4) *in the best case, and for any equal gaps, Powerset PATE has a higher privacy loss*, and (5) *higher noise multipliers will impact Powerset PATE more for similar gaps.*

## C  RDP-Gaussian mechanism for a multi-label setting

*Proof.*

$$D_\lambda(\mathcal{M}(X)||\mathcal{M}(X')) = D_\lambda(\mathcal{N}_k(f(X), \sigma^2 I)||\mathcal{N}_k(f(X'), \sigma^2 I)$$

$$= \frac{1}{\lambda-1} \log \left\{ \frac{1}{\sigma^k \sqrt{(2\pi)^k}} \int_{-\infty}^{\infty} \exp\left(\frac{-\lambda}{2\sigma^2} \|\theta - f(X)\|_2^2\right) \exp\left(\frac{-(1-\lambda)}{2\sigma^2} \|\theta - f(X')\|_2^2\right) d\theta \right\}$$

$$= \frac{1}{\lambda-1} \log \left\{ \frac{1}{\sigma^k \sqrt{(2\pi)^k}} \int_{-\infty}^{\infty} \exp\left(\frac{-\lambda \|\theta - f(X)\|_2^2 - (1-\lambda) \|\theta - f(X')\|_2^2}{2\sigma^2}\right) d\theta \right\}$$

$$= \frac{1}{\lambda-1} \log \left\{ \frac{\sigma^k \sqrt{(2\pi)^k}}{\sigma^k \sqrt{(2\pi)^k}} \exp\frac{(\lambda-1)\lambda}{2\sigma^2} \|f(X) - f(X')\|_2^2 \right\} = \frac{\lambda \|f(X) - f(X')\|_2^2}{2\sigma^2} \leq \frac{\lambda\Delta_2^2}{2\sigma^2}$$

$\blacksquare$

## D  Towards Private Extreme Multi-Label Classification

We face extreme multi-label classifications (Shen et al., 2020) in many real-world applications such as semantic segmentation (Zhou et al., 2017), hash-tag suggestions for user images (Denton et al., 2015), product categorisation (Agrawal et al., 2013) and webpage annotation (Partalas et al., 2015) where both input size and label size are extremely large. In this section, we investigate the privacy-accuracy tradeoffs of private semantic segmentation (a common and underlying example in extreme multi-label settings) that links each image pixel to its corresponding object class (an integer value) with a reasonable accuracy of above $60\%$ but an expensive privacy cost of $\varepsilon \approx 3,000$, for a relative "small" image of size $200 \times 200$, or $\approx 40,000$ pixels. We conclude this section by proposing future directions to alleviate the privacy-accuracy tradeoffs in extreme multi-label settings.

We consider MIT ADE20K semantic segmentation dataset (Zhou et al., 2017) that contains 150 objects including 35 stuff objects (e.g. sky, building) and 115 discrete objects (e.g. person, car). The label size for each image pixel is fixed (=150). However, the number of predicted labels for each image is the number of pixels, which varies across the dataset. To perform private semantic segmentation, we use PATE to label each image pixel. We split the training set of MIT ADE20K dataset into equally sized partitions for 20 teachers and train a Pyramid Pooling ResNet50-Dilated architecture of the Cascade Segmentation Module. The test accuracy of the ensemble of teachers (using 2000 of the test images) with respect to the PATE noise standard deviation $\sigma_G$ between 0 and 5 varies from $67\%$ to $47\%$. We observe that the level of noise must be quite small, $\sigma < 3$, or there is a steep drop in accuracy of more than 10 percentage points. The privacy cost is too high to provide meaningful guarantees $\varepsilon$, due to the small $\sigma$ and large number of pixels required to be labeled.

We believe that privacy analysis in extreme multi-label settings can be tightened by exploiting the semantics of inputs. For example in the semantic segmentation task, we can reduce the privacy costs by taking advantage of the dependency between pixels so that instead of releasing an answer per pixel, we can release only a single label per semantic region (a grouping of pixels). Exploring label dependence, rather than assuming label independence, may also enable tighter privacy loss analysis and improve accuracy, as our analysis of Proposition 3.1 suggests. Label dependence is prevalent in many tasks, e.g., in healthcare labels are naturally organised into tree-like hierarchies such that domain experts (e.g. doctors) perform observations and diagnoses conditioned upon their parent node Van Eeden et al. (2012). Figure 5 shows an example of the label structure where the root label node corresponds to the most generic disease of Opacity, while the leaf label node represents the most specific disease of Pneumonia Pham et al. (2021). Pneumonia implies the presence of both Consolidation and

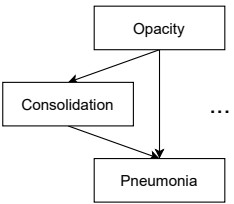

Figure 5: **Example hierarchical structure of labels** in Chest radiography setting.

Opacity diseases. Thus, there exist many possible methods to optimize the answering of queries. It may be possible to tighten the privacy loss due to the implications (or, correlations) between labels; or, to query labels in a specific order such that the all dependent nodes (Consolidation and Opacity) can be inferred by the agreed presence of parent nodes (Pneumonia) by the teacher ensemble.

Table 3: **Dependency Matrix** for the first 5 labels form the CheXpert dataset.

|  | **Atelectasis** | **Cardiomegaly** | **Consolidation** | **Edema** | **Effusion** |
|---|---|---|---|---|---|
| **Atelectasis** |  | *0.975* | *0.987* | *0.976* | *0.983* |
| **Cardiomegaly** | 0.736 |  | 0.836 | 0.784 | 0.869 |
| **Consolidation** | 0.527 | 0.591 |  | **0.631** | 0.790 |
| **Edema** | **0.625** | **0.665** | **0.758** |  | **0.822** |
| **Effusion** | 0.485 | 0.567 | 0.731 | 0.633 |  |

Table 4: **Exploit label dependencies for the multi-label classification**.

|  | # OF QUERIES ANSWERED | ACC | BAC | AUC | MAP |
|---|---|---|---|---|---|
| ANSWER ALL LABELS | *35* | *0.84* | *0.82* | *0.82* | *0.64* |
| INCRESE PRIVACY NOISE | 127 | 0.63 | 0.61 | 0.61 | 0.44 |
| EXPLOIT NEGATIVE DEPENDENCIES | 127 | 0.68 | 0.72 | 0.72 | 0.48 |

In addition to exploiting the knowledge of input and label domains, our analysis of the optimal settings for Binary PATE shows that privacy mechanisms can be tailored to the multi-label classifications. For example, k-fold adaptive bounds Kairouz et al. (2015) that draw tighter ($\ll$ sublinear) privacy bounds for homogenous privacy settings can be extended to heterogeneous $\varepsilon$ per label and per query settings of multi-label classification. However, it is unclear if and under what scenarios we can achieve a tighter bound. For instance, it is possible to take the maximum $\varepsilon$ across all queries, but if there is a large gap $k$-fold adaptive composition may yield looser bounds. These settings of coordinate dependence, high label correlations, and heterogeneous k-fold adaptive composition are interesting for future work.

We design an experiment where the baseline method obtains answers to all the labels while the new proposed method exploits the semantics and queries labels selectively.

We find that the positive dependencies (e.g., if disease A is present then disease B is present as well) constitute a small fraction of the whole dataset. This is because there are many more negative than positive examples in the CheXpert dataset, which is caused by a class imbalance, a common problem in medical datasets. For instance, we find that if both Pneumonia and Pneumothorax are present then Lung Opacity occurs in 83.3% of the cases. However, both Pneumonia and Pneumothorax are present in only 0.06% of samples of the dataset. Thus, we consider negative instead of positive dependencies. For example, if Atelectasis is absent then Consolidation is absent as well in 98.7% of the cases. After ignoring samples for which at least one of Atelectasis or Consolidation have missing values, the percentage of samples where both labels are negative is 83%. We obtain the negative dependencies using the training set and generate the dependency matrix 3.

We compare multi-label PATE executed for each label vs using the semantics and querying the first label (Atelectasis) only, followed by (1) skipping the remaining labels and setting them as negative if the first label is negative, or (2) querying the other labels if the first label is positive. As expected, leveraging the semantics increases the number of answered queries from 35 to 127 for the same privacy budget $\varepsilon = 8$ of at the cost of lower performance (less accurate answers to the queries). However, increasing the number of answered queries by adding more privacy noise ($\sigma = 67.5$) causes the answered queries to be less accurate than by exploiting the label dependencies. We show a detailed comparison in Table 4.

Note that in the above example we consider the first five labels from the CheXpert dataset. We use the same setup as for the comparison between DPSGD and multi-label PATE I.3. The metrics are computed on the same 127 queries (to obtain 127 answered queries for the *Answer all labels* we increase its privacy budget from 8 to 26.5).

## E    DETAILED COMPARISON WITH RELATED WORK

The main contributions from Zhang et al. (2021) on Adaptive DPSGD are two fold: (1) adaptive differentially private deep networks proposed by adding Gaussian noise to gradients whose scale

is linearly decaying instead of being static as in DPSGD, and (2) an analysis of the privacy loss using tCDP (truncated Concentrated Differential Privacy) (Bun et al., 2018), which is a refinement of differential privacy and of concentrated differential privacy, instead of using Moments Accountant (MA) (Abadi et al., 2016a) or Rény Differential Privacy (RDP) (Mironov, 2017). tCDP provides a tighter bound on the privacy leakage compared to MA because it can leverage privacy amplification via sub-sampling. Similarly to RDP, the analysis of Gaussian noise is particularly simple in tCDP and the composition of the randomized mechanism is straightforward. Adaptive DPSGD is applied to CheXpert, CIFAR-10, and MNIST datasets, where data is centralized. Our work is applicable in this centralized setting and the distributed collaborative setting where data is not centralized to a single database. We focus on the collaborative learning where multiple parties (e.g., hospitals) can collaborate via private inference. Another major difference between our approach and Adaptive DPSGD is that we do not reveal any intermediate results because of the use of cryptography, whereas the model trained using DPSGD is also a valuable asset that is released publicly. Finally, we note that Adaptive DPSGD only considers tuning of the model's last layer (the head layer). This task is much easier (it has fewer parameters and is linear) and thus incurs less privacy leakage but requires the assumption of a relevant public dataset to train a performant feature extractor. The comparison between our proposed multi-label PATE and Adaptive DPSGD can be found in Section I.3. The results show that the student models from the proposed multi-label PATE outperform DPSGD.

PATE can be used in different tasks, for example, single-label classification, multi-label classification, generative models based on GANs (discriminate fake from real images, and generate samples privately). The original PATE framework (Papernot et al., 2017; 2018) is only applicable to a single-label classification. We extend PATE to the multi-label classification and show how it can be applied in a collaborative setting. Another line of work applies PATE to the training of GANs. PATE-GAN (Yoon et al., 2019) uses an ensemble of teacher discriminators to create a student discriminator. To ensure differential privacy, the student discriminator is trained solely on inputs produced by the generator and labeled by the teacher discriminators. The final task is a binary classification to discriminate fake from real images, while our task is the multi-label classification. Yang et al. (2021) follows the same algorithm as PATE-GAN except that the generative models used are adversarial autoencoders instead of GANs. This algorithm demonstrates promising results on synthetic speech generation but was not demonstrated on vision tasks. GS-WGAN (Chen et al., 2020) allows releasing a sanitized form of the sensitive data with rigorous privacy guarantees. It reduces gradient sensitivity using the Wasserstein distance and applies gradient sanitization to ensure differential privacy for the generator. However, it considers the task of generating differentially private synthetic data, which is orthogonal to our multi-label classification. G-PATE (Long et al., 2019) is also a method for generating differentially private datasets, which diverges from previous methods such as DP-GAN (Xie et al., 2018) or PATE-GAN by creating a differentially private aggregation mechanism that directly uses information from teacher discriminators to train a student generator. Both, G-PATE and GS-WGAN train the discriminator(s) non-privately while only training the generator with DP guarantees. They differ in the way the gradient is sanitized. GS-WGAN points that G-PATE has two main limitations: (1) gradients have to be discretized using manually selected bins, and (2) high-dimensional gradients in the PATE framework incur a high privacy cost, which requires dimensionality reduction techniques.

Luo et al. (2021) leverage additional, public datasets to instill strong representations in large models, which are then adapted to private datasets at a minimal privacy cost. For the medical domain datasets, we train the models from scratch without using pre-trained models. In contrast to our large-scale medical domain and vision datasets, their method is applied to small-scale datasets, such as CIFAR-10, and for the standard single-label classification only.

Although there exist many differential privacy tree-based ensemble methods, including random forests, their non-private variants fall short of the SoTA performance achieved by deep neural networks on the vision tasks we consider. These methods are typically applied to tabular datasets (Fletcher & Islam, 2017; Bojarski et al., 2014; Xin et al., 2019). To approach the results achieved by deep neural networks, these works require the use of feature engineering (which incurs additional privacy loss) or public labeled data (which is a strong assumption that we do not use). For this reason, it is unlikely that these results will perform comparably to differentially private neural networks, so we limit our evaluation to neural networks. We remark that for the reasons above, DPSGD ensembling of neural networks is also not a viable approach for these tasks.

# F  NORMS FOR CLIPPING

Using an $\ell_\infty$ norm bound does not achieve tighter RDP benefits because we cannot upper-bound the $\ell_2$ norm of the mechanism's outputs (since $|x|_\infty \leq |x|_2$) as required to achieve an RDP benefit. This is analogous to why we achieved tighter RDP privacy loss bounds compared with Private kNN (Zhu et al., 2020): they use the $\ell_1$ norm to bound the $\ell_2$ norm from above, whereas we adapt the analysis to directly operate over the $\ell_2$ norm. Note that we are required to bound the $\ell_2$ norm because we use the Gaussian mechanism (that satisfies RDP for an $\ell_2$ norm bound).

We carried out a theoretical analysis of the bounded noise mechanism from (Dagan & Kur, 2020) for the multi-label classification. The bounded noise in terms of $\ell_\infty$-norm has theoretical properties that could lower the probability of an incorrect prediction for a label. However, the method had a very limited practical application, and for standard multi-label datasets, such as Pascal VOC or CheXpert, the method performed much worse than the binary PATE per label. We removed the bounded noise mechanism from consideration and focused on the binary PATE per label that was performing very well in the empirical evaluation.

# G  DATASETS AND MODEL ARCHITECTURES

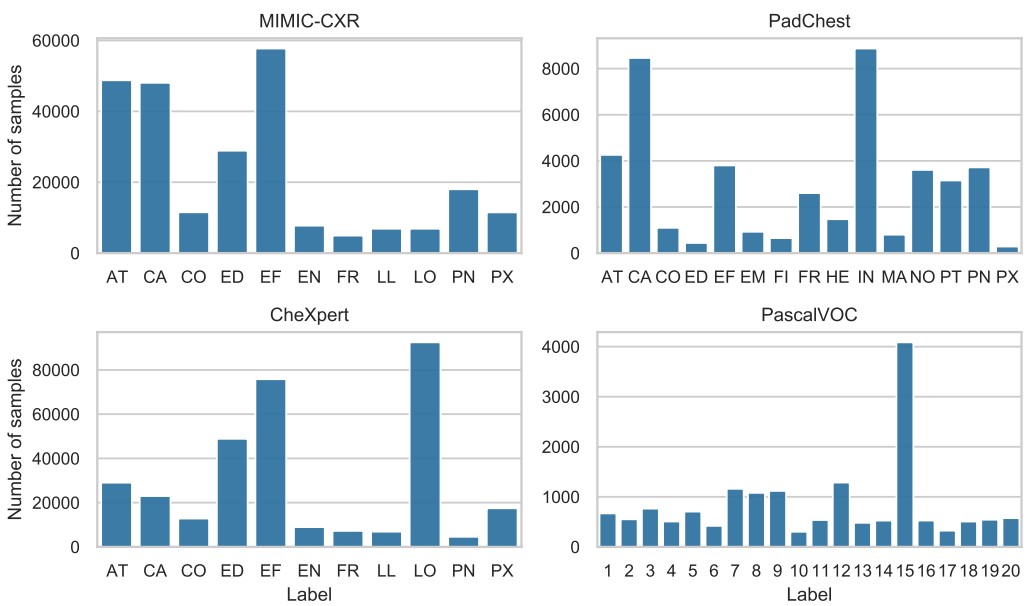

Figure 6: The label distribution for each multi-label dataset (Pascal VOC, CheXpert, MIMIC-CXR, and PadChest).

We experiment on four multi-label datasets. First, we use the common computer vision multi-label dataset, Pascal VOC 2012. Three other of these datasets are privacy sensitive large-scale medical datasets that are commonly used in ML for healthcare: CheXpert (Irvin et al., 2019), MIMIC-CXR (Johnson et al., 2019) and PadChest (Bustos et al., 2020),. These medical datasets present a realistic and large-scale application for multi-label CaPC.

**Pascal VOC 2012** contains $11,540$ images that are split into $5,717$ images for training and $5,823$ images for validation (Everingham et al., 2010). There are 20 classes of object labels (with their index in parentheses) – aeroplane (1), bicycle (2), bird (3), boat (4), bottle (5), bus (6), car (7), cat (8), chair (9), cow (10), dining table (11), dog (12), horse (13), motorbike (14), person (15), potted plant (16), sheep (17), sofa (18), train (19), and tv monitor (20). We use ResNet-50 model (He et al., 2016) that was pre-trained on ImageNet (Deng et al., 2009).

**Medical datasets** contain chest radiographs (X-ray) images. CheXpert (Irvin et al., 2019) has $224,316$ radiographs. MIMIC-CXR-JPG (Johnson et al., 2019) contains $377,110$, and PadChest (Bustos et al., 2020) has $160,868$. The goal in each dataset is to predict presence of pathologies. However, there are differences between pathology labels across these three datasets. X-ray images of CheXpert are annotated with 11 pathologies– Atelectasis, Cardiomegaly, Consolidation, Edema, Effusion, Enlarged Cardiomediastinum, Fracture, Lung Lesion, Lung Opacity, Pneumonia and Pneumothorax. MIMIC-CXR-JPG includes 11 pathologies– Enlarged Cardiomediastinum, Cardiomegaly, Lung Opacity, Lung Lesion, Edema, Consolidation, Pneumonia, Atelectasis, Pneumothorax, Pleural Effusion, Pleural Other, Fracture, Support Devices. PadChest includes 15 pathologies– Atelectasis, Cardiomegaly, Consolidation, Edema, Effusion, Emphysema, Fibrosis, Fracture, Hernia, Infiltration, Mass, Nodule, Pleural_Thickening, Pneumonia and Pneumothorax.

The pathology and its code (in parenthesis) is as follows: Atelectasis (AT), Cardiomegaly (CA), Consolidation (CO), Edema (ED), Effusion (EF), Emphysema (EM), Enlarged Cardiomediastinum (EN), Fibrosis (FI), Fracture (FR), Hernia (HE), Infiltration (IN), Lung Lesion (LL), Lung Opacity (LO), Mass (MA), Nodule (MO), Pleural_Thickening (PT), Pneumonia (PN), Pneumothorax (PX).

All datasets obtain labels from associated reports. Both CheXpert and MIMIC-CXR-JPG use the CheXpert labelling system, which is a rule based approach. PadChest obtains reports annotated by trained radiologists, then trains an attention-based recurrent neural network to predict on these annotations, and labels the remaining data. On all three datasets, we train a DenseNet-121 (Huang et al., 2017) model and filter only frontal images (AP and PA). The main difference between our setup and the one from Cohen et al. (2020) is that we use both frontal views *AP* and *PA*, while the cited work uses only one of the frontal views, thus either *AP* and *PA*.

The label distribution for each multi-label dataset (Pascal VOC, CheXpert, MIMIC-CXR, and PadChest) is presented in Figure 6. The label distribution for medical datasets is more unbalanced than for Pascal VOC, due to prevalence or rarity of certain diseases.

## H  EXPERIMENTAL DETAILS

Our experiments were performed on machines with Intel®Xeon®Silver 4210 processor, 128 GB of RAM, and four NVIDIA GeForce RTX 2080 graphics cards, running Ubuntu 18.04.

We train ResNet-50 (He et al., 2016) on the Pascal VOC 2012 (Everingham et al., 2010) dataset by minimizing multi-label soft margin loss function in 1000 epochs. As an optimiser, we use SGD with learning rate and weight decay of $0.001$ and $1e-4$, respectively. In case of 50 Pascal VOC models, we split the $5,717$ images of the training set to 50 private training sets in order to train 50 teacher models.

We train the medical datasets CheXpert, MIMIC-CXR, and PadChest using the standard DenseNet-121 architecture trained for 100 epochs, with weight decay of $1e-5$, Adam optimizer (learning rate=0.001) with the adam_amsgrad paremeter set. We reduce the learning rate when it plateaus, batch size is 64. The loss type is BCE (Binary Cross Entropy) with logits, and probability threshold is adjusted per label (where the standard value is $0.5$). For 50 CheXpert models, each model is trained on 3750 private train samples. PadChest uses 9223 private samples per each of 10 teacher models.

Regarding the metrics, balanced accuracy (BAC), area under the curve (AUC), and mean average precision are common metrics used to asses performance of the multi-label classification (Ben-Baruch et al., 2021). Note that because we are in a sparse-hot multi-label setting (where a sample has many 0's - negative labels) the accuracy is not a preferred metric because a model can achieve high accuracy by returning all 0's; we include the accuracy metric for completeness.

**For our CaPC experiments**, we sample uniformly, without replacement, data points from the respective training data distribution to create disjoint partitions $D_i$ of equal size for each party $i$. We use 50 total parties for Pascal VOC, CheXpert, and MIMIC-CXR, as well as 10 for PadChest; this choice depended on the sizes of the datasets and the performance of the models on the resulting partitioned data (here, we ensured no single model dropped below $60\%$ BAC). We use $Q=3$ querying parties and sample (without replacement) at least 1000 data points from the test distribution to form the unlabeled set for each querying party. We leave at least 1000 held-out data points for evaluating models. We first train party $i$'s model on their private data $D_i$, then simulate multi-label

CaPC learning by having each querying party complete the protocol with all other parties as answering parties. Using the new labelled data provided by multi-label CaPC, we retrain each querying party's model on their original data $D_i$ plus the new labelled data. We average metrics over at least 3 runs with each using a different random seed.

## I ADDITIONAL EXPERIMENTS

### I.1 MULTI-LABEL PATE: BINARY VS POWERSET

We compare Binary vs Powerset multi-label PATE in Tables 5, 8, and Figure 7. For each data point, we check how the performance of the Binary and Powerset PATE degrades as we increase the scale of Gaussian noise $\sigma_{GNMax}$, and select the scale that preserves high values of the metrics (ACC, BAC, AUC, MAP), with maximum performance drop by a few percentage points - we select this threshold arbitrarily. We align both Binary and Powerset methods according to the performance metrics. For a given values of the metrics, we find the $\sigma_{GNMax}$ starting from 1 to the highest possible value that gives performance above the chosen threshold and allows us to answer maximum number of queries. For example, in Figure 8, we select $\sigma_{GNMax} = 4$ for Powerset PATE and $\sigma_{GNMax} = 11$ for Binary PATE, while the values of the metrics are ACC=.95, AUC=.66, and MAP = .37. For CheXpert, we set $\sigma_{GNMax} = 5$ for Powerset PATE and $\sigma_{GNMax} = 20$ for Binary PATE, while the values of the metrics are ACC=.71, AUC=.68, and MAP = .45.

Table 5: **CheXpert: Performance of Binary PATE vs Powerset PATE** w.r.t. number of answered queries, ACC, BAC, AUC, and mAP (as measured on the test set with the specified $\sigma_{GNMax}$. PB ($\varepsilon$) is the privacy budget. We use 50 teacher models. When we limit number of labels per dataset, we select the first $k$ labels.

| DATASET | MODE | # OF LABELS | QUERIES ANSWERED | PB ($\varepsilon$) | $\sigma_{GNMAX}$ | ACC | BAC | AUC | MAP |
|---|---|---|---|---|---|---|---|---|---|
| CHEXPERT | BINARY | 1 | **988** | 20 | 8 | .71 | .71 | .71 | .63 |
| CHEXPERT | POWERSET | 1 | **988** | 20 | 8 | .71 | .71 | .71 | .63 |
| CHEXPERT | BINARY | 2 | **898** | 20 | 18 | .73 | .72 | .72 | .62 |
| CHEXPERT | POWERSET | 2 | **872** | 20 | 12 | .72 | .72 | .72 | .60 |
| CHEXPERT | BINARY | 3 | **399** | 20 | 13 | .75 | .77 | .77 | .55 |
| CHEXPERT | POWERSET | 3 | **674** | 20 | 10 | .74 | .76 | .76 | .55 |
| CHEXPERT | BINARY | 4 | **554** | 20 | 17 | .74 | .75 | .75 | .60 |
| CHEXPERT | POWERSET | 4 | **323** | 20 | 5 | .76 | .78 | .78 | .60 |
| CHEXPERT | BINARY | 5 | **320** | 20 | 17 | .74 | .76 | .76 | .59 |
| CHEXPERT | BINARY | 5 | **932** | 20 | 29 | .70 | .70 | .70 | .54 |
| CHEXPERT | POWERSET | 5 | **582** | 20 | 10 | .74 | .76 | .76 | .54 |
| CHEXPERT | BINARY | 6 | **299** | 20 | 18 | .74 | .75 | .75 | .56 |
| CHEXPERT | POWERSET | 6 | **392** | 20 | 7 | .74 | .75 | .75 | .55 |
| CHEXPERT | BINARY | 7 | **157** | 20 | 12 | .73 | .74 | .74 | .51 |
| CHEXPERT | POWERSET | 7 | **338** | 20 | 7 | .73 | .72 | .72 | .51 |
| CHEXPERT | BINARY | 8 | **145** | 20 | 13 | .73 | .73 | .73 | .51 |
| CHEXPERT | POWERSET | 8 | **200** | 20 | 5 | .73 | .71 | .71 | .50 |
| CHEXPERT | BINARY | 9 | **159** | 20 | 16 | .71 | .70 | .70 | .50 |
| CHEXPERT | POWERSET | 9 | **241** | 20 | 6 | .71 | .70 | .70 | .50 |
| CHEXPERT | BINARY | 10 | **105** | 20 | 11 | .74 | .72 | .72 | .50 |
| CHEXPERT | POWERSET | 10 | **146** | 20 | 4 | .73 | .71 | .71 | .50 |
| CHEXPERT | BINARY | 11 | **201** | 20 | 5 | .71 | .68 | .68 | .45 |
| CHEXPERT | POWERSET | 11 | **152** | 20 | 20 | .71 | .68 | .68 | .44 |

### I.2 POWERSET WITH $\tau$-VOTING

We present how to incorporate $\tau$-voting into the Powerset method. A given teacher is allowed to set up to $\tau$ positive labels for a given query sample. This approach limits the number of positive classes from $C = 2^k$ to $C = \binom{k}{0} + \binom{k}{1} + \cdots + \binom{k}{\tau}$ classes. Since we have fewer classes: (1) it increases the possibility of achieving a consensus between teachers (a higher gap between max and runner-up numbers of votes can be achieved), (2) the data independent bound is lower (computed as:

Table 6: **Pascal VOC: Performance of Binary PATE vs Powerset PATE** w.r.t. number of answered queries, ACC, BAC, AUC, and mAP as measured on the test set with the specified $\sigma_{\text{GNMax}}$. PB ($\varepsilon$) is the privacy budget. We use 50 teacher models. When we limit number of labels per dataset, we select the first $k$ labels.

| Dataset | Mode | # of Labels | Queries Answered | PB ($\varepsilon$) | $\sigma_{\text{GNMax}}$ | ACC | BAC | AUC | mAP |
|---|---|---|---|---|---|---|---|---|---|
| Pascal VOC | Binary | 1 | **5464** | 20 | 2 | .98 | .84 | .84 | .75 |
| Pascal VOC | Powerset | 1 | **5464** | 20 | 2 | .98 | .84 | .84 | .75 |
| Pascal VOC | Binary | 2 | **5442** | 20 | 5 | .97 | .74 | .74 | .54 |
| Pascal VOC | Powerset | 2 | **5464** | 20 | 5 | .97 | .74 | .74 | .55 |
| Pascal VOC | Binary | 3 | **3437** | 20 | 7 | .97 | .72 | .72 | .51 |
| Pascal VOC | Powerset | 3 | **3416** | 20 | 7 | .97 | .72 | .72 | .51 |
| Pascal VOC | Binary | 5 | **2398** | 20 | 8 | .96 | .66 | .66 | .39 |
| Pascal VOC | Powerset | 5 | **2040** | 20 | 8 | .96 | .65 | .65 | .33 |
| Pascal VOC | Binary | 8 | **1543** | 20 | 7 | .96 | .69 | .69 | .46 |
| Pascal VOC | Powerset | 8 | **1192** | 20 | 7 | .95 | .68 | .68 | .37 |
| Pascal VOC | Binary | 11 | **1190** | 20 | 11 | .95 | .66 | .66 | .37 |
| Pascal VOC | Powerset | 11 | **702** | 20 | 4 | .95 | .66 | .66 | .37 |
| Pascal VOC | Binary | 14 | **888** | 20 | 12 | .95 | .65 | .65 | .36 |
| Pascal VOC | Powerset | 14 | **417** | 20 | 3 | .95 | .65 | .65 | .37 |
| Pascal VOC | Binary | 15 | **465** | 20 | 8 | .94 | .66 | .66 | .36 |
| Pascal VOC | Powerset | 15 | **165** | 20 | 3 | .94 | .66 | .66 | .36 |
| Pascal VOC | Binary | 16 | **461** | 20 | 7 | .95 | .65 | .65 | .38 |
| Pascal VOC | Powerset | 16 | **94** | 20 | 2 | .95 | .65 | .65 | .38 |
| Pascal VOC | Binary | 18 | **446** | 20 | 2 | .95 | .65 | .65 | .36 |
| Pascal VOC | Powerset | 18 | **94** | 20 | 2 | .95 | .65 | .65 | .36 |
| Pascal VOC | Binary | 20 | **427** | 20 | 7 | .95 | .65 | .65 | .37 |
| Pascal VOC | Powerset | 20 | **78** | 20 | 2 | .95 | .65 | .65 | .33 |

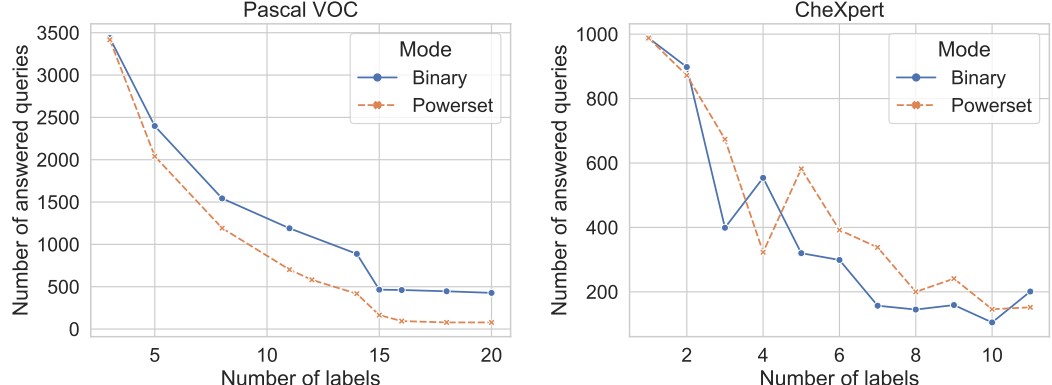

Figure 7: **Binary vs Powerset PATE: number of answered queries**. We compare the number of answered queries vs number of $k$ first labels selected from the Pascal VOC and CheXpert datasets. We keep the privacy budget $\varepsilon = 20$. The private multi-label classification with Binary PATE performs better (answers more queries) than Powerset PATE for Pascal VOC. The CheXpert dataset is more noisy (in terms of labeling, the pathologies are sparse and difficult to detect) and does not give us a clear preference of the multi-label PATE method.

$1 - (1/C)$), and (3) the method has higher performance (faster run-time). On the other hand, it does not lower the sensitivity of the Powerset mechanism because changing one of the teachers potentially decreases the vote count from one class and increase it in another one. Thus, the sensitivity remains 2.

Intuitively, up to $\tau$ positive labels could be selected based on the confidence of the positive value for a label, where we would choose the top $\tau$ labels with the highest confidence of being positive. However, given that most deep multi-label models use independent predictive heads (i.e., a separate sigmoid

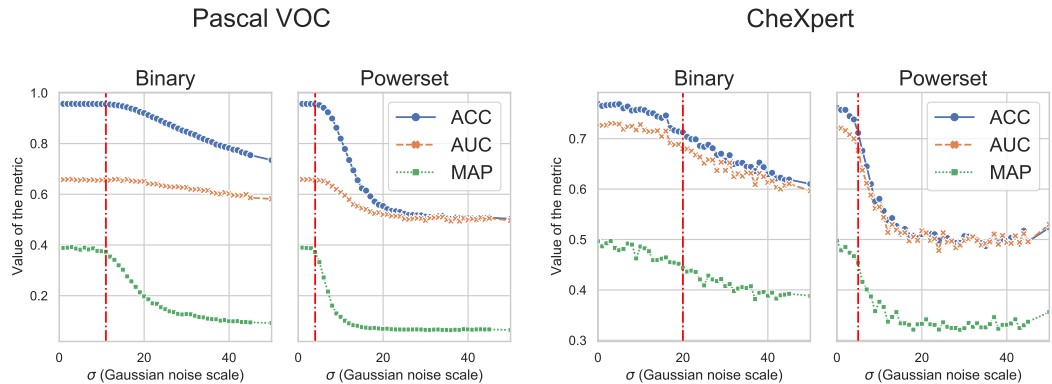

Figure 8: **Binary vs Powerset PATE: performance**. We compare the drop in performance in terms of accuracy (ACC), area under the curve (AUC), and mean average precision (MAP) between Binary PATE and Powerset PATE as we increase the scale of the Gaussian noise $\sigma_{GNMax}$. We select the first 11 labels in Pascal VOC and all 11 labels from CheXpert. We keep the privacy budget $\varepsilon = 20$. The private multi-label classification with PATE performs better (preserves higher values of the metrics) using the Binary approach.

activation per output label), naively comparing these values may not lead to the best performance. Thus, no clear notion of what the desired subset is in the case that $> \tau$ candidates are present.

For the Pascal VOC dataset, the same threshold of $0.5$ probability is used per label, so the prediction heads for each label are aligned. For the CheXpert dataset, different probability thresholds are set per label, so the problem of deciding which of the labels should remain positive (if there are more positive labels than the max $\tau$ of positive labels) is difficult to resolve. This would likely require some form of domain knowledge.

For the original train set of the Pascal VOC dataset (with 5823 samples), the average number of positive labels per example is 1.52, with 20 total labels. More detailed statistics on the train set are presented in Table 7.

Table 7: More statistics about the positive labels per data point in the Pascal VOC dataset.

| NUMBER OF POSITIVE LABELS | NUMBER OF TRAIN SAMPLES |
|---|---|
| 6 | 1 |
| 5 | 17 |
| 4 | 105 |
| 3 | 484 |
| 2 | 1677 |
| 1 | 3539 |
| 0 | 0 |

When the $\tau$-voting is applied with the fixed privacy budget, we are able to answer more queries, while the performance metrics (e.g., accuracy) remain comparable. When the initial number of labels is set to $k = 5$, for $\tau = 2$ the number of classes is 16 instead of 32 ($\tau = 5$, which is equivalent to no clipping) and we are able to answer 5% queries more, for $\tau = 1$, we have only 6 classes and almost 14% more answered queries. For $\tau = 5$ there is only 1 more class than for $\tau = 4$ and we observe a very small difference in the number of answered queries (namely 8). The number of answered queries using the Binary method is still higher than with Powerset even when $\tau = 1$ (the number of classes is $C = k + 1$).

Table 8: **Pascal VOC: Performance of Powerset PATE with $\tau$-clipping** w.r.t. number of answered queries with the specified $\sigma_{\text{GNMax}}$. The ACC, BAC, AUC, and mAP are measured on the answered queries from the test set. PB $(\varepsilon)$ is the privacy budget. We use 50 teacher models. When we limit the number of labels per dataset, we select the first $k$ labels. The $\tau$ denotes a maximum number of positive labels that a teacher is allowed to return per data sample.

| # OF LABELS | $\tau$ | QUERIES ANSWERED | PB $(\varepsilon)$ | $\sigma_{\text{GNMAX}}$ | ACC | BAC | AUC | MAP |
|---|---|---|---|---|---|---|---|---|
| 1 | 1 | **5464** | 20 | 2 | .98 | .84 | .84 | .75 |
| 2 | 1 | **5464** | 20 | 5 | .97 | .75 | .75 | .55 |
| 2 | 2 | **5464** | 20 | 5 | .97 | .74 | .74 | .55 |
| 3 | 1 | **3437** | 20 | 7 | .97 | .71 | .71 | .48 |
| 3 | 2 | **3421** | 20 | 7 | .97 | .71 | .71 | .49 |
| 3 | 3 | **3416** | 20 | 7 | .97 | .72 | .72 | .51 |
| 4 | 1 | **2547** | 20 | 7 | .97 | .68 | .68 | .43 |
| 4 | 2 | **2527** | 20 | 7 | .97 | .69 | .69 | .43 |
| 4 | 3 | **2485** | 20 | 7 | .97 | .69 | .69 | .43 |
| 4 | 4 | **2485** | 20 | 7 | .97 | .69 | .69 | .43 |
| 5 | 1 | **2321** | 20 | 8 | .96 | .65 | .65 | .36 |
| 5 | 2 | **2151** | 20 | 8 | .96 | .65 | .65 | .34 |
| 5 | 3 | **2077** | 20 | 8 | .96 | .65 | .65 | .34 |
| 5 | 4 | **2048** | 20 | 8 | .96 | .65 | .65 | .33 |
| 5 | 5 | **2040** | 20 | 8 | .96 | .65 | .65 | .33 |
| 8 | 8 | **1192** | 20 | 7 | .95 | .68 | .68 | .37 |
| 10 | 1 | **947** | 20 | 5 | .97 | .65 | .65 | .36 |
| 10 | 2 | **896** | 20 | 5 | .96 | .65 | .65 | .36 |
| 10 | 3 | **895** | 20 | 5 | .96 | .65 | .65 | .33 |
| 10 | 4 | **885** | 20 | 5 | .96 | .65 | .65 | .33 |
| 10 | 5 | **877** | 20 | 5 | .96 | .65 | .65 | .33 |
| 10 | 6 | **861** | 20 | 5 | .96 | .64 | .64 | .30 |
| 10 | 7 | **849** | 20 | 5 | .96 | .64 | .64 | .29 |
| 10 | 8 | **848** | 20 | 5 | .96 | .66 | .66 | .34 |
| 10 | 9 | **848** | 20 | 5 | .96 | .64 | .64 | .32 |
| 10 | 10 | **848** | 20 | 5 | .96 | .64 | .64 | .30 |
| 11 | 11 | **702** | 20 | 4 | .95 | .66 | .66 | .37 |
| 14 | 14 | **417** | 20 | 3 | .95 | .65 | .65 | .37 |
| 15 | 15 | **165** | 20 | 3 | .94 | .66 | .66 | .36 |
| 16 | 16 | **94** | 20 | 2 | .95 | .65 | .65 | .38 |
| 18 | 18 | **94** | 20 | 2 | .95 | .65 | .65 | .36 |
| 20 | 1 | **111** | 20 | 2 | .95 | .63 | .63 | .33 |
| 20 | 2 | **99** | 20 | 2 | .96 | .67 | .67 | .41 |
| 20 | 3 | **81** | 20 | 2 | .96 | .61 | .61 | .32 |
| 20 | 4 | **78** | 20 | 2 | .96 | .63 | .63 | .33 |
| 20 | 5 | **78** | 20 | 2 | .96 | .63 | .63 | .34 |
| 20 | 10 | **78** | 20 | 2 | .95 | .65 | .65 | .34 |
| 20 | 20 | **78** | 20 | 2 | .95 | .65 | .65 | .33 |

## I.3 DPSGD VS PATE ON CHEXPERT

There are a few key differences between PATE and DPSGD in a direct comparison. DPSGD requires centralization of data whereas PATE does not, while only additionally assuming some pool of public unlabeled samples to be used for labeling by teachers and for the semi-supervised learning (e.g., with MixMatch). The centralization of data is a strong requirement for DPSGD which is often not applicable to the tasks we consider (e.g., medical or financial data) because these institutions may be blocked from centralizing data due to regulations.

The Adaptive DPSGD method from Zhang et al. (2021) does not publish the code, thus the following is our best effort. The results presented below come from Figure 5 in Zhang et al. (2021). Similar to their experimental setup, we use DenseNet-121 pre-trained on ImageNet and fine-tune only the last fully connected layer while keeping all the other (convolutional) layers fixed. Across all of the experiments, we use $\varepsilon = 8$ as the privacy budget. We use the whole CheXpert test set, and here report results for $\delta = 10^{-4}$. To evaluate the performance of the models, we use the CheXpert test set (from

the valid.csv file in CheXpert-v1.0-small). As shown in the Table 9, in this setting, we outperform the Adaptive DPSGD.

We use the following parameters to obtain our results for the Binary multi-label PATE:

```
sigma gnmax = 7.0
sigma threshold = 0
threshold = 0
method = multilabel
batch size = 20
learning rate = 0.001
epochs = 100
weight decay = 0
X-ray views = ['AP', 'PA']
Multilabel_prob_threshold = [0.53, 0.5, 0.18, 0.56, 0.56]
```

Table 9: **DPSGD vs PATE on Chexpert** for the first 5 labels.

| METHOD | AT | CA | CO | ED | EF | AVERAGE |
|---|---|---|---|---|---|---|
| NON-PRIVATE | *0.84* | *0.80* | *0.87* | *0.90* | *0.91* | *0.87* |
| DPSGD | 0.56 | 0.53 | 0.66 | 0.56 | 0.62 | 0.58 |
| ADAPTIVE DPSGD | 0.75 | 0.73 | **0.84** | **0.79** | 0.79 | 0.78 |
| BINARY PATE | **0.78** | **0.75** | **0.84** | 0.76 | **0.81** | **0.79** |

### I.3.1 CDF OF GAPS

We show the Cumulative Distribution Function (CDF) for the gaps in Figure 9. We observe that the ensemble of teachers is confident about the answers for most labels for the Binary PATE and there are very small gaps for most queries when using Powerset PATE, which shows much lower confidence of teachers in choosing the same (super) classes, which are created from aggregated label predictions (the values of the labels are collected in a binary vector that constitutes a class).

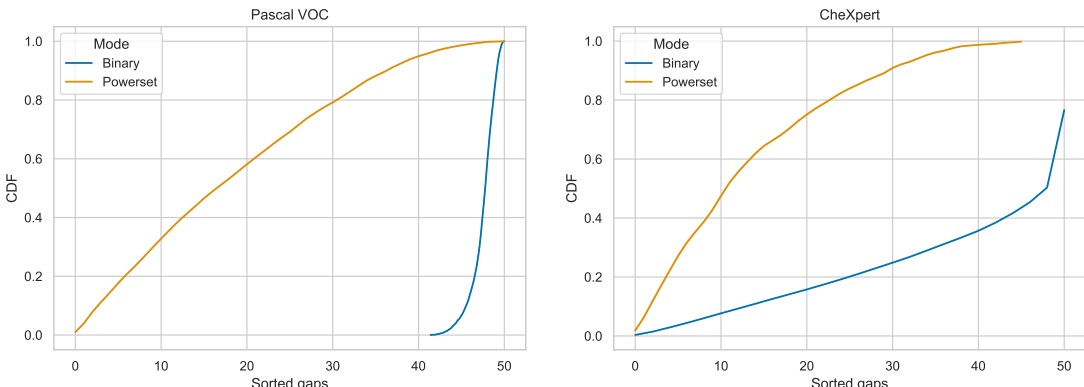

Figure 9: **Binary vs Powerset PATE: CDF of gaps** (differences between vote counts). We use 50 teacher models trained on the Pascal VOC and CheXpert datasets. These are raw gaps without adding any noise to the vote histograms. Most of the gaps are relatively large ($> 40$) for the Binary PATE per label, which shows that teachers are confident about the answers to queries. The average gap for the Powerset method is relatively low (only 18 for Pascal VOC and 13 for CheXpert).

### I.3.2 GAPS AND PERFORMANCE METRICS

With more labels for the Powerset method, we have more possible classes and the votes become more spread out. This can be measured by the gap, which is the difference between the maximum number

of votes per class and the runner-up (the number of votes for the next class with the highest number of votes). In Figures 10 and 11, we plot the average gap (on the y-axis) for the test set across all histograms for a given number of labels (presented on the x-axis). We compare the Powerset PATE (denoted as Powerset) vs the Binary PATE per label (denoted as Binary). The average gap between votes is comparable for different number of labels of Binary PATE since this method considers each label separately and there are always only two classes. The average gap between votes decreases very fast for Powerset because of the exponential growth of number of classes with more labels considered. Additionally, the performance metrics: acc (accuracy), bac (balanced accuracy), aread under the curve (auc), and mean average precision (map), are also higher for the Binary than Powerset method in case of CheXpert dataset and comparable for Pascal VOC.

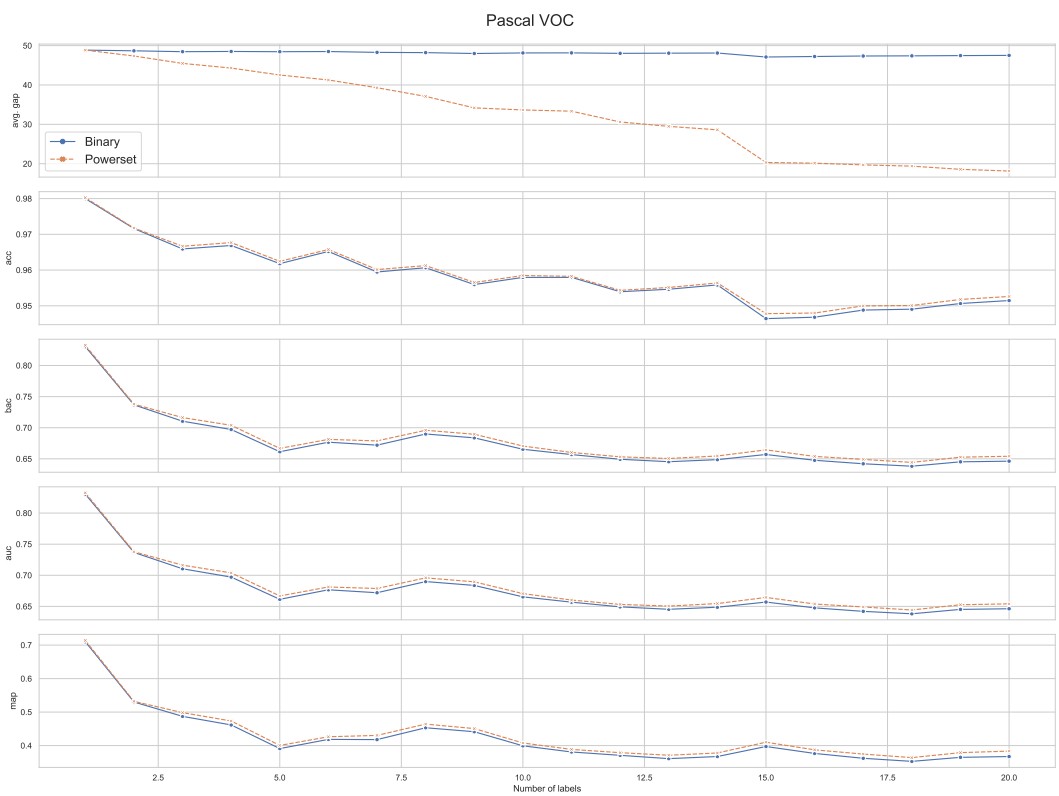

Figure 10: **Binary vs Powerset PATE: gaps and metrics for Pascal VOC**. We use 50 teacher models. These are raw gaps without adding any noise to the vote histograms.

## I.4 GENERAL TUNING OF HYPER-PARAMETERS

We set the hyper-parameters by considering the utility and privacy of our proposed method. Based on Lemma 3.2, we minimize the privacy budget by maximizing the $\sigma$ parameter of the Gaussian noise and minimizing the $\tau$ parameters. In Figure 1, we tune the value of the $\sigma$ parameter of the Gaussian noise based on the utility metrics: accuracy (ACC), balanced accuracy (BAC), mean average precision (MAP). We set $\sigma$ as 9, 10, and 7 for Pascal VOC, MIMIC, and CheXpert datasets, respectively. This allows us to answer many queries with high performance in terms of ACC, BAC, and MAP.

Similarly, we tune the $\tau$ values for $\ell_2$ norm in Figure 12 and for $\ell_1$ norm in Figure 13.

Regarding the values of parameters $\sigma_G$, $T$, and $\sigma_T$, we follow the original work on PATE (Papernot et al., 2017) and perform a grid search over a range of plausible values for each of the hyper-parameters.

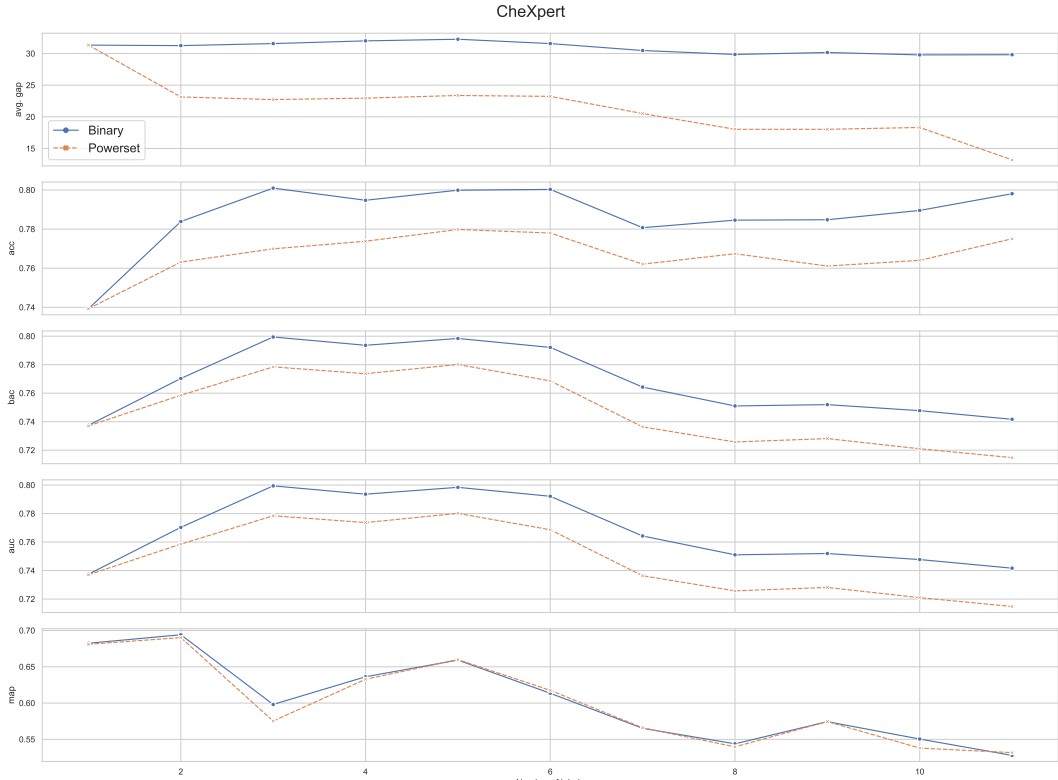

Figure 11: **Binary vs Powerset PATE: gaps and metrics for CheXpert**. We use 50 teacher models. These are raw gaps without adding any noise to the vote histograms.

The detailed analysis of tuning the hyper-parameters are for: $\tau$-s in Section I.5, $\sigma_G$ in Section I.6, PATE's thresholding ($T$, and $\sigma_T$) in Section I.7, and probability thresholds in Section I.10.

### I.5 TUNING $\tau$-CLIPPING

We determine the minimum value of $\tau$ for clipping in $\ell_2$ and $\ell_1$ norms for each dataset so that the performance as measured by accuracy, BAC, AUC, and mAP, is preserved. We present the analysis in Figures 12 and 13.

### I.6 TUNING $\sigma_{GNMax}$

We determine how much of the privacy noise, expressed as the scale of Gaussian noise $\sigma$, can be added so that the performance as measured by accuracy, BAC, AUC, and mAP, is preserved. The experiment is run using the binary PATE per label and presented in Figure 14.

### I.7 TUNING PATE

There are three parameters to tune: the differential privacy Gaussian noise standard deviation $\sigma_G$, the count threshold $T$, and the thresholding Gaussian noise standard deviation $\sigma_T$. We first disable the thresholding ($\sigma_T$ and $T$) and tune $\sigma_G$ to achieve a high BAC of the noisy ensemble while maximizing the number of answered queries. In Figure 19 we tune the $\sigma_G$ parameter from PATE for different configurations of retraining. The goal is to maintain a high accuracy while selecting $\sigma_G$ with high value so that as many queries as possible are answered. After tuning $\sigma_G$, we grid-search $\sigma_T$ and $T$. Thresholding is useful if we want to add a relatively small amount of Gaussian noise ($\sigma_G$) in the noisy max. For example, for $\sigma_G = 7$ and $50$ teacher models trained on CheXpert, the maximum

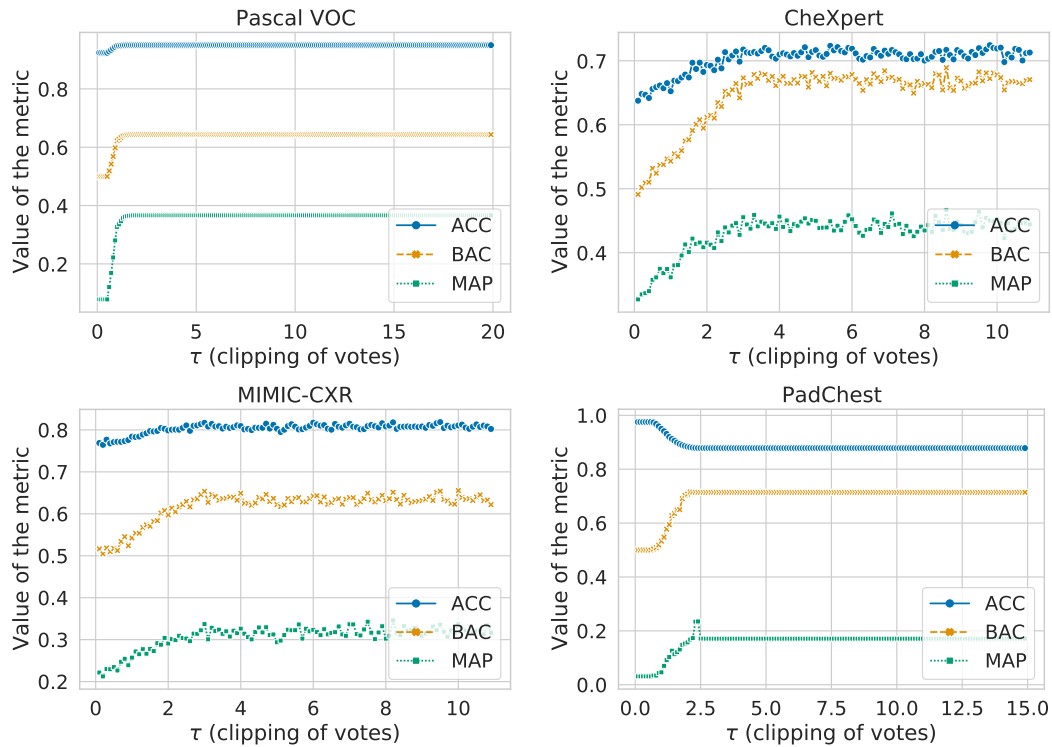

Figure 12: **Value of the metric vs $\tau$-clipping of votes in $\ell_2$ norm.** For a given $\tau$, we plot accuracy (ACC), balanced accuracy (BAC), and mean average precision (mAP).

number of labels answered without thresholding is 737 (67 queries) with a BAC above $0.69$. With thresholding, e.g., at $T = 50$ and $\sigma_T = 30$, we can answer an average of 756 labels, up to $834$. A well tuned threshold can also help reduce $\sigma_G$, which is the major influencing factor on the final BA of the noisy ensemble. Note that tuning the threshold benefits from a confident ensemble: here, we find the ensemble is often confident with a positive-negative vote difference of $40$ of max $49$. Also note that the BAC can be higher on easier queries.

### I.8 COMPARE DIFFERENT METHODS

In Figure 15, we directly compare the Binary PATE (denoted as PATE), $\tau$-PATE, and clipping in $\ell_2$ and $\ell_1$ norms. The error regions are for the three querying parties.

### I.9 VARYING PRIVACY BUDGET $\epsilon$

In Table 12, we show the performance of the retrained models when using CaPC with various privacy budgets $\varepsilon$. In Figure 16 we show the detailed per label change in BAC for the retraining with privacy budget $\varepsilon = 10$. We observe an increase in average BA by around $0.03$ after retraining when the privacy budget is set to $\varepsilon = 10$. We present detailed analysis of the Binary PATE performance in Table 10 on the Pascal VOC dataset when selecting the privacy budget $\varepsilon$ in the range from 1 to 20.

We also compare the performance after re-training with and without the (confidence) thresholding mechanism used in PATE (represented by $\sigma_T$ and threshold $T$ parameters). We observe that in case of the medical datasets, there can be a slightly higher increase of the metrics (accuracy, BAC, AUC, mAP), when we do not perform the thresholding. For example, for the CheXpert dataset, when thresholding is not used, the improvement is higher by about one percentage point across all metrics when compared to the option with thresholding. For PadChest, such increase is for AUC and mAP, for BAC we see a drop by one percentage point, and there is no difference in terms of accuracy (remains at the level of about $0.86$). This requires further investigation. Intuitively, we observe that the metrics

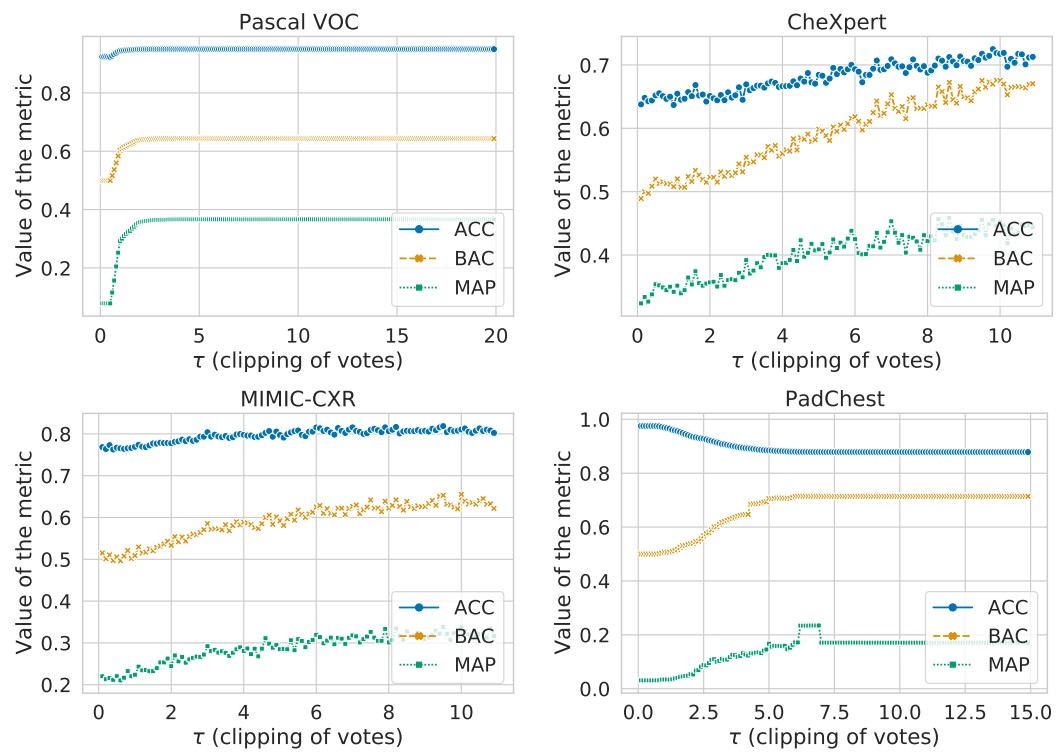

Figure 13: **Value of the metric vs $\tau$-clipping of votes in $\ell_1$ norm.** For a given $\tau$, we plot accuracy (ACC), balanced accuracy (BAC), and mean average precision (mAP).

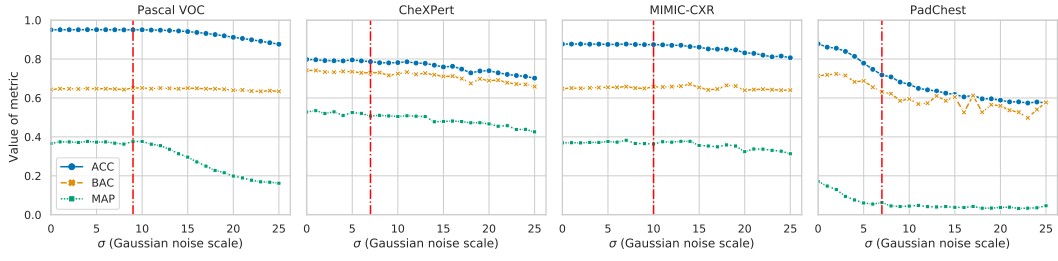

Figure 14: **Value of the metric versus noise standard deviation $\sigma$.** For a given $\sigma$, we plot accuracy (ACC), balanced accuracy (BAC), and mean average precision (mAP).

can increase substantially with thresholding for the labels (pathologies) that are easy to classify. However, the most difficult to predict pathologies are left without answer and no improvement is made on them. On the other hand, without thresholding, we have to answer all labels and correct predictions on the hardest labels can produce substantial improvement in the classification of these hard labels.

### I.10 TUNING PROBABILITY THRESHOLD

We tune the global (applied to all labels) probability threshold $\gamma$ that determines if a given probability denotes positive ($P > \gamma$), or a negative ($P \leq \gamma$) vote. The best global $\gamma$ for the 50 models trained on the CheXpert dataset is $0.32$ with the balanced accuracy of $0.702$ and the $\mu$ value of $0.45$, where $\mu$ is the average probability of predictions after applying the element-wise sigmoid function to the logits.

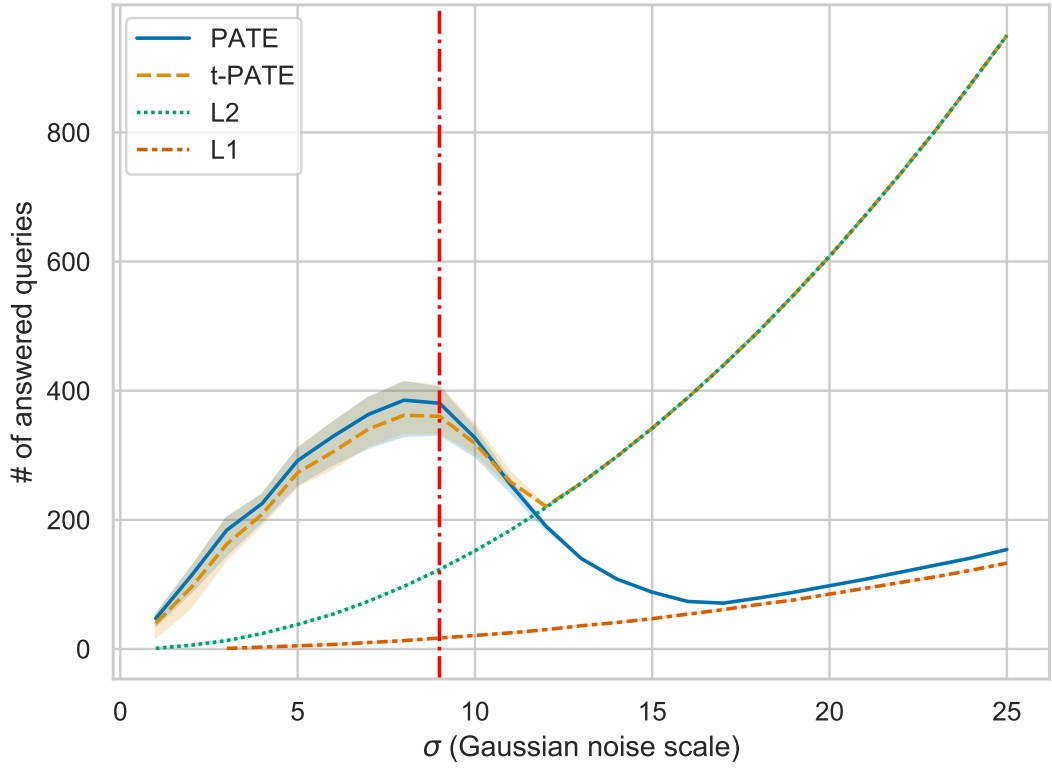

Figure 15: **Compare methods: number of answered queries vs** $\sigma$ - using the Pascal VOC dataset, with $\tau_1 = 3.4$ (set for $\ell_1$−norm clipping, and $\tau_2 = 1.8$ set for $\tau$-PATE and $\ell_2$−norm clipping. The red vertical line denotes the selected value of $\sigma$.

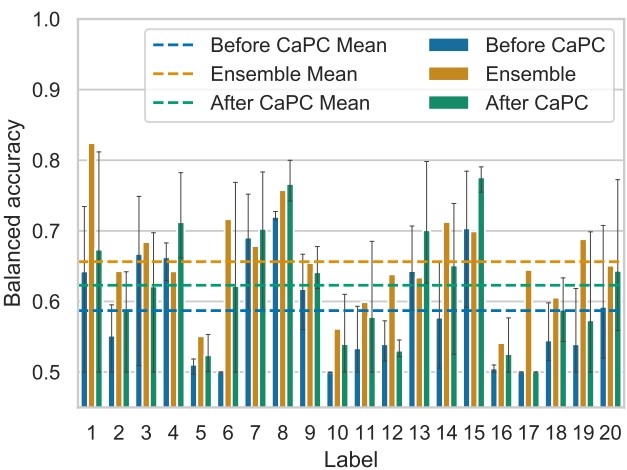

Figure 16: **Retraining with privacy budget** $\varepsilon = 10$ for the Pascal VOC dataset.

We also tune the $\gamma$ threshold per label for CheXpert. We find the following probability thresholds per label using the validation set on the ensemble: $0.53, 0.5, 0.18, 0.56, 0.56, 0.21, 0.23, 0.46, 0.7, 0.2, 0.32$. Next, we select the 3 teacher models with the lowest BAC and retrain them with CaPC. The average metrics BAC, AUC, and mAP

Table 10: **Pascal VOC with 20 labels: Performance of Binary PATE for different values of** $\varepsilon$ w.r.t. number of answered queries, ACC, BAC, AUC, and mAP as measured on the test set with the specified $\sigma_{\text{GNMax}}$. PB ($\varepsilon$) is the privacy budget. We use 50 teacher models. When we limit number of labels per dataset, we select the first $k$ labels. We set $\sigma_{\text{GNMax}} = 7$

| PB ($\varepsilon$) | QUERIES ANSWERED | ACC | BAC | AUC | MAP |
|---|---|---|---|---|---|
| 1 | **0** | - | - | - | - |
| 2 | **6** | .86 | .62 | .62 | .44 |
| 3 | **13** | .93 | .67 | .67 | .53 |
| 4 | **22** | .93 | .64 | .64 | .44 |
| 5 | **31** | .95 | .63 | .63 | .39 |
| 6 | **40** | .95 | .67 | .67 | .45 |
| 7 | **64** | .95 | .64 | .64 | .35 |
| 8 | **81** | .95 | .66 | .66 | .40 |
| 9 | **101** | .95 | .60 | .60 | .28 |
| 10 | **113** | .96 | .63 | .63 | .30 |
| 11 | **135** | .96 | .64 | .64 | .33 |
| 12 | **165** | .96 | .65 | .65 | .35 |
| 13 | **199** | .96 | .63 | .63 | .32 |
| 14 | **217** | .96 | .64 | .64 | .35 |
| 15 | **239** | .96 | .63 | .63 | .32 |
| 16 | **272** | .96 | .63 | .63 | .31 |
| 17 | **306** | .96 | .63 | .63 | .30 |
| 18 | **332** | .96 | .63 | .63 | .31 |
| 19 | **362** | .96 | .63 | .63 | .30 |
| 20 | **403** | .96 | .63 | .63 | .30 |

Table 11: **Performance of multi-label CaPC** w.r.t. ACC, BAC, AUC, and mAP, on Pascal VOC (PA), CheXpert (CX), MIMIC (MC), and PadChest (PC). (-) denotes N/A. PB ($\varepsilon$) is the privacy budget.

| DATASET | # OF MODELS | STATE | PB ($\varepsilon$) | ACC | BAC | AUC | MAP |
|---|---|---|---|---|---|---|---|
| PA | 1 | INITIAL | - | .97 | .85 | .97 | .85 |
| | 50 | BEFORE CAPC | - | .93±.02 | .59±.01 | .88±.01 | .54±.01 |
| | 50 | AFTER CAPC | 10 | **.94±.01** | **.62±.01** | .88±.01 | .54±.01 |
| | 50 | AFTER CAPC | 20 | **.94±.01** | **.64±.01** | **.89±.01** | **.55±.01** |
| CX | 1 | INITIAL | - | .79 | .78 | .86 | .72 |
| | 50 | BEFORE CAPC | - | .77±.06 | .66±.02 | .75±.02 | .58±.02 |
| | 50 | AFTER CAPC | 20 | .76±.07 | **.69±.01** | **.77±.01** | **.59±.01** |
| MC | 1 | INITIAL | - | .90 | .74 | .84 | .51 |
| | 50 | BEFORE CAPC | - | .84±.07 | .63±.03 | .78±.03 | .43±.02 |
| | 50 | AFTER CAPC | 20 | **.85±.05** | **.64±.01** | **.79±.01** | **.45±.03** |
| PC | 1 | INITIAL | - | .86 | .79 | .90 | .37 |
| | 10 | BEFORE CAPC | - | .90±.01 | .64±.01 | .79±.01 | .16±.01 |
| | 10 | AFTER CAPC | 20 | .88±.01 | .64±.01 | .75±.01 | .14±.01 |

before retraining are $0.63, 0.70, 0.53$, and after retraining, we observe a significant improvement to $0.68, 0.75, 0.58$, respectively. Thus, the value for each metric increases by around $0.05$. We find the biggest performance improvement for these weakest models after learning from better teachers and retraining. We present detailed results per label and for BA as well as AUC metrics in Figure 18.

## I.11 CROSS-DOMAIN RETRAINING

In real-world healthcare scenarios, there are often rare diseases that may be difficult to accurately model with machine learning. Even coalitions of hospitals sharing similar data distributions of the same domain may not see benefits, due to poor aggregate performance. However, cross-domain collaboration through multi-label CaPC can help improve performance in these cases. The hospital

Table 12: **Performance of multi-label CaPC with $\tau$-PATE** w.r.t. ACC, BAC, AUC, and mAP, on Pascal VOC (PA), CheXpert (CX), MIMIC (MC), and PadChest (PC). (-) denotes N/A. PB ($\varepsilon$) is the privacy budget. T (Y/N) in the table refers to the PATE thresholding i.e. corresponding to the parameters $T$ and $\sigma_T$ being used (Y) or not used (N). Note that the probability thresholding per layer was used in these experiments (as described in Section I.10).

| DATASET | # OF MODELS | STATE | T(Y/N) | PB ($\varepsilon$) | ACC | BAC | AUC | MAP |
|---|---|---|---|---|---|---|---|---|
| PA | 1 | INITIAL | - | - | .88 | .97 | .91 | |
| | 50 | BEFORE CaPC | - | - | .93±.01 | .59±.01 | .89±.01 | .54±.02 |
| | 50 | AFTER CaPC | N | 20 | **.94±.01** | **.60±.01** | .89±.01 | .54±.02 |
| CX | 1 | INITIAL | - | - | .79 | .78 | .86 | .72 |
| | 50 | BEFORE CaPC | - | - | .75±.02 | .69±.01 | .77±.01 | .59±.01 |
| | 50 | AFTER CaPC | Y | 20 | .73±.01 | **.69±.01** | .76±.01 | **.59±.01** |
| | 50 | AFTER CaPC | N | 20 | .74±.01 | **.70±.01** | **.77±.01** | **.59±.01** |
| MC | 1 | INITIAL | - | - | .90 | .74 | .84 | .51 |
| | 50 | BEFORE CaPC | - | - | .84±.07 | .63±.03 | .78±.03 | .43±.02 |
| | 50 | AFTER CaPC | Y | 20 | .84±.02 | **.64±.04** | .77±.02 | **.44±.01** |
| PC | 1 | INITIAL | - | - | .86 | .79 | .90 | .37 |
| | 10 | BEFORE CaPC | - | - | .82±.01 | .64±.01 | .79±.01 | .17±.01 |
| | 10 | AFTER CaPC | Y | 20 | **.86±.04** | .61±.01 | .71±.03 | .14±.02 |
| | 10 | AFTER CaPC | N | 20 | **.86±.02** | .60±.01 | .72±.02 | .15±.03 |

annotation discrepancies may pose barriers: here, we simulate this by the different X-ray image labels between PadChest and CheXpert (see Supplement Section G).

To overcome this, we take the union of labels between the medical datasets and follow the experimental setup of Cohen et al. (2020). We observe poor performing models on the PadChest dataset, with a low average performance of BAC = 0.57. Because of this, the benefits of multi-label CaPC within this coalition of hospitals are limited, since the ensemble is only marginally better. However, if this group of hospitals collaborated with another from a different but related domain, here represented by CheXpert, they may be able to see additional benefits. The models trained on this dataset achieve a higher BA (particularly on the first 5 shared pathologies as presented in Figure 17). Thus, the ensemble of all models engage in multi-label CaPC and the models trained on CheXpert act as answering parties to provide labels for querying parties from PadChest. Using a $\sigma_G = 9, T = 50$, and $\sigma_T = 30$, we observe a higher BAC on all of those 5 pathologies, where we do not see a significant decrease on the other pathologies. Thus, multi-label CaPC provided benefits for the PadChest models in this cross-domain scenario.

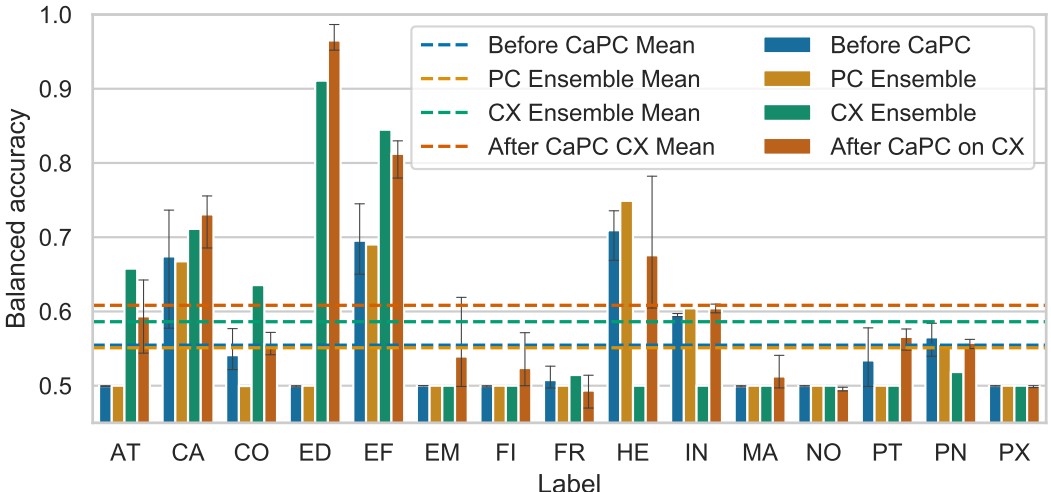

Figure 17: **Cross-domain retraining with CaPC.** We train 10 models on PadChest (PC) and compare their performance on PadChest test set against the ensemble of these models (PC Ensemble), ensemble of 50 CheXpert models (CX Ensemble), and finally retrain the 10 PadChest models via binary multi-label PATE using the CheXpert ensemble (after CaPC on CX).

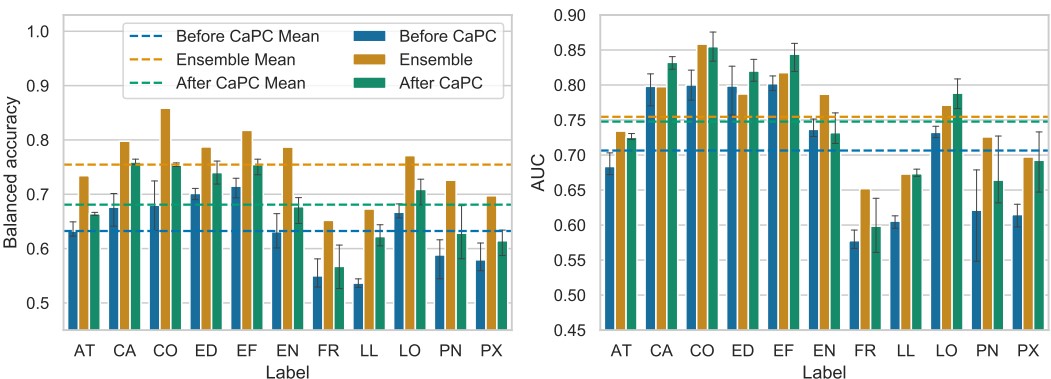

Figure 18: **Using CaPC to improve the weakest models.** *Dashed lines represent mean values of the metrics: Balanced Accuracy (BAC) and AUC.* We retrain a given model using additional CheXpert data labelled by the ensemble of all the other models trained on CheXpert. All metrics are improved after retraining by around $0.05$ on average.

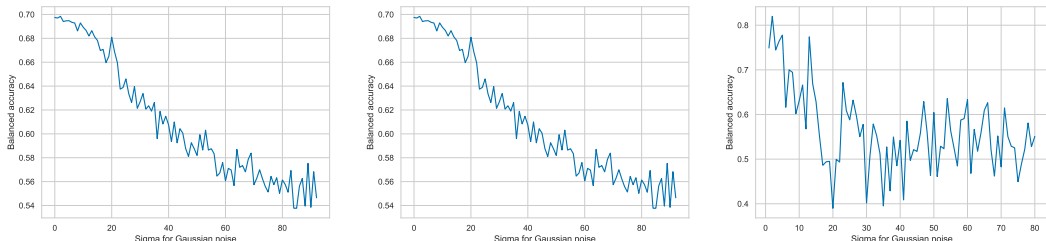

Figure 19: The analysis of the balanced accuracy (y-axis) of votes from the ensemble as we increase the (sigma of the Gaussian) noise (x-axis) from the binary multi-label PATE. We search for variance of the noise $\sigma_G$ that can preserve high BAC for the ensemble. **Left**: 50 CheXpert models with train and test on CheXpert, $\sigma_G \leq 9.0$ preserves more than 0.69 of BAC. **Middle**: 50 CheXpert models with train on CheXpert and test on PadChest test set, $\sigma_G \leq 9.0$ preserves more than 0.75 of BAC. **Right**: 10 PadChest models with train and test on PadChest, $\sigma_G \leq 5.0$ preserves more than 0.75 of BAC.

## J    THE CAPC PROTOCOL

The CaPC protocol is as follows:

1. **Private inference** is run for each answering party.
   (a) **Compute logits.** A querying party (or, student) sends an encrypted query (unlabeled $x$) to each answering party (or, teacher). The teachers run the private inference and upon completion return encrypted logits: $Enc(r)$.
   (b) **Secret sharing** between the answering and querying parties. Each answering party generates a random vector $\hat{r}$ and subtracts it from the vector of logits $r$. Then, each answering party sends the encrypted result: $Enc(r - \hat{r})$ to the querying party who decrypts the message to obtain $(r - \hat{r})$.
   (c) **One-hot encoding** of the logits. The answering parties provide $\hat{r}$ and the querying party provides $(r - \hat{r})$ to run Yao's garbled circuit protocol (Yao, 1986b). They compute the argmax of the logits and transform them into one-hot representations. Each one-hot vector is split into shares $s$ and $\hat{s}$. The $s$ share is returned to the querying party and the $\hat{s}$ share is sent to the Privacy Guardian (PG). The PG is a common facilitator in these setups, i.e., a trusted third-party, and is honest-but-curious. The sum of shares $s$ and $\hat{s}$ is the one-hot encoding of the predicted label.

2. **DP mechanism.** This step follows PATE to provide privacy for each teacher's data. In PATE, Gaussian noise is added to the argmax. In CaPC, Gaussian noise is added to the sum of $\hat{s}$ shares by the PG, after being collected from each of the answering parties.

3. **Noisy argmax** to compute the final label. The PG and the querying party run Yao's garbled circuit to sum the noisy $\hat{s}$ values with the $s$ values. The result is a noisy histogram representing the number of votes for each class. The argmax of this determines which class received the maximum number of votes. This class is returned to the querying party as the final predicted label.

In CaPC learning Choquette-Choo et al. (2021), the main tool for confidentiality is secure multi-party computation (MPC) (Yao, 1986a) and homomorphic encryption. MPC is used when distrusting parties want to jointly evaluate a function on their input without revealing anything beyond the output. The MPC protocols can be divided into two main classes: (1) generic protocols, which compute any function with the Malkhi et al. (2004) security goal, and (2) specialized protocols, which are used to compute selected functions, for instance, the private set intersection (Pinkas et al., 2020) or secure machine learning (Mohassel & Zhang, 2017). Even though the specialized protocols are less general, they are more efficient in terms of the execution time. Protocols in both categories use similar cryptographic building blocks, including (fully) homomorphic encryption (Gentry, 2009), secret sharing (Shamir, 1979), oblivious transfer (Rabin, 2005), garbled circuits (Yao, 1986a). Private inference can be computed in many ways and is a component of the CaPC protocol that directly scales with improvements in these techniques: CaPC used the HE-transformer library that uses both MPC as well as the homomorphic encryption but the private inference library can be immediately swapped with others.

---

**Algorithm 1** CaPC protocol

---

**Input:** Querying Party QP. $x$ - query (unlabeled data item) from QP. Answering Parties $AP_1$,...,$AP_n$ with ML models $f_i(x) = y$, where $i \in \{1, ..., n\}$, and $y$ is the vector of logits. Privacy Guardian PG.
**Output:** label $\ell$ for query $x$ (QP receives $\ell$ for its $x$).

1: QP encrypts $x$ to $Enc(x)$.
2: QP sends $Enc(x)$ to $AP_1$,...,$AP_n$.
3: **for all** $AP_i$ **do**
4:     $Enc(r)_i \leftarrow f_i(Enc(x))$
5:     Generate random $\hat{r}_i$
6:     $Enc(r_i - \hat{r}_i) \leftarrow Enc(r_i) - \hat{r}_i$                    ▷ Enabled by Homomorphic Encryption
7:     Send $Enc(r_i - \hat{r}_i)$ to QP
8: **for all** $AP_i$ and QP **do**          ▷ 2PC (Secure 2-Party Computation) between each AP and the QP
9:     QP decrypts $r_i - \hat{r}_i \leftarrow Decrypt(Enc(r_i - \hat{r}_i))$
10:     QP sends $r_i - \hat{r}_i$ to 2PC
11:     $AP_i$ sends $\hat{r}_i$ 2PC
12:     2PC: $r_i \leftarrow (r_i - \hat{r}_i) + (\hat{r}_i)$
13:     2PC: $one - hot_i \leftarrow argmax(r_i)$
14:     2PC: $\hat{s}_i + s \leftarrow one - hot_i$                    ▷ Secret Sharing
15:     $AP_i$ receives $\hat{s}_i$ from 2PC
16:     QP receives $s_i$ from 2PC
17: **for all** $AP_i$ **do**
18:     Send $\hat{s}_i$ to PG.
19: QP: $S \leftarrow \sum_{i=1}^{n} s_i$
20: QP sends $S$ to 2PC
21: PG: $\hat{S} \leftarrow \sum_{i=1}^{n} \hat{s}_i$
22: PG: Generate Gaussian noise $\hat{n} \leftarrow N(0, \sigma_G)$
23: PG sends $\hat{S}$ and $\hat{n}$ to 2PC
24: 2PC: $\hat{h} \leftarrow S + \hat{S} + \hat{n}$                    ▷ Noisy histogram $\hat{h}$
25: 2PC: $\ell \leftarrow argmax(\hat{h})$
26: 2PC sends $\ell$ to QP

---

# K  ALGORITHMS FOR THE MULTI-LABEL CLASSIFICATION

---

**Algorithm 2** Multi-label classification with $\tau$-clipping in $\ell_2$-norm and with the Confident GNMax.

---

**Input:** Data point $x$, clipping threshold $\tau_2$, Gaussian noise scale $\sigma_G$, Gaussian noise scale for Confident GNMax $\sigma_T$, $n$ teachers, each with model $f_j(x) \in \{0,1\}^k$, where $j \in [n]$

**Output:** Aggregated vector $V$ with $V_j = 1$ if returned label/feature present, otherwise $V_j = 0$.

  1: **for all** teachers $j \in [n]$ **do**

  2:     $v_j \leftarrow \min(1, \frac{\tau_2}{\|f_j(x)\|_2}) f_j(x)$                                                $\triangleright$ $\tau$-clipping in $\ell_1$ norm

  3:  $V^1 = \sum_{j=1}^n v_j$                                        $\triangleright$ Number of positive votes per label

  4:  $V^0 = n - V^1$

  5: **for all** labels $i \in [k]$ **do**

  6:     **if** $\max\{V_i^0, V_i^1\} + \mathcal{N}(0, \sigma_T) < T$ **then**

  7:        $V_i = \perp$                                            $\triangleright$ Confident-GNMax

  8:     **else**

  9:        $V_i^0 \leftarrow V_i^0 + \mathcal{N}(0, \sigma_G)$                     $\triangleright$ Add Gaussian noise for privacy protection

10:        $V_i^1 \leftarrow V_i^1 + \mathcal{N}(0, \sigma_G)$

11:        **if** $V_i^1 > V_i^0$ **then**                           $\triangleright$ Decide on the output vote

12:           $V_i = 1$

13:        **else**

14:           $V_i = 0$

---

**Algorithm 3** Multi-label classification with $\tau$-clipping in $\ell_1$-norm from Private kNN by Zhu et al. (2020).

---

**Input:** Data point $x$, clipping threshold $\tau_1$, Gaussian noise scale $\sigma_G$, $n$ teachers, each with model $f_j(x) \in \{0,1\}^k$, where $j \in [n]$.

**Output:** Aggregated vector $V \in \{0,1\}^k$ with $V_i = 1$ if returned label present, otherwise $V_i = 0$, where $i \in [k]$.

  1: **for all** teachers $j \in [n]$ **do**

  2:     $v_j \leftarrow \min(1, \frac{\tau_1}{\|f_j(x)\|_1}) f_j(x)$                                               $\triangleright$ $\tau$-clipping in $\ell_1$ norm

  3:  $V^1 = \sum_{j=1}^n v_j$                                         $\triangleright$ Number of positive votes per label

  4:  $V^0 = n - V^1$

  5:  $V^0 \leftarrow V^0 + \mathcal{N}(0, \sigma_G)$                         $\triangleright$ Add Gaussian noise for privacy protection

  6:  $V^1 \leftarrow V^1 + \mathcal{N}(0, \sigma_G)$

  7: **for all** labels $i \in [k]$ **do**

  8:     **if** $V_i^1 > V_i^0$ **then**                                 $\triangleright$ Decide on the output vote

  9:        $V_i = 1$

10:     **else**

11:        $V_i = 0$

---

## L    DATA-DEPENDENT PRIVACY ANALYSIS

Our binary voting mechanism does leverage both the smooth sensitivity and propose-test-release methods. For example, the Confident GNMax (proposed by Papernot et al. (2018)) is a form of the propose-test-release method described in Section 3.2 in (Vadhan, 2017).

First explored by Cormode et al. (2012), **differential privacy guarantees can be data-dependent**. These guarantees can lead to better utility with a long history, with several works using them (Cormode et al., 2012; Papernot et al., 2017; 2018; Chowdhury et al., 2020). One main caveat is that the released epsilon score must now also be noised because it is itself a function of the data. Papernot et al. (2018) provide a way to do this via the smooth sensitivity (Section B in Appendix). Indeed our data-dependent differential privacy guarantees are formally proven and are based on the following intuition. Take the exponential mechanism which gives a uniform privacy guarantee. For instance, when the top score and the second score are very close, applying the exponential mechanism to this data there is a nearly uniform chance of picking either coordinate. However, for some inputs, the utility can be very strong—when the top score is much higher than the second score, then this mechanism is exponentially more likely to pick the top score than the second. This does not require local sensitivity of stability based methods, but is rather derived from the likelihood of picking either coordinate for this mechanism given the gap in the scores.

