# OpenReview forum: "Private Multi-Winner Voting For Machine Learning"
_ICLR.cc/2022/Conference — ICLR 2022 Submitted_

### Official Review · Reviewer_3vmh · 2021-11-01

**Correctness:** 3
**Technical Novelty And Significance:** 2
**Empirical Novelty And Significance:** 3
**Recommendation:** 3
**Confidence:** 4

**Main Review:**

Final update: After the discussion period, I still vote for rejection. There are two main reasons.
1) The authors do not seem to understand the issue I raised and clarified over and over again. Now I started to doubt whether the algorithm implemented in the experiment is end-to-end differentially private with the designed privacy budget. (I) I do not agree that data-dependent privacy is a different privacy guarantee or a relaxed version of differential privacy. I believe it is not private and give my reasons through the Alice Bob example. I understand this is not the first paper to do it under data-dependent privacy and I have made it clear that I strongly disagree with all the other works especially Papernot et al. (II) The authors responded by citing the monograph by Dwork and Roth, which clearly shows that they do not understand the work they quoted and the issue I raised here. In the entire monograph, the differential privacy is based on the worst-case dataset (data-independent). The authors claim that "data-dependent analysis is an improvement over propose-test-release". The propose-test-release is an algorithm. The data-dependent privacy guarantee is a weak privacy notion. The data-dependent analysis referred to as data-dependent accuracy bound in this book is about utility analysis, not privacy. I do understand how privacy notion can be an improvement on an algorithm. Most importantly, this paragraph by Dwork and Roth is saying, you can improve your utility with local sensitivity because privacy has to be for the worst-case dataset, but the utility is only for the good dataset. Typically, to propose a private algorithm, you first prove the privacy for any dataset without any assumptions on the data and then prove the utility for some good datasets like Gaussian data. (This monograph explicitly used the word data-dependent **Accuracy** bounds not privacy bounds)  (II) The authors claim that they can fix it by controlling different numbers of queries. However, the authors did not formally prove it. More specifically, in the authors' data-dependent analysis, each query (each sample in the public unlabeled dataset) has a different privacy cost.  I do not see how Alice could determine the number of queries before running it. Now suppose the privacy budget is 10, and Alice has reached 9.9. when Alice found that her last query has privacy cost 0.2 and exceeds the overall privacy budget, she cannot hide it from the public anymore. Thus, the overall process would break the eps=10 privacy promise. I believe this is fixable since there is some new work on private hyper-parameter tuning. But like I said, the main point here is you have to prove it satisfies (data-independent)-DP. (III) My main concern is about the released data being non-private not the epsilon. (IV) The comparison with Bayesian DP is not helpful here. As long as you privacy depends on the dataset, the same problem arises here for both notions. That is why all the other privacy notions are all data-independent.

2)The main contribution is to generalize PATE to multi-label. I think this is a valid problem setting. But the novelty is not enough for being a separate paper considering the fact that the three proposed methods are either minor modifications of prior works or simple generalizations of PATE. The tau mechanism is just a simple l2 generalization of Zhu et al 2019.

Update: You are missing the point. Like I said previously, there are many ways to make a "data-dependent mechanism" be provably data-independently differentially private. As long as you can prove your end-to-end algorithm is eps-10 differentially private for any input dataset, then you can make the promise. But it is not here in the current version. For example, you could spend some budget checking the dataset first. If it is good, run your mechanism. If it is bad, output fail. And this is exactly propose-test-release. The methods I mentioned below are all end-to-end data-independently differentially private. You could run your algorithm on all the possible input dataset S and find the largest one, i.e. $\max_S \varepsilon(S)$. This would also satisfy data-independent DP. The point is you have to prove that your algorithm is data-independently DP. Alice as a privacy engineer cannot make any assumptions about the nature that generates the dataset. There is a nice blog post https://differentialprivacy.org/average-case-dp/. It has a discussion about the pitfalls of Bayesian DP, which is (in my opinion) similar to data-dependent DP (data-dependent DP: $\varepsilon(S)$, Bayesian DP: $E_{S\sim P}[\varepsilon(S)])$, data-independent DP: $\max_S\varepsilon(S)$).  I believe the proposed algorithm in this paper can be proved to be data-independently private even without any change to the algorithm itself. You might want to prove the data-independent RDP of Confident GNMax. (Theorem B.2 is data-dependent). Another caveat I forgot to mention is about comparisons with other works. The gap between data-dependent DP and data-independent DP can be huge (also reported in this paper). It is really unfair to compare with data-independent DP methods like DP-SGD.

Update regarding DP and data-dependent DP: This is already **irrelevant** to the contributions of this paper and the proposed algorithm since the proposed method can also be analyzed by traditional (data-independent) DP. First of all, I never claimed that the proposed algorithm is not differentially private(or completely vulnerable against membership inference attacks). And I never said data-dependent DP is not formal. My main concern is about the definition of privacy and the wording. My critique of data-dependent DP is based on the fact that this notion is a function of the dataset. Now suppose we are the nature that generates the dataset. Alice is the privacy engineer who runs some private algorithm and releases the output to the public. Bob is the adversary who tries to attack the sensitive information in the dataset with the released information. Differential privacy is a promise made by Alice to defend against attacks such as membership inference attack. Now when Alice claims that her algorithm is eps=10-DP. We would expect that the output to be eps=10-DP differentially private with respect to whatever the dataset we give to Alice. If not, the next time, when we generate a "bad" dataset such that it is not eps=10-DP, Bob can perform a membership inference attack with higher accuracy than promised by Alice. Stability-based methods/propose-test-release/smooth-sensitivity-based methods are all adaptive to the dataset by using local sensitivity but satisfy promised privacy budget for whatever input dataset. Different variations of differential privacy like RDP/CDP/zCDP are all with respect to any input datasets(data-independent). However, if you calculate and output an epsilon of your algorithm for a particular dataset (even if it is private with respect to the dataset), you get different epsilons for different datasets (even if you assume the same number of queries). Then how can you make that promise? This is why I believe data-dependent privacy notions are useless. You could and should get a tight and data-independent epsilon by exactly the same algorithm (Confident GNMax). Listing previous papers do not make things correct. Please answer my doubts directly.

After the authors' response: 1) I understand this is not the main subject of this paper.  I still think data-dependent privacy is useless. Theoretically, you could come up with a mechanism and a dataset such that data-dependent epsilon is around 1 and data-independent epsilon is 1000, which is practically useless in defending say membership inference attack. I know this may raise some controversy since there are some prior works that compute some data-dependent epsilon, but I would like to emphasize that the reason why differential privacy is used as a gold standard for private data analysis is that it is a worst-case definition. 2) Overall, I think for this specific problem, this paper might be doing the correct thing with the correct analysis. But the technical contributions are not enough.

Strengths:
1. this problem is an extension of PATE to a multi-label setting, which is practically interesting.
2. The authors compare three mechanisms empirically. The empirical analysis is solid.

Weaknesses:
1. Although the considered problem is practically useful, the proposition 3.1 and appendix A are not theoretically interesting. Basically it says there exists example such that the sensitivity is achieved. (Any private algorithm should have tight analysis of sensitivity) The authors claim that this shows binary mechanism is optimal. However, even for single winner setting, we do not know if Report noisy Argmax is optimal or not. (You could potentially improve it by stability-based methods/ propose-test-release, depending on the problems.)

2. It would be better if this paper contains a preliminary section for "data-dependent privacy bounds". In general, differential privacy should not be data dependent. So I am a little bit confused when I first read this concept. If I understand this correctly, I think this is related to local sensitivity/ or stability based methods (see Section~3 of [1]). For example, when the top candidate and the second candidate has large gap (high consensus), the local sensitivity can be small. You could potentially improve it by stability-based methods. But the end-to-end algorithm must satisfy data-independent differential privacy.

3. The emprical comparisons with DP-SGD is not fair. DP-SGD algorithms do not assume a public unlabeled dataset. I understand this comparison exists in prior works. It would be better if the authors could acknowledge it.

4. The discussion about related work Zhu et al seems confusing. As far as I know, Zhu et al (2020) is also model agnostic. The only thing different from original PATE is that they replace teacher model with k nearest neighbor classifiers. Unlike DP-SGD, the privacy mechanism does not scale with the architecture of teacher models. Also the improvement of Tau mechanism is from l2 clipping, which is straightforward. As Zhu et al pointed out in their appendix, they do not use stability-based methods( or they called noisy screening/ Confident GNMax) because it is hard for a data sample to have high consensus for every label. I am just curious how this is resolved in this paper.

Some minor comments:
1. Is there a typo in equation (2)? I am a little bit confused about the brackets.

[1] Vadhan, Salil. "The complexity of differential privacy." Tutorials on the Foundations of Cryptography. Springer, Cham, 2017. 347-450.

**Summary Of The Paper:**

This paper considers differentially private multi-winner voting. This problem is a generalization of single-winner voting, which is widely used in PATE type private semi-supervised learning. The authors give three private mechanisms and perform empirical comparisons on multi-label semi-supervised learning settings.  Specifically, when there is no total votes constraint on each ballot, the binary mechanism performs report noisy argmax for each candidate. When there is a total votes constraint, the tau mechanism performs l2 clipping first and then performs report noisy argmax for each candidate as the binary mechanism. The powerset mechanism converts the multi-label problem into a single label problem and performs regular report noisy argmax.

**Summary Of The Review:**

Although this is a good practical problem, I think this paper needs some revision.

---

> ### Author Response · Authors · 2021-11-17
> **Related Work, $\tau$-clipping, DP-SGD, Binary voting & Confident GNMax, Definition 4**
>
> >**The discussion about related work by Zhu et al. seems confusing. As far as I know, Private kNN by Zhu et al (2020) is also model agnostic. The only thing different from the original PATE is that they replace the teacher model with $k$ nearest neighbor classifiers.**
>
> We updated the related work in Section 2.3. PATE can operate on any models, for example, random forest (e.g., see the original PATE paper [[1]](https://openreview.net/forum?id=HkwoSDPgg), Section C in Appendix), which is not the case for Private kNN (from the work by Zhu et al (2020)), which needs an embedding for inputs. Private kNN assumes the existence of a publicly-available embedding model, which is used to project high-dimensional data onto a sufficiently low-dimensional manifold for the meaningful computation of distances (to determine the nearest neighbors). Our setup does not assume the existence of any publicly-available model.
>
> [1] [Semi-supervised Knowledge Transfer for Deep Learning from Private Training Data](https://openreview.net/forum?id=HkwoSDPgg). Nicolas Papernot, Martín Abadi, Úlfar Erlingsson, Ian Goodfellow, Kunal Talwar. ICLR 2017.
>
> >**Also the improvement of $\tau$ mechanism is from $\ell_2$ clipping, which is straightforward.**
>
> We also think that this is straightforward in retrospect but the crux is that it gives much better privacy guarantees than the original proposal by Zhu et al. Furthermore, our work significantly departs from theirs by proposing new analytical methods (e.g., incorporating both the data-dependent and improved data-independent analysis), advancing their theorems and code, as well as providing extensive evaluation across many multi-label tasks. We hope that our work can help facilitate further research in this area.
>
> >**Unlike DP-SGD, the privacy mechanism does not scale with the architecture of teacher models.**
>
> Indeed, in DP-SGD, the mechanism scales with the architecture because the output is the gradient itself. The benefit of our method is that for larger models we do not consume more privacy budget. This is an important benefit of our semi-supervised learning approach. It was shown that private ERM (and analogously DP-SGD) suffers from an additive error growing in the square root of the ambient dimension. However, our approach instead noisily labels data so that the model can use non-private training. This may significantly improve performance when training on the private data.
>
> >**As Zhu et al. pointed out in their appendix, they do not use stability-based methods (or they called noisy screening / Confident GNMax) because it is hard for a data sample to have high consensus for every label. I am just curious how this is resolved in this paper.**
>
> We agree that it is “hard” to have consensus for every label of an input. In cases where there is lacking consensus for only some labels, the binary voting mechanism leverages the Confident GNMax (noisy screening/propose-test-release) on a per-label basis to refrain from answering these “hard labels”. This mitigates large privacy loss that may be incurred in favor of the much smaller privacy loss for testing.  Algorithm 2 of Appendix K shows this approach, where for each label of sample $i$, the noisy number of positive or negative votes is compared against a threshold $T$.
>
> This is also resolved in our optimality analysis of the Binary voting mechanism and our analysis comparing it with our proposed Powerset voting. In particular, Powerset voting cannot perform the per-label Confident GNMax algorithm because it must release all or no labels. Thus, the Powerset method may incur larger privacy loss when there are “hard” labels entangled with “easy” ones.
>
> >**Is there a typo in equation (2) -> Definition 4? I am a little bit confused about the brackets.**
>
> We added additional brackets in Definition 4. In this DP Binary voting mechanism, we add the vote counts for each label (i) separately across all voters (j), then add noise, and then compare to a threshold. We are happy to further clarify if the reviewer would like us to.

---

> ### Author Response · Authors · 2021-11-17
> **Empirical comparisons with DP-SGD**
>
> >**The empirical comparisons with DP-SGD are not fair. DP-SGD algorithms do not assume a public unlabeled dataset. I understand this comparison exists in prior works. It would be better if the authors could acknowledge it.**
>
> We would like to thank the reviewer for pointing this out, we indeed followed the way DP-SGD and PATE were compared in prior publications. As suggested by the reviewer, we updated the following sentences in Section 5.2 to clarify the differences between DP-SGD and PATE:
>
> “Though the DP-SGD algorithm does not assume a public unlabeled dataset, we compare with it because it can be directly applied to this multi-winner setting. DP-SGD for multi-label was studied in prior work [1], and cannot directly leverage public unlabeled data. On the other hand, our PATE-based approaches leverage a public pool of unlabeled samples that have noisy labels provided by the ensemble of teachers. The added noise protects the privacy of the centralized training data. We then train the student on the newly labeled samples.”
>
> We compare PATE with DP-SGD as it is the only (to our knowledge) DP mechanism for machine learning that can be directly applied to this multi-winner setting. We agree that DP-SGD does not leverage unlabeled data. However, using these unlabeled samples for DP-SGD remains an open research question. Thus, even in scenarios where there exist many unlabeled samples, it is not entirely clear how to leverage this data - unlike in our approach.
>
> **References:**
>
> [1] [Adaptive privacy-preserving deep learning algorithms for medical data.](https://bit.ly/3qJsRfX) Xinyue Zhang, Jiahao Ding, Maoqiang Wu, Stephen T. C. Wong, Hien Van Nguyen, Miao Pan. WACV 2021.

---

> > ### Author Response · Authors · 2021-11-25
> > **Comparison with data-independent DP methods like DP-SGD**
> >
> > >**Another caveat I forgot to mention is about comparisons with other works. [...] It is really unfair to compare with data-independent DP methods like DP-SGD.**
> >
> > To enable easy comparison, we also compute the data-independent epsilons for our methods and clearly display them next to the data-dependent epsilons in Table 2 of the paper (also presented below). We run additional experiments where we use the same data-independent bound for both PATE and DP-SGD. As there is no other work in DP multi-label machine learning, we choose these baselines to provide benchmarks in this understudied area. We believe that our work helps establish grounds for future work. We remark that to the best of our knowledge there are currently not any data-dependent DPSGD algorithms to compare with.
> >
> > Our *Binary PATE* with sanitized $\varepsilon=20$ outperforms *DP-SGD* with $\varepsilon=20$ across all the metrics. Our new data independent method given by *L2-norm $\tau$-clipping DI* outperforms *DP-SGD* in terms of accuracy, balanced accuracy, and AUC, and has only lower mean average precision.
> >
> > **Table 2: Comparison between Binary PATE, DP-SGD, L2-norm $\tau$-clipping, and a Non-private model.** DI denotes Data Independent, and DD represents Data Dependent privacy analysis. We use the Pascal VOC dataset.
> >
> > | Method                    | Number of answered queries | Epsilon | Sanitized Epsilon | Accuracy (ACC) | Balanced Accuracy (BAC) | AUC (Area-Under-the-Curve) | mean Average Precision (mAP) |
> > |---------------------------|-------------------|---------|------------------|----------------|-------------------------|----------------------------|------------------------------|
> > | Non-private               | -                 | -       | -                | 0.97           | 0.85                    | 0.97                       | 0.85                         |
> > | DP-SGD                     | -                 | 20      | -                | 0.92           | 0.5                     | 0.68                       | 0.4                          |
> > | Powerset PATE DD          | 78                | 20      | 26            | 0.94           | 0.58                    | 0.7                        | 0.29                         |
> > | Binary PATE DD            | 427               | 20      | 26            | 0.94           | 0.62                    | 0.85                       | 0.57                         |
> > | Binary PATE DD            | 298               | 16      | 20            | 0.94           | 0.59                    | 0.8                        | 0.49                         |
> > | L2-norm $\tau$-clipping DI | 123               | 20      | -                | 0.94           | 0.57                    | 0.75                       | 0.33                         |

---

> ### Author Response · Authors · 2021-11-17
> **Data-dependent privacy analysis**
>
> >**It would be better if this paper contains a preliminary section for "data-dependent privacy bounds". In general, differential privacy should not be data-dependent. So I was a little bit confused when I first read this concept. If I understand this correctly, I think this is related to local sensitivity / or stability-based methods (see Section 3 of [1]). For example, when the top candidate and the second candidate have a large gap (high consensus), the local sensitivity can be small. You could potentially improve it by stability-based methods. But the end-to-end algorithm must satisfy data-independent differential privacy.**
>
> We have added a brief preliminaries section for data-dependent privacy in Appendix Section L with reference from the background in main text Section 2.1.
>
> Our binary voting mechanism does leverage the smooth sensitivity and propose-test-release methods. For example, the Confident GNMax (proposed in [4]) is a form of the propose-test-release method described in Section 3.2 in [1]. We have added a description in the main text of Sections 3.1 and 4.1 to clarify this.
>
> First explored by [2], differential privacy guarantees can be data-dependent. These guarantees can lead to better utility with a long history, with several works using them [2, 3, 4, 5]. One main caveat is that the released epsilon score must now also be noised because it is itself a function of the data. [4] provides a way to do this via the smooth sensitivity (Section B in Appendix). Indeed the data-dependent differential privacy guarantees are formally proven in [4] and are based on the following intuition:
>
> Take the exponential mechanism which gives a uniform privacy guarantee. For instance, when the top score and the second score are very close, applying the exponential mechanism to this data there is a nearly uniform chance of picking either coordinate. However, for some inputs, the utility can be very strong - when the top score is much higher than the second score, then this mechanism is exponentially more likely to pick the top score than the second. This does not require local sensitivity of stability-based methods but is rather derived from the likelihood of picking either coordinate for this mechanism given the gap in the scores.
>
> **References:**
>
> [1] [The complexity of differential privacy.](https://bit.ly/3qEvCiN) Salil Vadhan. Tutorials on the Foundations of Cryptography. Springer, Cham, 2017. 347-450.
>
> [2] [Differentially private spatial decompositions.](https://arxiv.org/abs/1103.5170) Graham Cormode, Magda Procopiuc, Entong Shen, Divesh Srivastava, Ting Yu. 2012 IEEE 28th International Conference on Data Engineering. IEEE, 2012.
>
> [3] [Data-Dependent Differentially Private Parameter Learning for Directed Graphical Models.](https://proceedings.mlr.press/v119/chowdhury20a.html) Amrita Roy Chowdhury, Theodoros Rekatsinas, Somesh Jha. ICML 2020.
>
> [4] [Scalable Private Learning with PATE.](https://openreview.net/forum?id=rkZB1XbRZ) Nicolas Papernot, Shuang Song, Ilya Mironov, Ananth Raghunathan, Kunal Talwar, Úlfar Erlingsson. ICLR 2018.
>
> [5] [Semi-supervised Knowledge Transfer for Deep Learning from Private Training Data](https://openreview.net/forum?id=HkwoSDPgg). Nicolas Papernot, Martín Abadi, Úlfar Erlingsson, Ian Goodfellow, Kunal Talwar. ICLR 2017.

---

> ### Author Response · Authors · 2021-11-17
> **Binary mechanism**
>
> We thank you for your detailed review.
>
> >**Although the considered problem is practically useful, proposition 3.1 and appendix A are not theoretically interesting. Basically, it says there exists an example such that the sensitivity is achieved. (Any private algorithm should have a tight analysis of sensitivity) The authors claim that this shows that the Binary mechanism is optimal. However, even for a single-winner setting, we do not know if Report noisy Argmax is optimal or not. (You could potentially improve it by stability-based methods / propose-test-release, depending on the problems.)**
>
> We agree that we do not know if the Report noisy Argmax mechanism is optimal; however, our proof does show that a Binary voting mechanism is optimal in the coordinate-independent setting. We have reworked the text in this section to clearly disentangle this statement with our proposed Binary voting mechanism based on the noisy argmax mechanism. We remark that though not provably optimal, the noisy argmax is widely used because of its highly favorable privacy guarantees arising from the tight sensitivity. Our approach does in fact leverage propose-test-release and stability-based methods because we leverage the data-dependent analysis of PATE for this binary setting.
>
> Though not theoretically interesting, we use this result to inform our empirical evaluation: if a setting is coordinate-independent or near-coordinate independent, then there is nothing to gain by using a coordinate-dependent approach. In fact, our theoretical analysis comparing the specific Binary and Powerset mechanisms that we propose demonstrates that in many cases, Binary voting mechanisms can significantly outperform coordinate-dependent mechanisms even when there is dependence due to tighter data-dependent analysis. This highlights the need for tighter data-dependent analysis in this setting. We accordingly focused our explanation of this setting on the insights gained rather than the technical details of the proof.

---

> ### Author Response · Authors · 2021-11-21
> **Data-dependent privacy analysis & Technical contributions**
>
> >**After the authors' response: 1) I understand this is not the main subject of this paper. I still think data-dependent privacy is useless. Theoretically, you could come up with a mechanism and a dataset such that data-dependent epsilon is around 1 and data-independent epsilon is 1000, which is practically useless in defending say membership inference attack. I know this may raise some controversy since there are some prior works that compute some data-dependent epsilon, but I would like to emphasize that the reason why differential privacy is used as a gold standard for private data analysis is that it is a worst-case definition. 2) Overall, I think for this specific problem, this paper might be doing the correct thing with the correct analysis. But the technical contributions are not enough.**
>
> We thank the reviewer for the new comments. These are our main points:
>
> 1. We do not provide new data-dependent guarantees but rather create new mechanisms that satisfy already existing peer-reviewed data-dependent guarantees that are well-regarded.
> 2. Data-dependent privacy bound does satisfy a formal DP guarantee which, among other properties, protects against membership inference attacks.
> 3. PATE offers strong empirical privacy despite a very conservative theoretical privacy analysis [5].
> 4. There are many works that use data-dependent DP: it is not new and is well-regarded in the machine learning and privacy communities.
> 5. Further, there is no (to the best of our knowledge) work that has studied issues regarding data-dependent DP.
> 6. Multi-label Differential Privacy is understudied and our mechanisms lead to SOTA results, outperforming the traditional approach of DP-SGD, which also relies on the data-dependent analysis [1].
>
> **DP versus Privacy Loss**
>
> We first disentangle the notion of a differential privacy guarantee and a data-independent or data-dependent privacy loss. A differential privacy guarantee must satisfy Definition 1 in our paper which is the standard differential privacy guarantee [1, 2, 3, 4]. What this essentially says is that the mechanism’s outputs must not deviate too much under the inclusion or exclusion of a single data point. To measure the notion of “deviation”, Definition 1 uses a multiplicative (exponential) epsilon between these two outcomes (inclusion or exclusion), this epsilon is known as a privacy budget. A core part of using a DP mechanism is calculating what the privacy loss was: the more queries to a mechanism the higher the loss is, indicating that an adversary can infer more about the underlying data.
>
> This is where the notion of data dependence or data independence comes into play. When calculating the privacy loss under the use of a DP mechanism, we can either calculate it using a data-independent bound or a data-dependent bound. **Both bounds satisfy DP and thus provide the formal guarantee.** The difference between these two is that the former calculates the loss based on the assumption that the mechanism is uniformly likely to reveal any output. However, when we empirically observe that this is not the case, i.e., that the mechanism will (nearly) always output the same response, then in fact the observed privacy loss is much lower. Note that this is still a formal DP guarantee, the only change is that the calculated privacy loss is lower. Also note that we do not propose any new data-dependent guarantee (we do propose a new data-independent one, however) in our work, but rather, leverage existing guarantees whenever possible.
>
> **Why does a data-dependent guarantee still protect against membership inference?**
>
> Using our mechanisms, which are one of our main contributions, we can extend existing privacy-preserving semi-supervised learning techniques from single-label (multi-class) settings to multi-label settings. We do this by replacing the existing mechanism with our mechanisms. In doing so, we create multi-label PATE. The data-dependent privacy budgets that we calculate protect against membership inference because the PATE formulation provides a DP guarantee w.r.t. the training data. In PATE, each contributed ballot (from a different teacher), was only influenced by one partition of the training data. This means that when excluding or including any data point, only one vote can be affected. So, high consensus (meaning we get a tighter data-dependent bound) means that we would have reached the same mechanism output **regardless of that data point, because most other ballots (from teachers that were not trained on that data point) reached that same conclusion as well.**  This exactly matches protecting against membership inference (and, in this case, also satisfying DP) because no individual data point largely influences the mechanism output. The data-independent bound in this case is just a looser bound: it does not better reflect the true privacy loss but is rather an overestimate of the true privacy loss.

---

> > ### Author Response · Authors · 2021-11-21
> > **References**
> >
> > [1] [Deep Learning with Differential Privacy.](https://arxiv.org/pdf/1607.00133.pdf
> > ) Martín Abadi, Andy Chu, Ian Goodfellow, H. Brendan McMahan, Ilya Mironov, Kunal Talwar, Li Zhang. CCS 2016.
> >
> > [2] [Semi-supervised Knowledge Transfer for Deep Learning from Private Training Data](https://openreview.net/forum?id=HkwoSDPgg).
> > Nicolas Papernot, Martín Abadi, Úlfar Erlingsson, Ian Goodfellow, Kunal Talwar. ICLR 2017.
> >
> > [3] [Scalable Private Learning with PATE](https://openreview.net/forum?id=rkZB1XbRZ). Nicolas Papernot, Shuang Song, Ilya Mironov, Ananth Raghunathan, Kunal Talwar, Úlfar Erlingsson. ICLR 2018.
> >
> > [3] [Concentrated differential privacy.](https://arxiv.org/abs/1603.01887) Cynthia Dwork and Guy N Rothblum. 2016
> >
> > [4] [Concentrated Differential Privacy: Simplifications, Extensions, and Lower Bounds](https://arxiv.org/abs/1605.02065). Mark Bun, Thomas Steinke. 2016
> >
> > [5] [Antipodes of Label Differential Privacy: PATE and ALIBI.](https://openreview.net/forum?id=sR1XB9-F-rv) Mani Malek Esmaeili, Ilya Mironov, Karthik Prasad, Igor Shilov, Florian Tramer. NeurIPS 2021.

---

> ### Author Response · Authors · 2021-11-22
> **Different datasets: same epsilons but different number of answered queries**
>
> >**Alice is the privacy engineer who runs some private algorithm and releases the output to the public. Bob is the adversary who tries to attack the sensitive information in the dataset with the released information. Differential privacy is a promise made by Alice to defend against attacks such as membership inference attack. Now when Alice claims that her algorithm is eps=10-DP. We would expect that the output to be eps=10-DP differentially private with respect to whatever the dataset we give to Alice. If not, the next time, when we generate a "bad" dataset such that it is not eps=10-DP, Bob can perform a membership inference attack with higher accuracy than promised by Alice. [...] However, if you calculate and output an epsilon of your algorithm for a particular dataset (even if it is private with respect to the dataset), you get different epsilons for different datasets (even if you assume the same number of queries). Then how can you make that promise?**
>
> We thank the reviewer for clarifying the concerns about data-dependent DP guarantees.
>
> In the scenario described above, one would use the data-dependent guarantee as follows:
> 1. Alice makes the promise that her private algorithm will be eps=10-DP.
> 2. Bob sends a dataset to Alice.
> 3. Alice trains her ensemble of models on the dataset sent by Bob.
> 4. While Bob sends queries to Alice:
>     - If the data-dependent privacy budget exceeds eps=10-DP: Alice stops responding.
>     - If the data-dependent privacy budget is still below eps=10-DP: Alice keeps responding and continues to add the privacy cost of the query to the privacy budget.
>
> This ensures that regardless of the dataset the private algorithm was run on, the private algorithm will always comply with the promise Alice made of being eps=10-DP.

---

> ### Author Response · Authors · 2021-11-25
> **Data-independent RDP of GNMax, The gap between data-dependent DP and data-independent DP, Ensuring end-to-end DP**
>
> >**You might want to prove the data-independent RDP of Confident GNMax. (Theorem B.2 is data-dependent).**
>
> Theorem B.2 is for the GNMax aggregator (and not for the *Confident* GNMax aggregator). We provided the data-independent RDP of GNMax in Lemma 3.2. (RDP-Gaussian mechanism for a multi-label setting). The proof is presented in Appendix C.
>
> >**The gap between data-dependent DP and data-independent DP can be huge (also reported in this paper).**
>
> We also proved a new data-independent bound that achieves tighter guarantees. These results are displayed in Figure 1 under the label L2-DI, which denotes L2-norm $\tau$ clipping of the ballots. L2-DI can be compared with the Binary PATE data-dependent analysis, which is under the label Binary PATE. For the “good” data sets like Pascal VOC, on which we achieve higher values of the metrics (Accuracy (ACC) 94%, BAC 64%, AUC 89%, MAP 55%), the data-dependent analysis allows us to release around 3.5x more queries (427 queries whereas the data-independent one only 123 queries, with all the parameters, such as the amount of Gaussian noise added, being equal). On the other hand, for the “worse” data sets like MIMIC (Accuracy (ACC) 85%, BAC 64%, AUC 79%, MAP 45%), the difference is less pronounced and the data-dependent analysis allows us to release around 1.9x more queries (94 queries whereas the data-independent one releases 48 queries).
>
> >**Ensuring end-to-end DP.**
>
> We agree that there can be privacy leakage through a released privacy budget if it is data-dependent and that this is the major differentiating factor between a data-dependent versus data-independent privacy bound. Knowing the unsanitized data-dependent privacy cost can enable side-channel attacks and makes additional assumptions about the trust model. To mitigate this, we add more details and emphasis surrounding the sanitizing of the privacy budget using smooth sensitivity analysis which was proposed in [1] to address this issue. Sanitizing the privacy budget ensures end-to-end protection of privacy and addresses the privacy implications of the data-dependent analysis. The analysis by [1] can be directly applied to our work (as we design mechanisms satisfying the requirements to leverage their guarantees). The sanitizing of the data-dependent privacy budget ensures that an adversary has no additional information of the end-to-end mechanism through observing the privacy parameters (e.g., prevents side-channel attacks). This end-to-end epsilon contains all privacy costs including the cost of noising the data-dependent budget. Their work shows that this final epsilon is only marginally larger <25% than the data-dependent epsilon. The following is the text we add as subsection 4.2 Data-Dependent Guarantees.
>
> "Because we use data-dependent privacy budgets, we cannot release these without first sanitizing them. Not doing so could reveal private information to a user of the private mechanism by using the unsanitized epsilons to learn additional information about the distributions of the database. This can lead to privacy leakage in cases such as those where the adversary has auxiliary information. Thus, we follow the procedure of [1] and publish sanitized privacy budgets using smooth sensitivity analysis of the privacy loss function. We pick the smoothness parameter $\beta$ to be $\frac{0.4}{\lambda}$ where $\lambda$ is the optimal RDP order obtained. From this, the smooth sensitivity can be calculated. We choose an optimal $\sigma_{SS}$ controlling the noise for the privacy budget to be the optimal value between $2 \cdot \sqrt{(\lambda+1) / \varepsilon}$ and $4 \cdot \sqrt{(\lambda+1) / \varepsilon}$ chosen by grid search. The final expected data-independent privacy cost is calculated using Theorem 23 of [1] using these parameters, added to the data-dependent privacy cost, and reported. We find that this increases the privacy cost by less than 30%, aligning with the results of [1]. Importantly, this still enables the sanitized data-dependent privacy cost to realize a significant reduction compared to the data-independent analysis. This final expected privacy cost protects the end-to-end DP of the data and can be safely published because the privacy cost $\varepsilon$ has been properly sanitized according to the smooth sensitivity of the mechanism."
>
> We further add additional details emphasizing that in addition to the above, the privacy engineer must take the minimum of the data-dependent bound and the data-independent bound: this ensures no additional privacy leakage by inspecting large data-dependent bounds.
>
> We present the results with sanitized epsilon values in Table 1 below.
>
> **References:**
>
> [1] [Scalable Private Learning with PATE.](https://openreview.net/forum?id=rkZB1XbRZ) Nicolas Papernot, Shuang Song, Ilya Mironov, Ananth Raghunathan, Kunal Talwar, Úlfar Erlingsson. ICLR 2018.

---

> > ### Author Response · Authors · 2021-11-25
> > **Table 1: Model improvements through retraining with multi-label CaPC.**
> >
> > |Dataset   |Number of Models|State      |Privacy noise $\sigma$|Data-Dependent Epsilon|Sanitized Epsilon| # of queries Binary PATE DD|# of queries Binary PATE DI|# of queries for L2-norm $\tau$ clipping|$\tau$-clipping norm|Accuracy (ACC)      |Balanced Accuracy (BAC)|Area-Under-the-Curve (AUC)|mean Average Precision (mAP)|
> > |----------|----------------|-----------|----------------------|----------------------|-----------------|--------------------------------|--------------------------------|-----------------------------------------|--------------------|--------------------|-----------------------|--------------------------|----------------------------|
> > |Pascal VOC|1               |Initial    |-                     |-                     |-                |-                               |-                               |-                                        |-                   |0.97                |0.85                   |0.97                      |0.85                        |
> > |          |50              |Before CaPC|-                     |-                     |-                |-                               |-                               |-                                        |-                   |.93$\pm$.02         |.59$\pm$.01            |.88$\pm$.01               |.54$\pm$.01                 |
> > |          |50              |After CaPC |9                     |10                    |12.97            |126                             |6                               |50                                       |1.8                 |.94$\pm$.01|.62$\pm$.01   |.88$\pm$.01               |.54$\pm$.01                 |
> > |          |50              |After CaPC |9                     |20                    |26.00            |427                             |19                              |123                                      |1.8                 |.94$\pm$.01|.64$\pm$.01   |.89$\pm$.01      |.55$\pm$.01        |
> > |CheXpert  |1               |Initial    |-                     |-                     |-                |-                               |-                               |-                                        |-                   |0.79                |0.78                   |0.86                      |0.72                        |
> > |          |50              |Before CaPC|-                     |-                     |-                |-                               |-                               |-                                        |-                   |.77$\pm$.06         |.66$\pm$.02            |.75$\pm$.02               |.58$\pm$.02                 |
> > |          |50              |After CaPC |7                     |20                    |25.80            |52                              |19                              |26                                       |2.8                 |.76$\pm$.07         |.69$\pm$.01   |.77$\pm$.01      |.59$\pm$.01        |
> > |MIMIC     |1               |Initial    |-                     |-                     |-                |-                               |-                               |-                                        |-                   |0.9                 |0.74                   |0.84                      |0.51                        |
> > |          |50              |Before CaPC|-                     |-                     |-                |-                               |-                               |-                                        |-                   |.84$\pm$.07         |.63$\pm$.03            |.78$\pm$.03               |.43$\pm$.02                 |
> > |          |50              |After CaPC |10                    |20                    |25.80            |94                              |39                              |48                                       |3                   |.85$\pm$.05|.64$\pm$.01   |.79$\pm$.01      |.45$\pm$.03        |
> > |PadChest  |1               |Initial    |-                     |-                     |-                |-                               |-                               |-                                        |-                   |0.86                |0.79                   |0.9                       |0.37                        |
> > |          |10              |Before CaPC|-                     |-                     |-                |-                               |-                               |-                                        |-                   |.90$\pm$.01         |.64$\pm$.01            |.79$\pm$.01               |.16$\pm$.01                 |
> > |          |10              |After CaPC |7                     |20                    |25.80            |19                              |19                              |29                                       |2.7                 |.88$\pm$.01         |.64$\pm$.01            |.75$\pm$.01               |.14$\pm$.01                 |

---

> > > ### Author Response · Authors · 2021-11-25
> > > **Nomenclature for Table 1**
> > >
> > > The Binary PATE DD (Data Dependent) uses the standard binary PATE per label with the data-dependent analysis. The Binary PATE DI (Data Independent) uses the data-independent bound for each binary classification per label. The L2-norm clipping clips the ballots according to the specified $\tau$ value. # of queries represents the number of answered queries for a given method.

---

> ### Author Response · Authors · 2021-11-25
> **The data-dependent analysis is: (1) an improvement over the propose-test-release, and (2) uses the worst-case sensitivity, which is different from Bayesian or average-case DP.**
>
> >**There are many ways to make a "data-dependent mechanism" be provably data-independently differentially private. As long as you can prove your end-to-end algorithm is eps-10 differentially private for any input dataset, then you can make the promise. But it is not here in the current version. For example, you could spend some budget checking the dataset first. If it is good, run your mechanism. If it is bad, output fail. And this is exactly propose-test-release. The methods I mentioned below are all end-to-end data-independently differentially private. You could run your algorithm on all the possible input dataset S and find the largest one, i.e. $max_S \varepsilon(S)$. This would also satisfy data-independent DP. The point is you have to prove that your algorithm is data-independently DP. Alice as a privacy engineer cannot make any assumptions about the nature that generates the dataset. There is a nice blog post https://differentialprivacy.org/average-case-dp/. It has a discussion about the pitfalls of Bayesian DP, which is (in my opinion) similar to data-dependent DP (data-dependent DP: $\varepsilon(S)$, Bayesian DP: $E_{S \sim P}[\varepsilon(S)]$, data-independent DP: $max_S \varepsilon(S)$). I believe the proposed algorithm in this paper can be proved to be data-independently private even without any change to the algorithm itself. You might want to prove the data-independent RDP of Confident GNMax. (Theorem B.2 is data-dependent). Another caveat I forgot to mention is about comparisons with other works. The gap between data-dependent DP and data-independent DP can be huge (also reported in this paper). It is really unfair to compare with data-independent DP methods like DP-SGD.**
>
> We thank the reviewer for the update and appreciate the discussion.
>
> Our three main points are:
> 1. The data-dependent analysis is an improvement over the propose-test-release, as described in the Dwork & Roth book [2] (Chapter 13).
> 2. The data-dependent analysis uses the worst-case sensitivity, which is different from Bayesian or average-case DP.
> 3. Both our data-dependent and $\ell_2$ data-independent analysis outperform DPSGD for the same $\varepsilon$, including costs required for sanitizing the data-dependent privacy cost.
>
> We respond to specific concerns individually below:
>
> >**Validity of data-dependent DP.**
>
> In response to the reviewer’s comments on propose-test-release:
>
> >*“For example, you could spend some budget checking the dataset first. If it is good, run your mechanism. If it is bad, output fail. And this is exactly propose-test-release.”*
>
> We reference the relationship between data-dependent DP and this mechanism from the privacy book [1] (Chapter 13 - Reflections):
>
> “*Quit When You are NOT Ahead*. This is the philosophy behind Propose-Test-Release, in which we test in a privacy-preserving way that small noise is sufficient for a particularly intended computation on the given data set.
>
> *Algorithms with Data-Dependent Accuracy Bounds*. This can be viewed as a generalization of Quit When You are Not Ahead. Algorithms with data-dependent accuracy bounds can deliver excellent results on “good” data sets, as in Propose-Test-Release, and the accuracy can degrade gradually as the “goodness” decreases, an improvement over Propose-Test-Release.”
>
> >**There is a nice blog post https://differentialprivacy.org/average-case-dp/. It has a discussion about the pitfalls of Bayesian DP, which is (in my opinion) similar to data-dependent DP (data-dependent DP: $\varepsilon(S)$, Bayesian DP: $E_{S \sim P}[\varepsilon(S)]$, data-independent DP: $max_S \varepsilon(S)$).**
>
> The data-dependent DP bounds from PATE [2] are different from Bayesian DP. The average-case methods calibrate noise to the average sensitivity rather than the worst-case sensitivity. These methods also model a prior which changes the trust model and assumptions of the adversary. Rather, data-dependent analysis is tighter accounting of the privacy budget by worst-case bounding the probabilities of other outcomes on the empirically observed data. Rather than being average-case and potentially not protecting outliers, the data-dependent method only aims to provide tighter analysis when it can, and falls back to the data-independent guarantees gracefully on outlier points.
>
> **References:**
>
> [1] [The algorithmic foundations of differential privacy.](https://www.cis.upenn.edu/~aaroth/Papers/privacybook.pdf) Dwork Cynthia, and Aaron Roth. Found. Trends Theor. Comput. Sci. 9.3-4 (2014): 211-407.
>
> [2] [Scalable Private Learning with PATE.](https://openreview.net/forum?id=rkZB1XbRZ) Nicolas Papernot, Shuang Song, Ilya Mironov, Ananth Raghunathan, Kunal Talwar, Úlfar Erlingsson. ICLR 2018.

---

> > ### Author Response · Authors · 2021-11-25
> > **Summary**
> >
> > We hope that our discussion and the paper can help facilitate a better understanding of the differences between data-dependent and data-independent analysis as well as enable a clear representation of the performance under either guarantee. It is important and interesting for future work to explore other data-independent DP multi-winner election mechanisms that can outperform DP-SGD or our proposed mechanisms.

---

> ### Author Response · Authors · 2021-11-28
> **Differential privacy guarantees & Main contributions**
>
> Thank you for your updated review.
>
> Our algorithm is end-to-end differentially private with both data-dependent and data-independent privacy guarantees. We also note that we do not need to fix our analysis as it provides correct guarantees as is.
>
> In particular:
>
> 1. Differential privacy guarantees:
>
> - (I, V) The data-dependent bounds we use are tight but importantly not underestimated. When sanitized, they release no additional private information. We provide details on how to sanitize the data-dependent bounds taken from [2] and do so ourselves.
>
> - (II) We first prove data-independent privacy bounds and then provide data-dependent bounds that work well when there is high consensus, i.e., a “good” dataset. We agree that the passage from Dwork and Roth [1] is about data-dependent accuracy (not privacy) bounds.
>
> - (III, IV) The released labels are differentially private where the formal guarantee can be quoted using either the data-dependent bounds (used from [2]) or our new data-independent bounds. Note that we proposed novel mechanisms to enable direct usage of the bounds from [2].
>
> 2. Main contributions.
> We propose novel mechanisms to extend PATE to the multi-label regime. We prove the optimality of our Binary PATE mechanism in the coordinate-dependent setting. Despite the limited settings, we also compare the performance tradeoffs with Powerset PATE to give both analytical and empirical evidence of its superior performance in many realistic settings. We also show that our approaches outperform DPSGD (and even adaptive DPSGD) using either of our data-dependent or data-independent privacy guarantees. This is an important step toward practical differentially private multi-label training. We also enable multi-label collaborative learning in the multi-site scenario by integrating our mechanisms into CaPC.
>
> [1] [The algorithmic foundations of differential privacy.](https://www.cis.upenn.edu/~aaroth/Papers/privacybook.pdf) Dwork, Cynthia, and Aaron Roth. Found. Trends Theor. Comput. Sci. 9.3-4 (2014): 211-407.
>
> [2] [Scalable Private Learning with PATE.](https://openreview.net/forum?id=rkZB1XbRZ) Nicolas Papernot, Shuang Song, Ilya Mironov, Ananth Raghunathan, Kunal Talwar, Úlfar Erlingsson. ICLR 2018.

---

### Official Review · Reviewer_Fb19 · 2021-11-02

**Correctness:** 3
**Technical Novelty And Significance:** 3
**Empirical Novelty And Significance:** 2
**Recommendation:** 5
**Confidence:** 2

**Main Review:**

This paper introduce mechanisms for private multi-winner voting and multi-label learning, which in important in machine learning. The paper structure and the writing are good. These three mechanisms are shown clear, but for the experiment, the choose of privacy term epsilon is somewhat larger than the common privacy requirements.

**Summary Of The Paper:**

Private multi-winner voting is the task of revealing k-hot binary vectors that satisfy a bounded differential privacy guarantee. They propose three new mechanisms. 1. Binary voting operates independently per label through composition. 2. \tau voting bounds votes optimally in their l2 norm.  3. Powerset voting operates over the entire binary vector by viewing the possible outcomes as a power set. They prove that Powerset voting requires strong correlations between labels to outperform Binary voting.
They also use these mechanisms to enable privacy-preserving multi-label learning. They empirically compare techniques with DPSGD on large real-world healthcare data and standard multi-label benchmarks. Their techniques outperform all others in the centralized setting, and show that mechanisms can be used to collaboratively improve models in a multi-site (distributed) setting.


**Summary Of The Review:**

It's good but may be not enough

---

> ### Author Response · Authors · 2021-11-17
> **Privacy term $\varepsilon$**
>
> We thank the reviewer for the comments and provide a detailed answer below.
>
> >**For the experiments, the choice of privacy term $\varepsilon$ is somewhat larger than the common privacy requirements.**
>
> Our choice of $\varepsilon$ is $\le 20$ which falls within the range that is generally considered in prior work and is deemed acceptable (though loose). We also presented experiments with $\varepsilon \le 10$ and showed that in this regime, our approach outperformed DPSGD as well. For instance, our results on the Pascal VOC dataset in Table 1 use an $\varepsilon=10$. Further, our results for the first 5 labels of CheXpert in Appendix I.2 use $\varepsilon=8$. These values are similar to those chosen in the original work of PATE and CaPC and though loose, have been found to be robust to privacy attacks [1, 2]. We note that because we deal with multi-label scenarios, rather than the typical multi-class, it is more difficult to attain tighter DP guarantees. We explained our choices of privacy budget in the Ethics Statement (Section 7), which we also include below in this response.
>
> “It is not well understood what the value of $\varepsilon$ should be and therefore improper usage and large values of $\varepsilon$-s could leak private information. This risk can be mitigated by setting an upper threshold for the possible values of $\varepsilon$ which can be used. The values of $\varepsilon$-s used in our experiments: 8,10, and 20, could be considered high. However, since our work provides multi-label classification, there are many labels which need to be released and therefore the privacy budget will naturally be higher than for single label classification. Our common choice of $\varepsilon=20$ is empirical. We select a value, which although is not tight, is a reasonable bound for our multi-label setting because it is in the range of values considered by most empirical evaluations (of the order 10).”
>
> To improve our exposition, we include experiments showing how our mechanism performs as we vary the privacy budget $\varepsilon$ from a tight guarantee to looser guarantees. In the table below, we present detailed analysis of the Binary PATE performance on the Pascal VOC dataset when selecting the privacy budget $\varepsilon$ in the range from 1 to 20 (we added the results to section I.9 in Appendix). The number of answered queries gradually increases with the higher values of $\varepsilon$.
>
> | $\varepsilon$ | # of Answered Queries | ACC | BAC | AUC | MAP |
> |----|--------------|-----|-----|-----|-----|
> | 1  | 0     | -   | -   | -   | -   |
> | 2  | 6   | .86 | .62 | .62 | .44 |
> | 3  | 13  | .93 | .67 | .67 | .53 |
> | 4  | 22  | .93 | .64 | .64 | .44 |
> | 5  | 31  | .95 | .63 | .63 | .39 |
> | 6  | 40  | .95 | .67 | .67 | .45 |
> | 7  | 64  | .95 | .64 | .64 | .35 |
> | 8  | 81  | .95 | .66 | .66 | .40 |
> | 9  | 101 | .95 | .60 | .60 | .28 |
> | 10 | 113 | .96 | .63 | .63 | .30 |
> | 11 | 135 | .96 | .64 | .64 | .33 |
> | 12 | 165 | .96 | .65 | .65 | .35 |
> | 13 | 199 | .96 | .63 | .63 | .32 |
> | 14 | 217 | .96 | .64 | .64 | .35 |
> | 15 | 239 | .96 | .63 | .63 | .32 |
> | 16 | 272 | .96 | .63 | .63 | .31 |
> | 17 | 306 | .96 | .63 | .63 | .30 |
> | 18 | 332 | .96 | .63 | .63 | .31 |
> | 19 | 362 | .96 | .63 | .63 | .30 |
> | 20 | 403 | .96 | .63 | .63 | .30 |
>
> **References:**
>
> [1] [Label-only membership inference attacks.](https://proceedings.mlr.press/v139/choquette-choo21a.html) Christopher A. Choquette-Choo, Florian Tramer, Nicholas Carlini, Nicolas Papernot. ICML 2021.
>
> [2] [Adversary instantiation: Lower bounds for differentially private machine learning.](https://arxiv.org/abs/2101.04535) Milad Nasr, Shuang Song, Abhradeep Thakurta, Nicolas Papernot, Nicholas Carlini. IEEE Symposium on
> Security and Privacy 2021.

---

> > ### Author Response · Authors · 2021-11-22
> > **Have concerns been addressed?**
> >
> > We would like to follow up on our answers, especially on the new experimental results that we provided. Do our replies adequately address the reviewer's concerns regarding the choice of the privacy term $\varepsilon$?

---

### Official Review · Reviewer_jFrr · 2021-11-03

**Correctness:** 4
**Technical Novelty And Significance:** 3
**Empirical Novelty And Significance:** 2
**Recommendation:** 5
**Confidence:** 2

**Main Review:**

Differentially private multi-label classification is an interesting and well-motivated problem.

The Binary voting mechanism is quite naive in the sense that separating the problem into k separate elections, each to be evaluated in a differentially private manner (i.e., with Gaussian noise). It’s somewhat surprising that this seems to be optimal among the mechanisms (in the case where voters aren’t constrained in how many candidates they can approve) unless there is a lot of correlation between votes, at which point the (exponentially-sized?) powerset voting method becomes better.

Is it correct to interpret the thresholds T in Definitions 4 and 6 as the cutoff point at which the sum term is at least n/2 (intuitively corresponding to a majority vote)? It could be useful to the reader to have a bit more explanation here.

To me, the most interesting contribution is that of \tau voting, where all votes are assumed to have \ell_2 norm at most \tau. I have a few questions here: (1) how did you go about choosing \tau in practice, and do you have proposals for how to choose it on new datasets; and (2) have you thought about an “average-case” approach where the average vote has \ell_2 norm at most \tau instead of requiring all votes to have \ell_2 norm at most \tau?

I found the writing quality to have some issues (some minor comments below).

Minor comments:
Define CaPC
Section 2.2, CaPC paragraph: model's --> models
Section 3.2, first paragraph: it is a popular method --> is a popular method
Before Definition 6: missing period
There are also many missing articles (a/an/the) throughout the paper

**Summary Of The Paper:**

The authors study differentially private multi-winner voting, which is designed for multi-label learning subject to a privacy constraint meant to limit information leakage about training data to an adversary. They propose three mechanisms: Binary, which essentially runs an existing differentially-private election for each label independently, \tau voting, which works with votes that have bounded \ell_2 norm, and powerset voting, which explicitly encodes each possible subset of winners as an alternative in an election and then votes over them. They show that Binary voting (the naive approach) generally outperforms powerset voting as long as there aren’t strong correlations between votes. Lastly, they show that they can use these multi-winner DP techniques to a extend single-label technique, PATE, and empirically demonstrate the effectiveness of their approach.

**Summary Of The Review:**

The problem is interesting and the results seem to be quite comprehensive. My main concern is that the voting mechanisms proposed are relatively naive (separate into independent elections or run one big single-winner election). However, the technical results seem robust.

---

> ### Author Response · Authors · 2021-11-17
> **Main concern: relatively naive voting mechanisms**
>
> >**My main concern is that the voting mechanisms proposed are relatively naive (separate into independent elections or run one big single-winner election).**
>
> We remark that the proposed methods work very well in practice and are simple extensions to the multi-winner election. The mechanisms are relatively naive but their analysis is non-trivial. Our work provides an important step into enabling differentially private mechanisms which have immediate applications to multi-label machine learning. Through empirical results, we show tradeoffs in these approaches and demonstrate significant improvement over the DPSGD baseline, which can be immediately applied to this setting. Importantly, the problem of private and collaborative multi-label classification has no prior work.
>
> We also tried more complicated methods, for example, based on the bounded noise mechanism, however, they do not work in practice. Recently, [Dagan & Kur, 2020](https://arxiv.org/abs/2012.03817) presented an $(\varepsilon,\delta)$-DP mechanism with optimal $\ell_{\infty}$ error for most values of $\delta$. The algorithm adds independent noise from a distribution with bounded support to each of the coordinates. For the $k$-label classification problem, instead of aggregating $k$ separate outputs by PATE, we used a counting-query mechanism with the bounded noise to average over the full prediction vector output by each teacher. We found that this method is sound in theory but cannot be applied in practice because its assumptions do not hold in real scenarios.

---

> ### Author Response · Authors · 2021-11-17
> **Binary vs Powerset, Threshold $T$, Choosing $\tau$, Typos**
>
> Thank you for your detailed review. We provide in-line answers below:
>
> >**Unless there is a lot of correlation between votes, at which point the (exponentially-sized?) powerset voting method becomes better.**
>
> Indeed, the number of classes for the Powerset voting is $2^k$, thus it grows exponentially with the number of labels $k$.
>
> Our optimality proofs show that a binary voting mechanism is the best mechanism unless there is a correlation between candidates. To understand how much correlation is required for other methods, e.g., our power voting to outperform binary voting, we perform both a theoretical and empirical analysis. We find that leveraging the data-dependent bounds of PATE (for our binary PATE) gives tighter privacy loss than powerset PATE under the same consensus. However, because powerset PATE releases only a single outcome per query (where binary PATE releases k for k candidates), it can sometimes outperform binary PATE. Our small-scale results in Appendix B.4 demonstrate these scenarios and our large-scale results show that this may not often happen in practice (though it does sometimes) as our binary voting mechanisms outperform powerset voting in the majority of cases.
>
> >**Is it correct to interpret the thresholds T in Definitions 4 and 6 as the cutoff point at which the sum term is at least n/2 (intuitively corresponding to a majority vote)? It could be useful to the reader to have a bit more explanation here.**
>
> Regarding Definition 4, since we consider binary classification per label $i$, $\sum_{j=1}^{n} b_j[i]$ represents the number of positive votes, and $T_i = n-\sum_{j=1}^{n} b_j[i]$ is the number of negative votes. In Definitions 4 and 7, we added the subscript $i$ to $T$ to indicate that this threshold changes per label $i$. In the case of the standard binary voting, as in Definition 3 (without noise), the outcome of the comparison between the number of positive votes and the number of negative votes is equivalent to the result of comparing the number of positive votes against the threshold $T = \frac{n}{2}$. However, the optimal threshold $T$ changes in the case of the binary voting with privacy, where we add the noise. For example, if we compare the number of positive votes with $T$ being the number of negative votes, then we can add more noise (increase $\sigma_G$) when compared to the threshold $T=\frac{n}{2}$. Similarly, in Definition 7, we select: $T_i = n-\sum_{j=1}^{n} v_j[i]$, which represents the number of *clipped* negative votes. We added the clarification in the updated manuscript in Section 3.1.
>
> >**(1) How did you go about choosing $\tau$ in practice, and do you have proposals for how to choose it on new datasets?**
>
> We pick $\tau$ to be marginally larger than the average number of positive labels in the training data, which we added to our explanation in Section 3.2. More sophisticated methods, like adaptively picking $\tau$ may also be possible.
>
> >**Have you thought about an "*average-case*" approach where the average vote has $\ell_2$ norm at most $\tau$ instead of requiring all votes to have $\ell_2$ norm at most $\tau$.**
>
> Relaxing the $\tau$ approach so that not all votes have their $\ell_2$ norm of at most $\tau$ is non-trivial to leverage. Our approach enforces that the $\ell_2$ norm is at most $\tau$ for all samples so as to restrict the global sensitivity. Not enforcing this for all samples increases the global sensitivity and thus the privacy loss. In short, either all votes must be clipped or there is no benefit (in terms of achieving tighter DP privacy loss).
>
> >**Minor comments: Define CaPC Section 2.2, CaPC paragraph: model's --> models Section 3.2, first paragraph: it is a popular method --> is a popular method Before Definition 6: missing period There are also many missing articles (a/an/the) throughout the paper.**
>
> Thank you for the comments, we corrected these mistakes and did a thorough pass to catch all others.

---

> ### Author Response · Authors · 2021-11-22
> **Have concerns been addressed?**
>
> We would like to follow up on our answers, especially regarding the voting mechanisms as well as the choices of $\tau$ and $T$. Do our replies adequately address the reviewer's concerns?

---

### Official Review · Reviewer_47xz · 2021-11-06

**Correctness:** 3
**Technical Novelty And Significance:** 3
**Empirical Novelty And Significance:** 3
**Recommendation:** 8
**Confidence:** 3

**Main Review:**

Strengths:
- The authors consider an interesting problem as designing a privacy preserving mechanism with multi-label outcomes would be applicable in many contexts.
- I also like the design choices as the proposed mechanisms can be easily implemented with existing DP aggregation mechanisms like PATE. Moreover, the theoretical bounds can be derived by building on top of the existing DP guarantees.

Weaknesses:
- As far as I understand, the parameter $\tau$ in the $\tau$-voting needs to be set based on the domain and there is no automatic way to choose a value.
- Powerset voting becomes intractable to implement when $k$ is large. But probably there is a way to combine $\tau$-voting and the powerset voting for large values of $k$.

Some questions for the authors.
1. From definition 3, it seems like the threshold T is independent of the candidate $i$. However, in definition 4, the constant $T$ depends on the index $i$ and it seems that the mechanism picks an index whose votes exceed $n/2$ (upto noise). So I think the presentation and the definition here is misleading.
2. How do you define the threshold $T$ in definition 7?
3. I think there is a range of design options between $\tau$-voting and powerset voting. For example, you can ask each voter to vote on the best subset of size b. Did the authors consider other alternatives?



**Summary Of The Paper:**

This paper considers the design of differentially private multi-label mechanisms. In particular, the authors employ multi-winner voting protocols to existing differentially private single-label learning algorithms e.g. PATE. They consider three multi-label voting protocols -- binary voting, $\tau$-voting and powerset voting.

Binary voting works by independently applying majority voting on each coordinate. It is obvious that such an aggregation mechanism has does not provide a better privacy guarantee than $k$ applications of binary voting. The privacy guarantee can be improved when the coordinates are dependent. This motivates the authors to consider $\tau$-voting where the $\ell_2$-norm of each ballot is bounded by $\tau$. Finally, in the powerset voting, the voting is done over the universe of all subsets of the alternatives i.e. $2^k$ alternatives.

The authors make two important observations through experiments. First, when there is high consensus and $\sigma_G \rightarrow 0$, binary voting performs best. On the other hand, $\tau$-voting outperforms with lower consensus and larger values of $\sigma_G$. Second, even in centralized setting, multi-label methods outperform existing benchmarks.

**Summary Of The Review:**

I think the paper considers an important problem as the design of differentially private aggregation mechanism for multi-label outcomes is applicable in a lot of domains. The proposed mechanisms are also simple and build on top of the existing DP aggregation mechanisms. However, I felt that the experiments were not quite exhaustive. For example, it's not clear if it is beneficial to apply powerset voting on real datasets.

---

> ### Author Response · Authors · 2021-11-17
> **Selecting $\tau$, Powerset with $\tau$, Threshold $T$**
>
> Thank you for your positive and constructive feedback. We agree that there are two important takeaways: first is how consensus and the noise parameter impact the best voting mechanism and second is that this mechanism outperforms the DPSGD baselines.
>
> We provide detailed answers inline:
>
> >**As far as I understand, the parameter $\tau$ in the $\tau$-voting needs to be set based on the domain and there is no automatic way to choose a value.**
>
> Though the $\tau$ parameter does require tuning and cannot be automatically set, determining a suitable value is largely similar to the methodology for determining a clipping value in DPSGD, which we show we outperform in our experiments. We select $\tau$ to be marginally larger than the average p-norm of labels in the training data (in the case of the $\ell_1$ norm, it corresponds to the average number of positive labels), which we added to our explanation in Section 3.2.
>
> >**Powerset voting becomes intractable to implement when $k$ is large. But probably there is a way to combine $\tau$-voting and the powerset voting for large values of $k$.**
>
> Indeed, the powerset voting mechanism performance can significantly deteriorate when the number of labels $k$ is large. This is because PATE requires us to add noise to the votes for all possible classes. The number of classes in the Powerset method is $2^k$, so it grows exponentially with $k$ and causes a performance bottleneck.
>
> Overall, it is not beneficial to apply the $\tau$-clipping to Powerset in the same way as it was done for the Binary voting. In the Powerset method, a binary vector represents a single class. The application of the $\tau$-clipping to the Powerset method may in fact introduce new classes, because coordinates may now be represented by any float rather than solely integers, decreasing the consensus. However, we agree with the reviewer that the 2nd approach proposed is promising, as explained below.
>
> >**Question 1: From Definition 3, it seems like the threshold $T$ is independent of the candidate $i$. However, in Definition 4, the constant $T$ depends on the index $i$ and it seems that the mechanism picks an index whose votes exceed $\frac{n}{2}$ (up to noise). So I think the presentation and the definition here are misleading.**
>
> Regarding Definition 4, since we consider binary classification per label $i$, $\sum_{j=1}^{n} b_j[i]$ represents the number of positive votes, and $T_i = n-\sum_{j=1}^{n} b_j[i]$ is the number of negative votes. In Definitions 4 and 7, we added the subscript $i$ to $T$ to indicate that this threshold changes per label $i$. In the case of the standard binary voting, as in Definition 3 (without noise), the outcome of the comparison between the number of positive votes and the number of negative votes is equivalent to the result of comparing the number of positive votes against the threshold $T = \frac{n}{2}$. However, the optimal threshold $T$ changes in the case of the binary voting with privacy, where we add the noise. For example, if we compare the number of positive votes with $T$ being the number of negative votes, then we can add more noise (increase $\sigma_G$) when compared to the threshold $T=\frac{n}{2}$. Similarly, in Definition 7, we select: $T_i = n-\sum_{j=1}^{n} v_j[i]$, which represents the number of *clipped* negative votes. We added the clarification in the updated manuscript in Section 3.1.
>
> >**Question 2: How do you define the threshold in Definition 7?**
>
> The threshold $T$ in Definition 7 should be defined as the noisy number of votes for the runner-up class (the class with the second-highest number of votes). Because in the Powerset method, we reduce the multi-label classification to the single-label classification, this can be simplified to taking the noisy argmax of the vote counts. We removed $T$ and added argmax to Definition 7 in the updated version of the manuscript.

---

> > ### Author Response · Authors · 2021-11-17
> > **Powerset with max $\tau$ positive votes per ballot**
> >
> > >**Question 3: I think there is a range of design options between $\tau$-voting and the Powerset voting. For example, you can ask each voter to vote on the best subset of size $b$. Did the authors consider other alternatives?**
> >
> > We present how to incorporate this $\tau$-voting into the Powerset method. A given teacher is allowed to set up to $\tau$ positive labels for a given query sample. This approach limits the number of positive classes from $2^{k}$ to ${{k}\choose{0}} + {{k}\choose{1}} + \dots + {{k}\choose{\tau}}$ classes. Since we have fewer classes: (1) it increases the possibility of achieving a consensus between teachers (a higher gap between max and runner-up numbers of votes can be achieved), (2) the data independent bound is lower (computed as $1 - (1 / C)$, where $C$ is a number of classes), and (3) the method has higher performance (faster runtime). On the other hand, it does not lower the sensitivity of the Powerset mechanism, since changing one of the teachers could decrease the vote from one class and increase it in the other. Thus, the sensitivity remains 2.
> >
> > To implement the above-proposed Powerset method, it is pertinent to pick which subset of $\tau$ positive labels will be retained and which are discarded (set to negative or 0). Intuitively, it is possible to pick the top $\tau$ labels with the highest confidence from the model. However, most deep multi-label models use independent predictive heads, e.g., a separate sigmoid activation per output label, which may make naive comparisons of these values degrade performance. Thus, there is no clear notion of what the desired subset is, and deciding it may require modifications to the architecture of the model.
> >
> > For example, for the Pascal VOC dataset, the same threshold of $0.5$ probability is used per label, so the confidence values are comparable. However, for the CheXpert dataset, different probability thresholds are set per label, so the problem of deciding which of the labels should remain positive (if there are more positive labels than the max $\tau$ of positive labels) is difficult to resolve. This would likely require some form of domain knowledge.
> >
> > We implement this $\tau$ Powerset method on the Pascal VOC dataset. For the original train set of the Pascal VOC dataset (with 5823 samples), the average number of positive labels per example is 1.52, with 20 total labels. More detailed statistics are given below:
> >
> > |Number of positive labels|Number of train samples|
> > |-------------------------|-----------------------|
> > |6                        |1                      |
> > |5                        |17                     |
> > |4                        |105                    |
> > |3                        |484                    |
> > |2                        |1677                   |
> > |1                        |3539                   |
> > |0                        |0                      |
> >
> > The table below presents the results of Powerset with clipping on the Pascal VOC dataset. The performance of the $\tau$ clipping Powerset PATE mechanism is measured using the number of answered queries (at a specified $\sigma_{\text{GNMax}}$) and the ACC, BAC, AUC, and MAP metrics as measured on the answered queries from the test set. PB ($\varepsilon$) is the privacy budget. We use 50 teacher models.  We vary the cardinality of the targets (i.e., the number of labels considered) from $k=1$ to $20$ which may impact the performance of the mechanism under a fixed $\tau$.
> >
> > We find that $\tau$-voting enables more queries to be answered while maintaining similar performance metrics (e.g., accuracy). However, the number of answered queries using the Binary method is still higher than with the Powerset method even when $\tau=1$. We show the detailed results below and added the above answer to the manuscript as Section I.2 in the Appendix.

---

> > > ### Author Response · Authors · 2021-11-17
> > > **Results for $\tau$ Powerset**
> > >
> > > |# Labels|$\tau$|# Queries|PB($\varepsilon$)|$\sigma_{GNMax}$|ACC |BAC |AUC |MAP |
> > > |--------|------|---------|-----------------|----------------|----|----|----|----|
> > > |1       |1     |5464     |20               |2               |0.98|0.84|0.84|0.75|
> > > |2       |1     |5464     |20               |5               |0.97|0.75|0.75|0.55|
> > > |2       |2     |5464     |20               |5               |0.97|0.74|0.74|0.55|
> > > |3       |1     |3437     |20               |7               |0.97|0.71|0.71|0.48|
> > > |3       |2     |3421     |20               |7               |0.97|0.71|0.71|0.49|
> > > |3       |3     |3416     |20               |7               |0.97|0.72|0.72|0.51|
> > > |4       |1     |2547     |20               |7               |0.97|0.68|0.68|0.43|
> > > |4       |2     |2527     |20               |7               |0.97|0.69|0.69|0.43|
> > > |4       |3     |2485     |20               |7               |0.97|0.69|0.69|0.43|
> > > |4       |4     |2485     |20               |7               |0.97|0.69|0.69|0.43|
> > > |5       |1     |2321     |20               |8               |0.96|0.65|0.65|0.36|
> > > |5       |2     |2151     |20               |8               |0.96|0.65|0.65|0.34|
> > > |5       |3     |2077     |20               |8               |0.96|0.65|0.65|0.34|
> > > |5       |4     |2048     |20               |8               |0.96|0.65|0.65|0.33|
> > > |5       |5     |2040     |20               |8               |0.96|0.65|0.65|0.33|
> > > |8       |8     |1192     |20               |7               |0.95|0.68|0.68|0.37|
> > > |10      |1     |947      |20               |5               |0.97|0.65|0.65|0.36|
> > > |10      |2     |896      |20               |5               |0.96|0.65|0.65|0.36|
> > > |10      |3     |895      |20               |5               |0.96|0.65|0.65|0.33|
> > > |10      |4     |885      |20               |5               |0.96|0.65|0.65|0.33|
> > > |10      |5     |877      |20               |5               |0.96|0.65|0.65|0.33|
> > > |10      |6     |861      |20               |5               |0.96|0.64|0.64|0.3 |
> > > |10      |7     |849      |20               |5               |0.96|0.64|0.64|0.29|
> > > |10      |8     |848      |20               |5               |0.96|0.66|0.66|0.34|
> > > |10      |9     |848      |20               |5               |0.96|0.64|0.64|0.32|
> > > |10      |10    |848      |20               |5               |0.96|0.64|0.64|0.3 |
> > > |11      |11    |702      |20               |4               |0.95|0.66|0.66|0.37|
> > > |14      |14    |417      |20               |3               |0.95|0.65|0.65|0.37|
> > > |15      |15    |165      |20               |3               |0.94|0.66|0.66|0.36|
> > > |16      |16    |94       |20               |2               |0.95|0.65|0.65|0.38|
> > > |18      |18    |94       |20               |2               |0.95|0.65|0.65|0.36|
> > > |20      |1     |111      |20               |2               |0.95|0.63|0.63|0.33|
> > > |20      |2     |99       |20               |2               |0.96|0.67|0.67|0.41|
> > > |20      |3     |81       |20               |2               |0.96|0.61|0.61|0.32|
> > > |20      |4     |78       |20               |2               |0.96|0.63|0.63|0.33|
> > > |20      |5     |78       |20               |2               |0.96|0.63|0.63|0.34|
> > > |20      |10    |78       |20               |2               |0.95|0.65|0.65|0.34|
> > > |20      |20    |78       |20               |2               |0.95|0.65|0.65|0.33|
> > >
> > > When the number of labels is set to $k=5$, for $\tau=5$, which is equivalent to no clipping, there are 32 classes. For $k=5, \tau=2$ the number of classes decreases to 16 and we are able to answer 5\% queries more than when no-clipping is applied ($\tau=5$). For $\tau=1$, we have only 6 classes and almost 14\% more answered queries. For $\tau=5$ there is only $1$ more class than for $\tau=4$ and we observe a very small difference in the number of answered queries (namely 8).

---

> > ### Author Response · Authors · 2021-11-17
> > **Powerset voting on real datasets**
> >
> > >**From the summary of the review: However, I felt that the experiments were not quite exhaustive. For example, it's not clear if it is beneficial to apply powerset voting on real datasets.**
> >
> > We present results for the Powerset voting in Table 2. An exhaustive comparison between Binary and Powerset methods on the Pascal VOC and CheXpert datasets can be found in Appendix I.1, where we do find that in some cases Powerset Voting does outperform Binary Voting (see Table 5).
> >
> > For the Pascal VOC dataset, Powerset Voting outperforms DPSGD in terms of the accuracy, balanced accuracy, and AUC (Area Under the Curve) metrics, however, it is worse in the case of the mAP metric. The average precision is low for Powerset since it requires that most teachers correctly predict *all* labels. On the other hand, for Binary voting, even if a teacher makes mistakes for a few labels, this still leads towards the correct final positive labels as long as most labels are predicted correctly. Overall, Binary outperforms both DPSDG and Powerset on the Pascal VOC dataset. For the CheXpert dataset, Powerset and Binary methods are on par and outperform DPSGD.
> >
> > If the reviewer has any particular additional dataset in mind, we would be happy to revise our manuscript and add results for it.

---

> ### Author Response · Authors · 2021-11-23
> **Have concerns been addressed?**
>
> We would like to follow up on our answers, and especially on the new data that we provided for the Powerset voting. Do our replies adequately address the reviewer's concerns?

---

### Official Review · Reviewer_6iWo · 2021-12-04

**Correctness:** 4
**Technical Novelty And Significance:** 3
**Empirical Novelty And Significance:** 2
**Recommendation:** 5
**Confidence:** 3

**Main Review:**

Strengths:
- The paper is relatively clearly written (see also the "Further comments" part though)
- The theoretical results related to privacy are interesting, and are discussed in sufficient depth and details.
- The solutions are clean and reasonably simple.

Weaknesses:
- The MLC learning part is not treated carefully. First of all, the canonical MLC metrics are not calculated in the experiments. (In fact, they are not even discussed in the paper.) This is important though, because this is one of the crucial, distinctive aspect of MLC. Therefore, it is not clear, how the proposed solutions perform in terms of these metrics.
- The proposed algorithms are not compared to the state of art MLC solutions, and thus we don't know how much we lose in terms of the MLC metrics when we want to DP guarantees.

Because of these two shortcomings, it is hard to assess the contribution of the paper. The main goal of the authors was to show how DP can be enforced in MLC but the results presented in the paper are not sufficient to draw any conclusion: although the DP requirements are analyzed, the MLC performance remains unclear.

Further comments:
- The acronym CaPC is used already on p1, but is only explained on p3.
- Why do you use different notations (d resp. X) for the same notion (datasets) in Def. 1 and Def. 2?
- What does p_{M(X)}(\theta) denote in Def. 2?
- Shouldn't Lemma 2.2 be stated with an inequality?
- Def. 4 mixes sets with their membership vectors. Which is fine, but this should be mentioned explicitly.
- Def. 5 is confusing. The meaning of the math formulation is that there is no element appearing in each of the P_i sets, whereas the text means that the P_i sets are all disjoint.
- Below Def. 6: "For any mechanism f, \Delta_p f is maximized when the mechanism’s output for each coordinate i is flipped from predicted to not predicted or vice versa. The pair of teacher votes achieving this differ in each coordinate i." Aren't we supposed to work here with teacher votes that differ only by one bit?

**Summary Of The Paper:**

The paper is proposing differential privacy (DP) solutions for multi-label classification (MLC). In particular, it introduces three novel noisy perturbation methods for the MLC setup, analyzes them in terms of DP, and test them experimentally for accuracy, AUC, and other metrics.

**Summary Of The Review:**

The DP results are solid, but the MLC aspects of the work has serious shortcomings.

---

> ### Author Response · Authors · 2021-12-05
> **Notation and Definitions**
>
> >**The acronym CaPC is used already on p1, but is only explained on p3.**
>
> Thank you for catching this. We will define CaPC as “Confidential and Private Collaborative” learning in the abstract where it is first used. In our other uses of the acronym, we include citations to the CaPC paper which defines the acronym and setup but will additionally include a forward pointer at the first main-text use on p2 (contributions bullet 3) to the background description that is on p3.
>
> >**Why do you use different notations (d resp. X) for the same notion (datasets) in Def. 1 and Def. 2?**
>
> Thank you for pointing this out. We will unify the notation and use X to denote a dataset in the updated manuscript.
>
> >**What does p_{M(X)}(\theta) denote in Def. 2?**
>
> In the original paper by Mironov [1] (Definitions 3 and 4), p is defined as the probability distribution over the possible outcomes of mechanism M and p(\theta) is the density of p at outcome \theta. To improve notation, we will rewrite it as $Pr[M(X) = \theta]$ and follow the form of definitions from [2,3]:
>
> **Definition 2 (Rényi Divergence).** For two probability distributions $P$ and $Q$ defined over $\mathcal{R}$, the Rényi divergence of order $\lambda > 1$ between them is defined as:
>
> $ D_{\lambda}(P || Q) \triangleq \frac{1}{\lambda - 1} \log\mathbb{E}_{\theta \sim P} \left[ \left( \frac{P(\theta)}{Q(\theta)}\right)^{\lambda - 1}\right] $
>
> **Definition 3 (Rényi Differential Privacy).** A randomized mechanism $\mathcal{M}$ is said to satisfy $\varepsilon$-Rényi differential privacy of order $\lambda$, or $(\lambda, \varepsilon)$-RDP for short, if for any adjacent datasets $X, X' \in \mathcal{D}$:
>
> $D_{\lambda}(\mathcal{M}(X) || \mathcal{M}(X')) = \frac{1}{\lambda - 1} \log\mathbb{E}_{\theta \sim \mathcal{M}(X)}\left[\left(\frac{{\rm Pr}[\mathcal{M}(X) = \theta]}{{\rm Pr}[\mathcal{M}(X') = \theta]}\right)^{\lambda - 1}\right] \leq \varepsilon$
>
> >**Shouldn't Lemma 2.2 be stated with an inequality?**
>
> Lemma 2.2 is commonly stated with equality as in the original Proposition 1 from [1] and later Theorem 4 in [2] and Lemma 5 in [4]. We can change the statement to inequality if desired or express it in the following way:
>
> **Lemma 2.2 (RDP-Composition).** If a mechanism $\mathcal{M}$ consists of a sequence of adaptive mechanisms such that $\mathcal{M}_1,\dots,\mathcal{M}_k$ and for any $\mathcal{M}_i$ where $i \in [k]$
>
> we have the guarantee $(\delta,\varepsilon_i)$ -RDP, then $\mathcal{M}$ guarantees $(\delta,\sum_{i=1}^{k} \varepsilon_i)$ -RDP.
>
> >**Def. 4 mixes sets with their membership vectors. Which is fine, but this should be mentioned explicitly.**
>
> Thank you for noting this, we will mention it explicitly in the revised manuscript.
>
> >**Def. 5 is confusing. The meaning of the math formulation is that there is no element appearing in each of the P_i sets, whereas the text means that the P_i sets are all disjoint.**
>
> Thank you, we clarify the mathematical definition to align with the textual definition with the following change: “$\forall i,j$ such that $i \neq j, P_i \cap P_j = \emptyset$”.
>
> >**Below Def. 6: "For any mechanism f, \Delta_p f is maximized when the mechanism’s output for each coordinate i is flipped from predicted to not predicted or vice versa. The pair of teacher votes achieving this differ in each coordinate i." Aren't we supposed to work here with teacher votes that differ only by one bit?**
>
> In the case of the standard single-label classification, teacher votes differ by one bit since the votes are one-hot vectors. In the case of the multi-label classification, in the worst case, changing a single data element in the training set of a teacher can cause a change in prediction where each bit is flipped in the multi-hot vector that represents a teacher vote.
>
> **References:**
>
> [1] [Renyi Differential Privacy.](https://arxiv.org/abs/1702.07476) Ilya Mironov. CSF 2017.
>
> [2] [Scalable Private Learning with PATE.](https://openreview.net/forum?id=rkZB1XbRZ) Nicolas Papernot, Shuang Song, Ilya Mironov, Ananth Raghunathan, Kunal Talwar, Úlfar Erlingsson. ICLR 2018.
>
> [3] [CaPC Learning: Confidential and Private Collaborative Learning](https://openreview.net/forum?id=h2EbJ4_wMVq).
> Christopher A. Choquette-Choo, Natalie Dullerud, Adam Dziedzic, Yunxiang Zhang, Somesh Jha, Nicolas Papernot, Xiao Wang, ICLR 2021.
>
> [4] [Private-kNN: Practical Differential Privacy for Computer Vision.](https://bit.ly/2P0tYad) Yuqing Zhu, Xiang Yu, Manmohan Chandraker, Yu-Xiang Wang. CVPR 2020.

---

> ### Author Response · Authors · 2021-12-05
> **MLC metrics and comparison with state-of-the-art**
>
> Thank you for the review. We appreciate the feedback.
>
> >**First of all, the canonical MLC metrics are not calculated in the experiments. (In fact, they are not even discussed in the paper.) This is important though because this is one of the crucial, distinctive aspects of MLC. Therefore, it is not clear, how the proposed solutions perform in terms of these metrics.**
>
> We have computed 4 metrics: accuracy, BAC, AUC, mAP - which are metrics used to report results for multi-label classification in other papers [1, 2]. The area under the receiver-operating curve (AUC) is a common metric on multi-label medical tasks [1, 4, 5], and mean average precision (mAP) is used frequently, e.g., [2]; we report both. We additionally compute the Balanced Accuracy (BAC) which was used in CaPC [3]. In the code, which is uploaded to this submission, please see file: *utils.py*, method: *compute_metrics_multilabel()*, line: *2758*. We followed the evaluation procedure from [2] and also computed the following metrics: true-positives, false-positives, false-negatives, true-negatives, average per-Class precision (CP), recall (CR), F1 (CF1), and the average Overall precision (OP), recall (OR) and F1 (OF1).
>
> We are happy to include other metrics; are there any that should be added?
>
>
> >**The proposed algorithms are not compared to the state of the art MLC solutions, and thus we don’t know how much we lose in terms of MLC metrics when we want to DP guarantees.**
>
> We discuss in Section 5.2 a comparison against non-private state-of-the-art models and the bottom line is that though we significantly outperform the DPSGD baseline, there is still much room for improvement in matching non-private performance with DP. This is shown in Table 2 and also below for the Pascal VOC dataset:
>
> | Method        | ACC | BAC | AUC | MAP |
> |---------------|-----|-----|-----|-----|
> | Non-private   | .97 | .85 | .97 | .85 |
> | DPSGD         | .92 | .50 | .68 | .40 |
> | Powerset PATE | .94 | .58 | .70 | .29 |
> | Binary PATE   | .94 | .62 | .85 | .57 |
>
> For the Pascal VOC, unfortunately, the state-of-the-art from, e.g., [2] is not open-sourced as explained in this thread on GitHub: https://github.com/Alibaba-MIIL/ASL/issues/1. We did our best to train the teacher models on Pascal VOC for PATE.
>
> For medical datasets, we use similar model architectures (DenseNet121) and training methods as proposed in [1] to obtain comparable results in terms of AUC, which is considered state-of-the-art for the medical tasks we use in our empirical evaluation.
>
> In the below, we show a direct comparison to [1] in terms of single non-private models that we used in our experiments.
>
> | AUC (%)  | Cohen et al. [1] (Figure 2) | Our method (Table 1) |
> |----------|-----------------------------|----------------------|
> | CheXpert | 80                          | 86                   |
> | MIMIC    | 83                          | 84                   |
> | PadChest | 85                          | 90                   |
>
> **References:**
>
> [1] [On the limits of cross-domain generalization in automated X-ray prediction.](http://proceedings.mlr.press/v121/cohen20a/cohen20a.pdf) Joseph Paul Cohen, Mohammad Hashir, Rupert Brooks, Hadrien Bertrand. MIDL 2020.
>
> [2] [Asymmetric Loss For Multi-Label Classification.](https://openaccess.thecvf.com/content/ICCV2021/papers/Ridnik_Asymmetric_Loss_for_Multi-Label_Classification_ICCV_2021_paper.pdf) Emanuel Ben-Baruch, Tal Ridnik, Nadav Zamir, Asaf Noy, Itamar Friedman, Matan Protter, Lihi Zelnik-Manor. ICCV 2021.
>
> [3] [CaPC Learning: Confidential and Private Collaborative Learning](https://openreview.net/forum?id=h2EbJ4_wMVq).
> Christopher A. Choquette-Choo, Natalie Dullerud, Adam Dziedzic, Yunxiang Zhang, Somesh Jha, Nicolas Papernot, Xiao Wang, ICLR 2021.
>
> [4] [Papers with Code - CheXpert Benchmark for Multi-Label Classification](https://paperswithcode.com/sota/multi-label-classification-on-chexpert).
>
> [5] [CheXpert: A Large Dataset of Chest X-Rays and Competition for Automated Chest X-Ray Interpretation.](https://stanfordmlgroup.github.io/competitions/chexpert/)

---

### Comment · Area_Chair_PUJc · 2021-11-10
**Discussion phase**

Dear Authors and Reviewers,

Let me first thank you for supporting ICLR 2022: Authors for submitting their contributions, and Reviewers for going through them and sending their comments and remarks!

As the discussion phase has just begun, let me ask Authors to answer all questions appearing in reviews and to defend your paper, and reviewers to check all other reviews to see whether you coincide and to be ready to respond to authors rebuttals.

We are also looking forward for public comments. I hope for a vivid discussion for this paper.

Best regards, AC for Paper 65

---

### Author Response · Authors · 2021-11-25
**Pending questions**

We would like to thank the reviewers for their questions and comments. The paper has definitely improved as a result. We would like to check one last time if there are any pending questions that we have not adequately addressed.

---

### Comment · Area_Chair_PUJc · 2021-11-27
**Multi-label task loss and privacy guarantees**

Dear Authors,

As AC I have briefly checked your paper and come to the question what is the relation between the task loss for which multi-label classification (MLC) model is optimized and the privacy guarantees. As in MLC we deal with label vectors, there is a multitude of task losses defined and used. They are usually of a different nature, having different properties (e.g., Bayes optimal decisions), leading to very different models. For example, the One-vs-All approach or optimizing the binary cross entropy is the right approach for Hamming loss, as marginal probabilities of labels are enough for making optimal decisions. This is however not the case of the subset 0/1 loss, for which we need to get the joint mode of the conditional distribution. This is somehow related to your binary voting and powerset mechanism, but I do not see such a discussion in your paper.

I would appreciate your response.

Best, AC for Paper 65

---

> ### Author Response · Authors · 2021-11-28
> **Multi-label task loss and privacy guarantees**
>
> Dear AC,
>
> Thank you for your question. There is no direct relationship between the task loss and the privacy guarantees of our mechanisms. We will add the following summary to Section 5 in our paper:
>
> “The canonical PATE [1], and by extension our multi-label PATE, makes no assumptions about the loss function. PATE enables heterogeneous models, each teacher can be optimized separately using different loss functions, optimization algorithms, or model architectures, so as to achieve the best per-teacher task performance. In the multi-site (distributed) setting, these teachers can be optimized on their respective datasets to be used for privately training a student model.”
>
> The above is a major advantage of PATE and our multi-label PATE. The privacy analysis in our mechanisms do not depend on the details of the machine learning techniques (e.g., batch selection, loss function, or optimization algorithm) used to train either the teachers or their student [1]; this is contrary to DPSGD [2] which represents the state-of-the-art in differentially-private deep learning. Thus, our methods enable more flexible privacy-preserving training in the multi-label scenario and outperform DPSGD.
>
> We do note that there is an indirect relation between the task loss and privacy guarantees through consensus/task performance. The DP mechanisms proposed in this paper are largely influenced by the following dynamics: (1) higher consensus between teachers leads to tighter privacy budgets and (2) better task performance leads to the better utility of the DP labels. Importantly, both can but need not be obtained simultaneously. Trivially, high consensus can be obtained with poor task performance by ensuring all models always output “0” for all labels. Consensus may be minimized despite high task performance if the teachers maximally disagree, which may be the case on highly non-i.i.d data between teachers. Figure 3 in Appendix B.4 shows that when disagreement is high, Powerset PATE outperforms Binary PATE because of the more favorable data-independent bound. On the other hand, Figure 4 in Appendix B.4 shows that when the agreement is high, Binary PATE outperforms Powerset PATE because of a more favorable data-dependent bound. Our results in Section 5 show that on realistic multi-label datasets, including those in the medical domain, our approaches outperform all others and are an important step toward practical as well as private multi-label machine learning.
>
> The task loss is a vital choice for multi-label classification as it can significantly influence the performance of the trained model under different metrics. For example, we follow [3] and use the Binary Cross-Entropy loss to train DenseNet-121 models on the medical datasets (CheXpert, MIMIC, and PadChest). We always optimize each per-teacher performance, which is made possible by our multi-label PATE setup.
>
> **References:**
>
> [1] [Semi-supervised Knowledge Transfer for Deep Learning from Private Training Data.](https://openreview.net/forum?id=HkwoSDPgg) Nicolas Papernot, Martín Abadi, Úlfar Erlingsson, Ian Goodfellow, Kunal Talwar. ICLR 2017 (Best paper award).
>
> [2] [Deep Learning with Differential Privacy.](https://arxiv.org/pdf/1607.00133.pdf
> ) Martín Abadi, Andy Chu, Ian Goodfellow, H. Brendan McMahan, Ilya Mironov, Kunal Talwar, Li Zhang. CCS 2016.
>
> [3] [On the limits of cross-domain generalization in automated X-ray prediction.](https://arxiv.org/abs/2002.02497) Joseph Paul Cohen, Mohammad Hashir, Rupert Brooks, Hadrien Bertrand. MIDL 2020.

---

### Decision · Program_Chairs · 2022-01-20

**Decision:**

Reject

**Comment:**

This is an interesting paper discussing differential privacy for multi-label classification. The initial reviews rated the paper with rather extreme scores, therefore I have invited an additional reviewer. This review did not clarify the issues raised by the most critical reviewer, but pointed out that the goal of showing how DP can be enforced in MLC is not fully obtained as there is a lack of the discussion concerning the MLC performance. This is also a problem raised in my comments. Taking this into account, I need to state that the paper is not ready for publication.